# Insights into 3D cloud radiative transfer effects for OCO-2

Steven. T. Massie[1], Heather Cronk[2], Aronne Merrelli[3], Sebastian Schmidt[1], and Steffen Mauceri[4]

[1]Laboratory for Atmospheric and Space Physics, University of Colorado, Boulder, Colorado, 80303, USA
[2]Cooperative Institute for Research in the Atmosphere, Colorado State University, Fort Collins, Colorado, 80523, USA
[3]Department of Climate and Space Sciences and Engineering, University of Michigan, Ann Arbor, Michigan, 48109-2143, USA
[4]Jet Propulsion Laboratory, California Institute of Technology, Pasadena, California, 91109, USA

*Correspondence to:* Steven T Massie (Steven.Massie@lasp.colorado.edu)

**Abstract.** Clouds impose radiance perturbations upon Orbiting Carbon Observatory (OCO-2) measured spectra. The Spherical Harmonic Discrete Ordinate radiative transfer Method (SHDOM) code is applied in both idealized bar cloud and scene specific calculations of 1D and 3D radiances in order to understand 3D cloud effects for a wide range of gas vertical optical depths, solar and sensor viewing geometries, for ocean and land scenes. SHDOM calculations for 36 scenes over the Amazon and the Pacific are co-analyzed with Moderate Resolution Imaging Spectroradiometer (MODIS) radiance-based cloud distance data, and OCO-2 Lite file rawXCO2 for both Quality Flag=0, (QF0, best quality) and Quality Flag=1 (QF1, poor quality) data. SHDOM calculations of the ocean and land scenes indicate that the 1D / 3D radiance intensity ratios and rawXCO2 decrease concurrently as nearest cloud distance decreases towards zero, especially for the ocean glint QF1 data, which provide the clearest evidence of 3D cloud effects in OCO-2 retrievals. Yearly analysis of OCO-2 $O_2$ A-band continuum radiances indicate that 3D cloud-brightening events predominant over cloud-shadowing events, therefore 1D / 3D intensity ratios are predominantly less than unity. Bias corrected (bcXCO2) at cloud distances between 0 and 20 km are calculated for 20° latitude bands for 2015-2018. These zonal averages are used to calculate 3D cloud effect biases for bcXCO2 data (with a positive bias indicating that OCO-2 underestimates bcXCO2). Averages of 3D cloud effect biases, weighted by the number of Lite file data points in each of the nearest cloud distance bins, in the northern and southern hemispheres, are 0.16 (1.31) and 0.26 (1.41) ppm, respectively, over the ocean, and - 0.13 (0.51) and -0.08 (0.47) ppm over land, for QF0 (QF1) data.

## 1 Introduction

The Orbiting Carbon Observatory (OCO-2) measures the column-averaged atmospheric $CO_2$ dry air mole fraction, referred to as XCO2, on a global basis (Eldering et al., 2017). The project measurement goal is to obtain measurements of XCO2 to the 0.25 % level, corresponding to a XCO2 accuracy of 1 ppm, since ambient XCO2 is near 410 ppm. Biases in XCO2 on regional scales as small as a few tenths of a part-per-million (ppm) in XCO2 can lead to spurious values of inferred $CO_2$ fluxes (Chevallier et al., 2010).

The OCO-2 satellite is in a sun-synchronous 98° inclination polar orbit, and is comprised of three spectrometers centered in the $O_2$ A-band (0.76 μm), weak $CO_2$ band (WCO2, 1.6 μm), and strong $CO_2$ band (SCO2, 2.06 μm). Spectral resolution $\lambda / \Delta\lambda > 17,000$ (Crisp et al., 2017) ensures

that individual spectral lines are observed. Each band contains 1016 data points, covering a wide range of optical depth. The OCO-2 experiment has several observing modes: ocean glint, land glint, land nadir, and target mode. In target mode the spectrometers are commanded to observe a specific small geographical region. In ocean glint observations the bright ocean glint spot is utilized to increase the observed radiance level. Land glint observations utilize glint observing geometry, and are not restricted to water surfaces. The land nadir observations have a sensor view angle near 0.3°. We focus on ocean glint, land glint, and land nadir scenes in this paper.

The ACOS (V10) retrieval of $CO_2$ is based upon the measurement of surface pressure in the $O_2$ A-band, and $CO_2$ absorption in the WCO2 and SCO2 bands. The retrieval applies the optimal estimation retrieval methodology of Rodgers (2004). The operational retrieval (OCO-2 L2 ATBD, 2020; O'Dell et al., 2018) solves for a state vector with many elements including XCO2, surface pressure, reflectance, and aerosol. Spectroscopic line cross sections for $O_2$, $CO_2$, and $H_2O$ are specified by the ABSCO V5.1 data files (Payne et al. 2020).

XCO2 generated by the operational retrieval is referred to as *raw* XCO2 (rawXCO2). A post-retrieval processing step then bias corrects the raw XCO2, yielding *bias corrected* XCO2 (bcXCO2). Bias correction (O'Dell et al, 2018) is achieved by comparing rawXCO2 to truth proxies: ground based XCO2, measured by the Total Carbon Column Observation Network (TCCON, Wunch et al. 2017), ensemble model XCO2, and small area analysis (in which differences of XCO2 values and the area average are calculated). The differences between the raw XCO2 and the truth proxies are related in a linear manner to several bias correction parameters (dP over the ocean, dPfrac over land, CO2graddel, and DWS). dP is the difference (in hPa) between the retrieved (Pretrieved) and a priori (Papriori) surface pressure evaluated at the strong SCO2 band geographic location, while dPfrac (in ppm) is

$$dPfrac = raw\ XCO2\ (1.00 - Papriori / Pretrieved)\qquad(1)$$

CO2graddel is a measure of the difference in the retrieved and prior $CO_2$ vertical gradient. DWS is the sum of the vertical optical depths of the dust, water, and sea salt aerosol components. The bias correction process takes the raw XCO2 and increases these values by approximately 2 ppm. The Version 10 OCO-2 Data Product User's Guide (2020) discusses the details of the raw to bcXCO2 equations, which are dependent upon footprint-land-ocean specifics. The need to bias correct is due to instrument calibration, spectroscopic line uncertainty, and physics not included in the operational retrieval code. Uncertainty quantification of OCO-2 measurements is discussed in Connor et al. (2016) and Hobbs et al. (2017).

Raw and bc XCO2, bias correction variables, and other data are conveniently contained in OCO-2 "Lite" files, with one Lite file per day that includes all daily operational retrievals. Data quality is indicated by Quality Flag=0 (QF0, best quality) and Quality Flag= 1 (QF1, poor quality) data flags. The OCO-2 team discourages use of QF1 data in XCO2 studies. In this paper we do examine QF1 data in addition to the QF0 data since the QF1 data provides insights into 3D cloud radiative effects.

Of the approximately one million daily observations which are collected by OCO-2, about 25% are passed into the operational retrieval due to prescreening for scenes contaminated by clouds and heavy aerosol loadings. Two cloud preprocessors (Taylor et al. 2016) exclude many of the soundings. One preprocessor only uses the $O_2$ A-band to provide a computationally quick determination of $O_2$ A-band surface pressure, which is compared to a priori meteorological data. An observation is excluded from the operational retrieval if the difference in surface pressure is greater than 25 hPa. The second preprocessor performs single band retrievals of XCO2 using both the WCO2 and SCO2 bands independently. If the ratio of the two columns deviates significantly from

unity, then the observation is excluded from the operational retrieval. This often identifies scenes with aerosol contamination due to the spectral dependence of aerosol absorption.

Not all known physics, however, is included in the operational retrieval. The OCO-2 operational retrieval does not know if clouds are outside of the row (frame) of the eight side by side detector footprints. The OCO-2 orbital track sweeps out a continuous swath of footprints with a swath width less than 20 km (see Fig. 2.2, OCO-2 L2 ATBD, 2019). The detailed shape of an individual footprint, on the order of 2 km, varies according to viewing geometry (Crisp et al. 2017).
Clouds outside of the swath can scatter photons into the region of the footprints but the OCO-2 experiment cannot determine the location of clouds outside of the swath. Until the wall-clock advantages of parallel computing can be implemented in an operational setting, 3D cloud effects will remain computationally expensive in an operational setting.

      In this paper we utilize MODIS radiance data at 250 m resolution to study 36 scenes in detail.
The 250 m radiances and MODIS cloud mask data is used to specify the locations of clouds in the vicinity of OCO-2 observations. The 36 scenes include 12 ocean glint, 12 land nadir, and 12 land glint scenes. A visual examination of NASA Worldview (https://worldview.earthdata.nasa.gov) MODIS Aqua imagery, and a listing of Lite file latitude, longitude, and QF flags suggested scenes in which several dozen Lite file $XCO2$ values are present in each scene. Many of the scenes have
sun-cloud geometry in which light is reflected off of a cloud feature oriented approximately perpendicular to the incident solar beam (Figure 6 is an example), with clear-sky gaps between the clouds. This geometry is conducive to study 3D cloud effects. For global analyses we use the nearest cloud distance files discussed in Massie et al. (2021), which are available at Zenodo (referred to as "3D metric files", https://doi.org/10.5281/zenodo.4008765). As discussed in Massie
et al. (2021), the nearest cloud distance data is based upon an analysis of auxiliary files (Cronk 2018) that contains MODIS 500 m radiances, cloud mask and geolocation information matched to OCO-2 observation geolocation.

      Previous studies of 3D radiative transfer (Merrelli et al. 2015) applied the Spherical Harmonics Discrete Ordinate Method (SHDOM) 3D radiative transfer code (Evans 1998) to perturb OCO-2
like spectra, and looked at OCO-2 like retrievals with and without the 3D radiance perturbations. Retrieved $XCO2$ values were *lower* than clear sky retrievals by 0.3, 3, and 5-6 ppm for surfaces characterized by bare soil, vegetation, and snow-covered footprints.

      Massie et al. (2021) calculated differences in $XCO2_{TCCON}$ and $XCO2_{Lite\ file}$ for QF0 and QF1 data for several years of OCO-2 measurements. Denoting $XCO2_{TCCON} - XCO2_{Lite\ file}$ as $Res_D$ at
distance D from the nearest cloud, residual differences ($Res_{10\ km} - Res_D$) varied between 0.0 and 0.4, and 0.0 and 2.5 ppm, for QF0 and QF1 data as nearest cloud distance D varied from 10 km to 0.5 km. The residuals (with positive values indicating a retrieval *underestimate* of $XCO2$) are present in both raw and bc $XCO2$, and therefore are residuals of 3D cloud effects that are not accounted for by the 1D operational retrieval framework. Massie et al. (2021) also demonstrated
that an extension of the *linear* bias-correction methodology, in relation to adding 3D cloud metrics to the bias correction parameters, only marginally improved the accuracy of bias corrected $XCO2$.

      The recent Emde et al. (2022) study discusses 3D calculations that covers a range of cloud scenes over Germany and surrounding countries, based on a large eddy simulation, and in addition a set of box-clouds with various solar zenith and viewing angles and optical depths. The
calculations were carried out in the $400 - 500$ nm and $O_2$ A-band spectral ranges. The Figs. in Emde et al. (2022) for the box cloud calculations are particularly instructive as they illustrate how reflectance varies as a function of distance from cloud edges for viewing geometry in which clouds enhance the radiance field and for viewing geometry in which cloud shadow effects are present.

The Emde et al. (2022) calculations are available at https://doi.org/10.5281/zenodo.5567616
(Emde, 2021). The study of Emde et al. (2022) motivated us to calculate a set of SHDOM idealized bar cloud 1D and 3D calculations.

This paper is follow-on work of Massie et al. (2021) and expands the previous research in significant ways. Massie et al. (2021) focused on quantifying 3D cloud effects based on comparisons of bcXCO2 and TCCON data. The geographical distribution of TCCON sites, however, is concentrated over north America and Europe, with sparse coverage over the tropics and over the ocean. The current paper calculates 3D cloud biases as a function of latitude using the Zenodo "3D metric" files for 275 times more data points. *The latitudinal dependence of 3D cloud biases is not addressed in Massie et al. (2021), but addressed in this paper.* Massie et al. (2021) demonstrated that 3D cloud biases frequently indicate an underestimation of XCO2 as nearest cloud distance decreases. The current paper provides a physical reason why this is the case. Massie et al. (2021) examined one 32 km x 32 km scene, while this paper examines 36 scenes over the ocean and land for a variety of viewing geometries over the Amazon and the oceans, areas which are problematic for OCO-2. The current paper presents side by side graphs of 1D / 3D intensity ratios (and other variables) as a function of nearest cloud distance to illustrate the non-linearities that are present in the OCO-2 data files. The non-linearities that are present in our graphs for several variables are non-linearities which Machine Learning (ML) bias correction methods (see Mauceri, Massie, and Schmidt 2022) will need to mitigate.

This paper is organized in the following manner. The data utilized in our study is discussed in Sect. 2. Details of the SHDOM 1D (IPA, independent pixel approximation) and 3D radiance calculations are reviewed in Sect. 3.1. In Sec. 3.2 SHDOM idealized bar cloud calculations are discussed, which provide insight as to the variation of 1D / 3D intensity ratios as a function of nearest cloud distance, gas vertical optical depth, solar zenith angle, cloud height, and surface reflectance. Partial derivatives in radiance with respect to changes in pressure, XCO2, reflectance and aerosol total optical depth are presented in graphical form in Sect. 3.3 to illustrate the zero-order physics associated with 3D cloud effects. In Sect. 4 individual scenes over the ocean and land are discussed to illustrate how the 1D / 3D radiance intensity ratio varies near and far away from clouds. In Sect. 5 SHDOM calculations for 36 scenes (12 ocean glint, 12 Amazon land nadir, and 12 Amazon land glint) are discussed. The QF0 and QF1 observations illustrate the relationships between nearest cloud distance and key variables (1D / 3D SHDOM radiance intensity ratios, raw XCO2, dP, surface reflectance, and aerosol optical depth). In Sect. 6 yearly calculations of dP and raw XCO2 are presented as a function of latitude and as a function of nearest cloud distance. Sect. 7 discusses calculations of zonal averages of 3D cloud radiative effect biases as a function of latitude for bcXCO2 over ocean and land. Sect. 8 presents our summary and conclusions.

## 2 Data

OCO-2 product files are available from the NASA Earthdata website (https://earthdata.nasa.gov/) and archived in the GES DISC repository. The first part of specific Version 10 (V10) file path names are given here in parenthesis to identify, in a general sense, the files used in our study. OCO-2 "Lite" files (oco2_LtCO2_..) contain raw and bcXCO2, bias correction variables (such as dP), and other information. The land / water flag is used in our study to identify ocean and land observations, and the operational mode flag is used to identify glint and nadir observations. Preprocessor (oco2_L2ABP_.., referred to as L2ABP) files specify cloud flags and continuum

radiances for all measurements (including those which are not successfully retrieved). Level 2 diagnostic files (oco2_L2Dia_..) specify full spectra in all three bands. Meteorological (oco2_L2Met..) surface pressure is specified by GEOS-5 FP-IT data, which is used to calculate the dP values. The ND characters in the granule-level file path names refer to nadir observations, while glint files contain GL as part of the file path name.

Co-analysis of MODIS and OCO-2 data is made possible by way of MODIS 250 m radiances (MYD02QKM..), geolocation (MYD03..), aerosol (MYD04..), cloud data (MYD06..), and weekly surface reflectance (MYD09..). The MY.. prefix indicates MODIS Aqua data files. OCO-2 flies six minutes in front of MODIS Aqua in the NASA "A-train", and therefore the MODIS cloud field in six minutes may differ slightly from the cloud field that impacts OCO-2 observations. The

MYD06 data file specifies cloud heights for a given scene. Cloud heights vary between 0.6 and 3.7 km for the scenes examined in this paper, with 1.4 km being the average.

The 3D metric files contain nearest cloud distance and other 3D cloud metrics which are discussed in Massie et al. (2021). The other metrics (H(3D), H(Continuum), and CSNoiseRatio) are based upon calculation of standard deviations of the MODIS radiance field, OCO-2 continuum

radiance field, and sub-footprint radiance fields, respectfully. These metrics are measures of *radiance spatial gradients*, which will be non-zero in the presence of clouds. In this paper we focus on nearest cloud distance since Várnai and Marshak (2009) demonstrated that MODIS reflectance at various wavelengths between 0.47 and 2.13 µm increases the closer one is to clouds (i.e. nearest cloud distance is a previously proven 3D cloud effect metric). There are 3D metrics

for each successful OCO-2 QF0 and QF1 retrieval for 2014-2019.

## 3 Calculations

### 3.1 The SHDOM radiative transfer code

The Spherical Harmonic Discrete Ordinate radiative transfer Method (SHDOM) 3D radiative transfer code (Evans, 1998; Pincus and Evans, 2009) calculates 1D (single column, independent pixel approximation) and 3D fields. In the 1D calculation there is no exchange of photons between adjacent columns. In the 3D calculation columns do exchange photons. Of particular interest to

215 our paper is the scattering of photons from cloud and ground to adjacent columns. This exchange of photons between columns yields a 3D Stokes radiation field which differs from the 1D Stokes field. The four components (I, Q, U, V) of the Stokes field can be used to define the amount of linear and circular polarization of the radiation field. We focus on the total intensity I component of the Stokes vector.

Input to SHDOM includes specification of the vertical (z-axis) gas optical depth structure, and the x-y-z mass content structure of the cloud and aerosol fields. In our calculations the horizontal x-y grid has a grid spacing of 0.5 km, and the input gas files have a 1 km grid spacing from ground to 30 km altitude. The number of x and y axis grid points is 64, so the full SHDOM x-y grid covers a 32 km by 32 km area. Since the cloud and aerosol input file has an altitude grid at sub-1 km

spacing near the ground, SHDOM interpolates the gas field extinction / km values to the sub-1km vertical grid of the cloud and aerosol input file. Gas field extinction / km values are calculated by utilizing the OCO-2 V5.1 ABSCO data files (Payne et al, 2020). The ABSCO molecular cross sections for $O_2$, $CO_2$, and $H_2O$, specified at a resolution of 0.010 cm$^{-1}$ (5.9 10$^{-7}$ µm in the $O_2$ A-band), are tabulated as a function of 64 pressures and 17 temperatures. Extinction / km values are

calculated for each x-y-z grid point according to each grid point temperature and pressure. The

SHDOM calculations do not iterate for the surface reflectance. A constant Lambertian surface reflectance in each band for land observations is specified (hardwired as an input to SHDOM) based on the Lite file retrieved values. These values produce SHDOM 3D top of atmosphere reflectance in good agreement with the observed (archived Lite file) reflectance. For ocean glint observations the Mishchenko and Travis (1997) implementation of the Cox-Munk windspeed dependent surface reflectance formulation is used in the SHDOM calculations, with windspeed specified (hardwired as an input to SHDOM) based on the Lite file retrieval of the windspeed.

For a specified wavelength, SHDOM calculates Rayleigh scattering, aerosol, and cloud optical parameters (optical depth, asymmetry parameter, and single scattering albedo) for each x-y-z grid cell. For the scenes discussed in this paper the aerosol and cloud radii are 0.1 and 10 μm, respectively. Water droplet clouds have a cloud base of 0.1 km up to a specified cloud height (e.g 1.4 km), with a vertically constant cloud droplet liquid water content (LWC, in g /m$^3$ units). The aerosol has equal valued extinction / km values from ground level to 1.8 km altitude. The input aerosol mass content values are adjusted such that the total vertical aerosol optical depth is near a desired value (in the 0.05 - 0.16 range, and usually less than 0.1). Since the aerosol optical depth is small, surface reflectance should have more influence on the top of atmosphere radiance than aerosol. SHDOM calculates the aerosol and cloud optical depth parameters by applying a Mie code based on the input cloud and aerosol file, and for aerosol the complex index of refraction is selected based on a sulfate aerosol.

SHOM is configured to write out the Stokes field at the top of the atmospheric grid for a set of sun-observation azimuth angles, solar zenith angle, sensor view angle, for 17 total vertical gas optical depths. For the O$_2$ A-band the gas optical depths vary from 8 10$^{-4}$ to 4.0. The lowest gas optical depth in each band corresponds to the band continuum.

The observed radiances are directly related to the SHDOM 3D radiances, since actual atmospheric radiances are the result of 3D radiative transfer processes. The 1D radiance field is not available from OCO-2 measurements, though the 3D radiances approach the 1D values as cloud distances become very large. Also, the OCO-2 radiances are dependent upon a linear combination of the I, Q, and U Stokes components (see equation 3-40, OCO-2 L2 ATBD, 2020). We focus on the SHDOM total intensities to gain insights in regard to 3D cloud effects for a variety of scenes, and do not make detailed comparisons of observed spectra and the SHDOM spectra.

### 3.2 Idealized bar cloud calculations

The choice of idealized bar cloud calculations is motivated by visual examination of various NASA Worldview scenes over the ocean and the Amazon. Amazon clouds are frequently distributed in "cloud streets" (Fig. 6 is a good example). Ocean scenes, in which an elongated cloudy area is associated with adjacent clear sky regions, are useful to study 3D cloud effects (Fig. 7 is a good example). The cloud distributions in both scenes can be geometrically approximated by one or more idealized bar clouds.

Figure 1 illustrates a pair of idealized bar clouds, referred to as *left* and *right* bar clouds. The x width of each cloud is 3 km, and the bar clouds extend the full y-axis of the 32 km by 32 km scene. The clouds are assigned a specific solar zenith angle and cloud altitude (1.4 km). The sun is along the x-axis and SHDOM is configured to calculate for 12 sensor azimuth angles from 0 to 360° in 30° steps. Figure 1 illustrates one of the 12 azimuth angles, with the sensor to the right of the right bar cloud. The inclusion of the two clouds in the scene allows for analysis of 1D and 3D radiances that result from scattering of photons off the right bar cloud back towards the observation

footprint (Obs), and for the case in which (for the left bar cloud) a cloud shadowing effect is present if the observation point is close to the left bar cloud. The spatial extent of the cloud shadowing is dependent on the height of the cloud and the solar zenith angle. Analysis of the 1D and 3D radiances as a function of distance D from the bar clouds yields insights into the nature of 3D radiative effects, as a function of gas vertical optical depth.

The 1D / 3D ratio, and its variation in a scene, is fundamental to this paper, since it is a measure of the size of the 3D cloud effect in the scene. Figure 2 presents 1D / 3D ratios as a function of cloud distance D from the right bar cloud for all three OCO-2 bands. Land nadir geometry is applied, with a solar zenith angle of 38° (with atmospheric model specifics associated with the Amazon scene discussed in Fig. 6 below). The curves pertain to a low gas vertical optical depth near 0.01. The 1D / 3D ratios approach unity as the cloud distance increases. The ratios for the optically thin regime however are not equal to 1.0 at the largest cloud distance since photon paths are present in the 3D case in which light propagates into the top of the cloud and exits out the sides, adding to the 3D radiances in the regions between the clouds. As the cloud distance D decreases towards zero the 1D / 3D ratio becomes small, near 0.8. *Notice that this drop off in the ratio is very non-linear, and takes place at cloud distances approximately less than 4 km.* Also notice that the curves for the three bands are quantitatively different. This implies that a detailed understanding of 3D radiative effects requires attention to the details in each of the three OCO-2 bands.

From Fig. 2 it is apparent that 1D /3D ratios asymptote for a length scale of approximately 10 km. The periodic boundary conditions used by SHDOM therefore do not cause clear- sky pixels near the left cloud to be impacted by photons that, after being scattered by the right-side cloud, move across the right edge of the scene and reappear at the left edge. These considerations motivated our selection of the Fig. 1 geometry and selection of a 32 km by 32 km SHDOM grid.

Considering the case where the 1D /3D ratio is less than unity due to 3D cloud effects, the OCO-2 experiment measures the true radiance, which is a 3D radiance since the real atmosphere exchanges photons between adjacent columns. The operational OCO-2 retrieval calculates 1D column radiances, and inserts no physics due to adjacent column 3D cloud effects. With 1D / 3D less than unity, the retrieval needs to "enhance" the 1D radiance by modifying variables (such as surface reflectance, aerosol, surface pressure, and XCO2) in order to bring forward model and observed radiances into agreement. This is illustrated below in Sect. 5 in relation to Figs. 8, 9, and 10.

Figure 3 displays 1D / 3D ratios as a function of cloud distance D for the left and right bar clouds for the $O_2$ A-band continuum (gas optical depth near 0.008). The sun and sensor are along the -x and +x branches of the x axis. For the left bar cloud (Fig. 3a) the ratio decreases slightly as cloud distance decreases from large values, then increases drastically (towards 5.4) as distance D approaches zero. This behavior is due to *cloud shadowing* effects. For graphical convenience the 1D / 3D cloud shadowing ratios greater than 1.1 are set to a maximum value of 1.1, and the minimum ratio of 0.45 was set to 0.70, so that y axis ranges are the same in both panels. The lowest 1D/ 3D ratio in Fig. 3a is due to an increase in 3D photons originating (leaking from) the adjacent cloud column. The 1D radiance is not susceptible to cloud shadowing since cloud shadowing originates from the sunward-adjacent column. The 3D radiance is susceptible to the shadowing from the adjacent column. 3D radiance fields are therefore susceptible to both cloud shadowing (dimming) and cloud brightening effects (Fig. 3b). The prevalence of cloud brightening versus cloud shadowing effects is discussed in Sect. 4.

Figure 4 indicates the sensitivity of 1D / 3D ratios to gas vertical optical depth, solar zenith angle, cloud height (cloud vertical extent), and surface reflectance. The curves are those from the cloud brightening calculations for the right bar cloud. The largest sensitivity is due to the gas optical depth (Fig. 4a). The sensitivity is largest for the smaller nearest cloud distances. As the gas optical depth increases the ratios become closer to unity, and the curves drop-off to lower ratios at increasingly smaller cloud distances. For the case where optical depths become very large the ratio approaches unity for all cloud distances. This is reasonable since at very large gas optical depths the vast majority of the photon paths are located at large heights above the surface, and these photons do not interact with the low-level clouds.

The second largest sensitivities are due to solar zenith angle (Fig. 4b) and band cloud height (Fig. 4c). The sensitivity to solar zenith angle is reasonable since for larger solar zenith angles photons are scattered off of the sides of the clouds, while for a solar zenith angle near zero the 3D radiative effect is constrained by photon paths passing through the top of the cloud (followed by some exiting of photons to the side) and/or secondary paths (sun to surface, surface to cloud, cloud back to surface).

In Fig. 4c the sensitivity to cloud thickness is illustrated (labeled by the cloud top height) for a cloud base of 0.1 km. As the vertical extent of the cloud is increased there is more side surface area present, increasing the number of sun to cloud to surface photon events. As noted by Taylor et al. (2016), the cloud preprocessor does a good job in screening for clouds, though the preprocessor can pass some cases in which low-level clouds are present.

Fig. 4d indicates that 1D / 3D ratios are not sensitive to surface reflectance for the land nadir view geometry. Since the 1D path (sun to surface to sensor) is dependent on the surface reflectance, and the 3D situation (with added sun to cloud to surface to sensor paths) is also dependent on the surface reflectance, some cancellation in the surface reflectance term is expected. 1D / 3D ratios were also calculated for several cloud LWC values (Fig. 4e), sulfate, sea salt (Fenn et al. 1985) and brown carbon aerosol (due to biomass fires, Alexander, Crozier and Anderson, 2008) (Fig. 4f). *While there are important variations in the details shown in Fig. 4, all panels display a noticeable non-linear decrease in 1D /3D ratios as nearest cloud distance decreases.*

A version of Fig. 4 for left bar clouds (not shown) has curves that differ from the right bar cloud calculations of Fig. 4 in the same way that Fig. 3a and Fig. 3b differ. The 1D / 3D intensity ratio curves decrease as nearest cloud distance decreases, then each curve begins to increase towards values greater than unity a few km from the shadow side of the left bar cloud. The nearest cloud distance for these inflection points varies from 2 to 4 km for solar zenith angles between 20°and 50° solar zenith angle (SZA). This is reasonable since the cloud shadow extends a distance $H_{cld}$ tan(SZA) for a cloud height in km of $H_{cld}$. A comparison of the two graphs indicates that the left bar curves are closer together (in y axis separation) for the four solar zenith angle curves than the solar zenith angle curves in Fig. 4, while the LWC curves have more y axis separation near the inflection points. The Zenodo site (https://doi.org/10.5281/zenodo.7655136) contains numerical data for Fig. 4 and its left bar cloud equivalent.

### 3.3 Radiance perturbation sensitivity

Figure 5 illustrates the sensitivity of radiances to perturbations in XCO2, surface pressure, surface reflectance, and total optical depth. The x axis variable specifies the total vertical optical depth (gas + aerosol) for each band, with each vertical optical corresponding to a specific ABSCO wavelength. while the y axis variable specifies the radiance perturbation. Lowest vertical optical

depths are in the continuum portion of the spectra, while the largest vertical depths are in absorption lines. The information in Fig. 5 is calculated from monochromatic high spectral-resolution ABSCO data. Monochromatic radiance derivatives are presented in Fig. 5.

Denoting R as the radiance intensity, the 3D curves are 100 ( $R_{3D} - R_{1D}$) / $R_{1D}$ values in each panel, where 3D and 1D refer to SHDOM 3D and 1D calculations. The model atmosphere (temperature profile, gas, aerosol, and cloud optical depth structure) is the same in the 1D and 3D curves, but the 3D radiances are those due to SHDOM accounting for the exchange of photons between columns. Once specified, the model atmosphere is fixed in a SHDOM simulation. The observation point (see Fig. 1) is 4 km leftward of the right bar cloud, with the sun (sensor) to the left (right) of the observation point. The other curves are partial derivatives of the 1D radiances, 100 ($R_{1D\ perturbation\ case} - R_{1D\ baseline\ case}$) / $R_{1D\ baseline\ case}$. The model atmosphere is the same in the 1D baseline and 1D perturbation cases, except for a perturbed value in one variable. Baseline conditions and perturbations are specified in Table 1. The solar zenith and sensor view angles are 38° and 0°, respectively, in all of the calculations. The atmospheric profile corresponds to the Amazon scene associated with Fig. 6 that is discussed in Sect. 4. Sulfate aerosol extends from ground to 1.8 km, with a total aerosol optical depth near 0.067 in the $O_2$ A-band, cloud LWC is 0.30 g / $m^3$, and the idealized bar clouds have a cloud top at 1.4 km.

Figure 5 illustrates that an *increase* in radiance comes about (in a partial derivative sense, with other variables held constant) if surface reflectance or aerosol optical depth is *increased,* if the surface pressure is *decreased*, or if XCO2 is *decreased* in the WCO2 and SCO2 bands. *Away from clouds (in the absence of radiance dimming due to cloud shadows), the 3D effect increases radiance.* Since the 3D effect is substantial at all optical depths, it is expected that the retrieval will definitely adjust surface reflectivity and/or aerosol in the state vector, since these variables have radiance partial derivatives that are also non-zero at all optical depths. The surface pressure perturbations in all three bands, and the WCO2 and SCO2 XCO2 perturbations, have radiance partial derivatives that are small at small gas optical depths, and appreciable only at the larger vertical optical depths. Figure 5 does not indicate how the 3D radiance perturbation is accounted for in the operational retrieval by perturbations in surface reflectance, aerosol, surface pressure, and XCO2. This question is addressed below in Sect. 5.

Figure 5 illustrates that OCO-2 observations are susceptible to 3D cloud effects. The 3D effect is present in all three bands, increasing 1D continuum (smallest optical depth) radiances by 3 %, 2 %, and 1% in the $O_2$ A-band, WCO2, and SCO2 bands when the observation point is 4 km from the right bar cloud. The spectral variations of the 3D radiance perturbations in Fig. 5 are *distinct* (different than the other perturbations), which forces adjustments in the retrieval state vector variables. The 3D perturbations are larger (smaller) for smaller (larger) cloud distances. As stated in the Introduction, the OCO-2 measurement goal is to measure XCO2 to the 1 ppm level. Perturbations in XCO2 of 1 ppm, however, perturb radiances on the 1 % level only at large optical depths, and less so at smaller optical depths. If the 3D radiance perturbations were substantially smaller than the radiance perturbation corresponding to a 1 ppm increase in XCO2, and if the spectral variations of the 3D radiance perturbations were not different than the other perturbations, then the OCO-2 observations would not be susceptible to 3D cloud effects, but this is not the case. Since 40% of all OCO-2 observations are within 4 km of clouds, 3D cloud radiative effects impact many OCO-2 observations (Massie et al. 2021).

The wavelengths selected in Fig. 5 are representative. A different set of wavelengths would produce derivatives, especially for the pressure and $CO_2$ derivatives in the SCO2 band at optical depths greater than two, that differ from those shown in Fig. 5. The key point of Fig. 5 is that the

pressure and $CO_2$ derivatives are negative, ranging from 0% to -1%, and are of similar absolute size to the 3D radiance perturbations, which vary from 0% to 3% for an observation 4 km from the nearest cloud. Figs. 4 and 5 are the only figures in this paper that presents information that relates to non-continuum wavelengths.

Fig. 5 illustrates the zero-order physics associated with 3D radiative transfer. The discussion then needs to proceed to ask how the operational retrieval responds to the 3D radiance enhancements due to 3D cloud brightening effects. Since the operational retrieval does not insert any 3D radiative transfer physics into the retrieval, there is a needed adjustment of state vector element values to bring forward model and observed radiances in agreement. The operational retrieval obtains a solution state vector with specified surface pressure, surface reflectivity, aerosol, and XCO2 which brings forward model radiances in line with observed radiances. There is no reason to assume that only the surface pressure, surface reflectivity, and aerosol state vector variable values are numerically adjusted by the forward model, yielding the needed radiance enhancement due to 3D cloud radiance brightening, with XCO2 not also being numerically adjusted by the forward model, with contributions to the needed radiance enhancement.

## 4 Amazon and Ocean Glint scenes

Figure 6 presents the detailed MODIS radiance field for a scene over the Amazon on 22 June 2015. The direction of the incident sunbeam is from the northwest at the solar zenith angle of 38°, while the OCO-2 sensor angle is 0°. The altitude, pressure, temperature model atmosphere used in the simulation is derived from the oco2_L2MetND file, with specifics listed in Table 1. Clouds are specified by analysis of the MODIS MYD02QKM 250 m radiance, MYD03 geolocation, and MYD06 cloud fields. The MYD06 cloud field identifies some clouds, and these clouds are used in conjunction with the MYD02QKM radiances to establish a cloud radiance threshold. The MYD06 cloud field, however, does not identify all clouds in a scene. This is apparent by examining the MODIS 250 m radiance field and the MYD06 cloud field (Massie et al. 2017). Once the cloud radiance threshold is established from examination of the MODIS radiance field, clouds are assigned to all x-y grid points if the MODIS radiance is greater than the threshold value. In Fig. 6a clouds are present when the radiance is greater than 80 W/m$^2$/sr/μm.

Note that the locations of the Lite V10 file data points are *between the clouds* and are indicated by the square ☐ and X symbols for the QF0 and QF1 retrievals. The cloud preprocessor does a good job in screening for observations over clouds and/or the operational retrieval does not converge successfully for these data points.

Table 2 (first column) presents statistics for the latitude range 10° S to 3° N on 22 June 2015, which includes many more data points than those displayed in Fig. 6a. Of 5162 OCO-2 observations, 589 observations (11%) were successfully retrieved, with 40% and 60% QF0 and QF1 retrievals, respectively. Approximately 80% of the retrievals are located within 4 km of clouds.

The prevalence of cloud brightening versus cloud shadowing effects for the Amazon scene is revealed in Table 2. Table 2 specifies the percentage of total retrievals which are associated with cloud shadows, assuming that the cloud heights are 2, 4, 6, or 8 km. The percentages for each cloud height are calculated based on the algorithm described in Appendix A. The algorithm utilizes O2ABP preprocessor cloud flags to identify clouds and clear observations and OCO-2 Level 1B data files that specify O$_2$ A-band continuum radiances. The continuum radiances and cloud flags are used together to specify clear and cloudy radiance thresholds.

Of the 589 successful retrievals for the Table 2 Amazon 150622 case, only eight retrievals (1.3%) are associated with shadows, and the other retrievals (100% – 1.3% = 98.7%) are associated with cloud brightening, assuming that all cloud heights are 8 km in vertical extent. The percentages are less for the assumed lower cloud heights. The retrievals associated with cloud shadows are QF1 data points, while Table 2 indicates that retrievals associated with cloud brightening have

QF0 percentages between 34% and 60% for the various cases. The additional columns of Table 2 indicate that the Pacific Glint observations (the 12 June 2016 (160622) case, discussed below), and yearly averaged percentages over the Amazon and Pacific in 2016 are less than 4%, even if all cloud heights are 8 km. Cloud heights, however, are less than 8 km. Application of NASA Giovanni (https://giovanni.gsfc.nasa.gov/giovanni/) analysis of MODIS MYD08 data files yields

histograms (not shown) of cloud top temperatures and pressure means which correspond to cloud top heights between 1 and 2 km for the 150622 and 160622 cases, and heights between 2 and 3 km for the 2016 Amazon and Pacific yearly averages. *Cloud brightening therefore is prevalent compared to cloud shadowing.*
       Figure 6b presents the SHDOM calculation of 1D / 3D ratios for the Amazon scene for a gas

vertical optical depth of 0.0008 (an optical depth in the $O_2$ A-band continuum). Since the sunbeam direction is from the northwest, the sunward side of clouds are located on the northwest side of the clouds. The V10 Lite file soundings have 1D / 3D ratios in the 0.56 – 0.96 range, with an average of 0.91. *Successful OCO-2 retrievals are therefore susceptible to significant 3D cloud radiative perturbations on the order of 9 %.*

Figure 7 presents a glint scene at 10° N on 12 June 2016 over the Pacific with a solar zenith angle of 24° and sensor view angle of 19°. Square □ and X symbols mark the locations of the Lite file QF0 and QF1 retrievals in the MODIS radiance field. The date is near the summer solstice and the sun beam direction is from the northwest. More QF0 data (the square symbols) are located to the south of the cloud in the center of the frame, with fewer square symbols on the sun-reflective

side of the cloud (the region northwest of the cloud). This situation is not, however, generally the case, since a visual examination of Figs. similar to Fig. 7 for the other ocean glint scenes listed in Table 3 did not show this behavior. An examination of the NASA Worldview imagery for the Fig. 7 scene did indicate that there are more very small "cloud remnants" north of the main cloud region with a very clear region south of the main cloud. The visual examination of the 12 scenes does

indicate that QF1 data points are consistently closer to clouds than the QF0 data points.
       Figure 7b presents the SHDOM 1D / 3D ratios for this oceanic scene. Figure 7 and Fig. 6 are similar in that the smallest 1D /3D ratios are located close to clouds. The V10 Lite file data points have 1D / 3D ratios in the 0.81 – 0.99 range, with an average of 0.98, which is larger than the Fig. 6b Amazon scene average V10 1D / 3D ratio of 0.91. The spatial extent of the lowest 1D / 3D

ratios near cloud edges in Fig. 7 is less than the spatial extent of the lowest 1D /3D ratios in Fig. 6b, which motivated the difference in the Fig. 6b and Fig. 7b color bar scales.

## 5 Analysis of multiple scenes

Lite file variables and SHDOM 1D and 3D radiance fields are analyzed for 36 individual scenes (12 ocean glint, 12 land nadir, and 12 land glint geometry). The specifics for the scenes are given in Table 3. A range of solar zenith angle from 20° to 55° characterizes the ocean glint scenes. The land scenes are situated over the Amazon. It is of interest to study Amazon scenes since there are relatively few successful QF0 data points over the Amazon, and the Amazon is of large importance

to the global carbon cycle. The majority of completed retrievals over the Amazon are QF1

retrievals. Several Amazon scenes were chosen purposely to make sure that there were at least some QF0 data points in the scenes.

Figure 8 presents results for individual sun glint retrievals over the Pacific. QF0 and QF1 data points are given by the green (*) and blue (+) symbols. In Fig. 8a SHDOM 1D / 3D ratios for continuum $O_2$ A-band (for the smallest gas optical depth) versus the nearest cloud distance taken from the 3D metric file are graphed. The 1D / 3D ratios are near unity for cloud distances greater than 4 km. The ratios become smaller, with smallest values near 0.3, as the nearest cloud distance approaches zero. The largest number of QF0 data points are for large cloud distances, while the largest number of QF1 data points are for small cloud distances.

In the operational OCO-2 bias correction processing step, a specified limited range for 31 variables determines if a retrieval is a QF0 data point (see Table 3.4 of the Orbiting Carbon Observatory–2 & 3 (OCO-2 & OCO-3) Data Product User's Guide, 2020). As an example, a retrieval is a QF0 data point if dP is between -7.5 and 8 hPa for land observations for the V10 data files.

In Fig. 8b the dP values for QF1 data points take on large negative values as cloud distance approaches zero. The interpretation of Fig. 8a and Fig. 8b is that the 1D / 3D ratio becomes small as the 3D cloud effect enhances (brightens) the radiances. In order for the retrieval to match the forward model with the observed radiance, the retrieval decreases the surface pressure to smaller values (compared to the meteorological surface pressure field) at small nearest cloud distance.

To place the various scene XCO2 values onto a common framework, we calculate the average QF0 XCO2 for a 6° range of latitude centered on the scene's latitude. The average is then subtracted from the QF0 and QF1 rawXCO2 for a specific scene, and these adjusted rawXCO2 values are placed into our graphs. In Fig. 8c the adjusted rawXCO2 varies from – 2 to 2 ppm for the QF0 data at all nearest cloud distances. For the QF1 adjusted rawXCO2 data the values take on increasingly negative values as nearest cloud distance decreases. *The concurrent decrease in the 1D /3D ratios and rawXCO2, as nearest cloud distance decreases, is evidence of 3D cloud effects in the OCO-2 retrievals, especially for the QF=1 observations.*

Figure 8d displays total aerosol optical depth, which takes on increasingly larger values as nearest cloud distance decreases for the QF1 data. The percentages of QF0 and QF1 data points at cloud distances less than 4 km are 51% and 86% of the total number of QF0 and QF1 data points, respectively. Since each scene has a different inherent aerosol optical depth, it is expected that there will be several green (*) sets of data points (the rightward directed spikes of total aerosol optical depth in Fig. 8d).

In the V10 retrieval surface reflectance is represented by the sum of Cox-Munk surface glint and Lambertian surface terms. Figure 8e displays the retrieved $O_2$ A-band Lambertian surface reflectance (Albedo1) values *added* to the Cox-Munk term. (The V10 Lite files do not specify the total surface reflectance values over the ocean, while the files do specify the total surface reflectance over land). Fig. 8f displays "delta Wind" values (the difference in retrieved wind speed and a priori wind speed, in m/sec units). Since an increase in wind speed generally leads to a smaller surface reflectance, the positive delta Wind (Fig. 8f) and Albedo1 values for QF=1 data indicate that the retrieval selects decreasing Cox-Munk and increasing Lambertian contributions to the total surface reflectance as nearest cloud distance varies from large to small values.

Figure 9 displays graphs for the Amazon land nadir scenes. The behavior of the data points is similar to the ocean glint data, though the range of nearest cloud distance is smaller than for the ocean glint scenes. Since the selected ocean and Amazon scenes were picked in a purposeful manner, this range difference is not generally true (see the next section for yearly analyses). The

1D / 3D ratios in Fig. 9a, however, do decrease as the cloud distance decreases. As cloud distance decreases the adjusted XCO2 values take on increasingly negative values Fig. 9c. The range in QF1 XCO2 for cloud distances greater than 5 km is larger for the Amazon land nadir scenes. Standard deviations for ocean glint XCO2 are generally smaller than land XCO2 (see Fig. 18 of O'Dell et al. 2018 and Table 4 of Massie et al. 2021). The percentages of QF0 and QF1 data points at cloud distances less than 4 km are 74% and 84% of the total number of QF0 and QF1 data points, respectively.

From Fig. 5 and Table 1 a change in 0.02 and 0.01 in $O_2$ A-band and WCO2 surface reflectance yields an increase in radiance which is *three and five times* as large, respectively, as the 3D radiance perturbation, so small changes in the QF0 and QF1 surface reflectance values (referred to as "albedo" values in the Lite files and in Figs. 9 and 10) add a sufficient increase in radiance that brings forward model and observed radiances into agreement. There is a noticeable difference in Figs. 8 and 9 in that the ocean glint scenes have dP less than -10 hPa for the QF1 data, while there are few dP values less than -10 hPa in Fig. 11 for the land nadir scenes.

Figure 10 displays Amazon land glint data. The 3D metric files have nearest cloud distance data for four of the 12 land glint scenes (the LG 150625 scenes, as specified in Table 3). Nearest cloud distance values for the LG 200603 and LG 200610 scenes of Table 3 were calculated based on Lite file longitude and latitudes and 250 m MODIS radiance fields. SHDOM 1D / 3D ratios and rawXCO2 decrease as cloud distance decreases, while total aerosol optical depth increases, especially for the QF1 data. In contrast to Fig. 9, the retrieval selects $O_2$ A-band and WCO2 band surface reflectivity at small cloud distances for some (~ dozen) of the data points, which are smaller than the surface reflectivity at large cloud distance.

3D cloud effects in ocean glint, Amazon nadir, and Amazon glint observations are evident in Figs. 8, 9, and 10 since 1D /3D ratios and raw XCO2 concurrently decrease as nearest cloud distance decreases, especially for the QF=1 data points. Using Fig. 5 as a rough guide, an increase in aerosol, an increase in surface reflectance, and decreases in surface pressure and XCO2, yield positive radiance perturbations. There are differences over ocean and land retrievals in that dP variations are smaller over land (compared to the large decrease in dP for the ocean glint retrievals, in Fig. 8b), and that there are some retrievals for land glint observations in which decreases in surface reflectance are present as nearest cloud distance decreases. There is commonality in Figs. 8, 9, and 10 in that the y-axis spread in dP, XCO2, aerosol, and "delta Wind" is largest as nearest cloud distance decreases.

The fact that cloud brightening dominates in Fig. 8, compared to cloud shadowing, has important implications for the Fig. 10 OCO-2 retrievals. From Fig. 3a cloud shadows are associated with SHDOM 1D / 3D ratios greater than unity. Figure 8a has observed Lite file retrievals over the ocean with SHDOM 1D / 3D ratios decreasing to low values as nearest cloud distance decreases, with little evidence of ratios greater than unity. The 1D / 3D intensity ratio averages are 0.98 and 0.96 for the QF=0 and QF=1 data points. The radiance brightening is associated with retrieval dP (retrieved – meteorological field) values less than zero, with a sharp decrease in dP for small cloud distances. Figs. 10a, 10b, and 10c are consistent with the presence of 3D cloud effects in OCO-2 data, since SHDOM 1D /3D ratios, retrieval dP, and raw XCO2 decreases as nearest cloud distance decreases. The cloud brightening is accounted for by the retrieval by a combination of increases in aerosol optical depth (Fig. 10d), and decreases in dP and XCO2, as nearest cloud distance decreases. This is most apparent in the QF=1 data points in Fig. 8. This is also illustrated by Figure 11 of Massie et al (2021), for a larger set of data points.

## 6 Yearly analysis

Generalization in differences in ocean glint, land nadir, and land glint are best made from an analysis of yearly averaged data using the daily Lite file and 3D metric files for each full year of OCO-2 data during 2015-2018. Since the 3D metric files do not include 1D / 3D SHDOM ratios, which are only available for the 36 scenes discussed above, the nearest cloud distance is the primary 3D metric utilized in this section.

The nearest cloud distance averages for 2016 are presented in Fig. 11 for ocean glint, land nadir and land glint observations. *It is apparent that clouds are closer on average for OCO-2 observations over the ocean than over land*. This is especially true for the QF1 data which is approximately 4 km on average from clouds over the ocean. The 4 km mark is important since the curves in Figs. 2 - 4 become very non-linear at distances near and below 4 km cloud distance. The
cloud distance is noticeably smaller over the equator over both land and ocean, with a symmetrical appearance with respect to the northern and southern hemispheres.

Averages of dP for several 10° latitude bins in the tropics are displayed in Fig. 12, with -10 referring to the 10° S to 0° S latitude bin, etc. The QF1 ocean glint curves are best defined (Fig. 12a), with large negative dP values at cloud distances less than 5 km, and dP between -1 and 1 hPa
for cloud distances greater than 5 km. This signature is evidence of 3D cloud effects that impact OCO-2 radiances. The data points in Fig. 8b are consistent with the Fig. 12a dP tropical averages. The land nadir and land glint dP curves (Fig. 12b and Fig12c) in the tropics, however, increase in dP by several hPa as cloud distance decreases, with dP values in the positive 0 to 4 hPa range.

Averages (denoted as Ave) of rawXCO2 for the same tropical latitude bins in Fig. 12 are
displayed in Fig. 13 for 2016. The QF1 XCO2 curves are again best defined for the ocean glint curves, with fewer oscillations in the curves than for those over land. Raw XCO2 decreases by 0.3 to 0.7 ppm as the cloud distance decreases in the tropics for the QF0 data, and decreases by 1.2 to 2.3 ppm for the QF1 data. *Since the 3D cloud-retrieval bias is given by the difference Ave(20 km) – Ave (0 km), the 3D cloud-retrieval biases for QF0 and QF1 data are between 0.3 and 0.7 ppm,*
*and between 1.2 to 2.3 ppm, respectively, in the tropics.* (The *3D cloud-retrieval bias* is the raw XCO2 bias introduced by the retrieval, and does not refer to the retrieval of cloud properties). The Fig. 13 average curves indicate that the operational retrieval *underestimates* rawXCO2 in the tropics. The rawXCO2 land nadir and land glint averages for QF0 data decreases by 0.3 to 0.6 ppm as cloud distance decreases, and by 1.5 to 3.0 ppm for the QF1 data.

A visual examination of the graphs (not shown) of the average dP and raw XCO2 curves for 2015, 2017, and 2018 display many similar qualitative features to those displayed by the 2016 Figs. 12 and 13 curves. While XCO2 has increased during 2015 – 2018 (by approximately 2.4 ppm per year), the shapes of the curves are qualitatively similar year to year.

Figure 14 displays ΔdP averages (i.e. dP (near 20 km) – dP (near 0 km) differences) as a
function of latitude for 2015 – 2018, with -40 referring to the 40°S to 30°S latitude bin, etc. Instead of just using the dP values at 20 and 0 km cloud distance, averages for 18-21 km and 0-3 km are calculated to bring more data into the averaging process. The ocean glint QF0 ΔdP averages are slightly positive at all latitudes. The ocean glint QF1 ΔdP averages are consistently positive at all latitudes, with values between 2 and 4 hPa for latitudes southward of 30° N.  There is a
hemispherical asymmetry in the ΔdP land values, with negative ΔdP QF1 values near -4 hPa at 20° S and values near zero in the northern hemisphere. The reason for this asymmetry is not known.

Figure 15 displays ΔrawXCO2 averages (i.e. raw XCO2 (near 20 km) – raw XCO2 (near 0 km) differences) as a function of latitude for 2015 – 2018. The ΔrawXCO2 averages are an

appropriate measure of 3D cloud-retrieval biases (in ppm units) that are present in raw XCO2 OCO-2 data files. Table 4 specifies the range of these biases for the six lines in Fig. 15. The ocean glint QF1 ΔrawXCO2 biases are consistently above 1 ppm, with a latitudinal average near 1.5 ppm. *This indicates that ocean glint QF1 data is underestimated by an amount roughly equal to the 1 ppm OCO-2 measurement goal*. The ocean glint QF0 data has an average ΔrawXCO2 near 0.4 ppm. Land glint and land nadir QF1 average ΔrawXCO2 is positive in the northern and southern hemispheres, while the QF0 average ΔrawXCO2 is positive (0.25 ppm) in the southern hemisphere and negative (-0.25 ppm) in the northern hemisphere. This asymmetry is likely related to the ΔdP hemispherical asymmetry present in the Fig. 12 curves.

3D cloud effects are expected to asymptote to zero as the cloud distance becomes very large. The ocean glint averages are the clearest evidence of 3D cloud effects (Figs. 12a, 12b, 13a, 13b) in the tropics. With respect to latitudinal averages, the ocean glint QF0 (QF1) latitude average ΔrawXCO2 biases (Fig. 15) are near 0.4 (1.5) ppm, while land nadir and land glint biases are near 0.25 and -0.25 ppm for QF0 data in the southern and northern hemispheres.

## 7 3D bcXCO2 cloud effect bias mitigation

This section discusses how one can calculate an empirical look-up table of 3D cloud effect biases, and use it to correct bcXCO2 for 3D cloud effects that are present in the V10 Lite files. The V10 3D cloud effect biases for bcXCO2 data will differ from the V11 Lite file bcXCO2 biases. V11 data currently is in the production phase.

Figure 16 displays a graph of QF0 ocean glint bcXCO2 as a function of nearest cloud distance D. Using the 20° S - 0° S band as an example, and a nearest cloud distance of 3km, the 3D cloud effect bias is given by bias B(D) value, calculated by drawing a line on the y axis that specifies the asymptotic bcXCO2 value at large nearest cloud distance, and the difference bcXCO2(20 km) – bcXCO2(3 km). The B(D) values can be calculated for all nearest cloud distances and all latitude bands. Figure 17 presents curves of these biases for the six latitude bands for QF0 and QF1 data. The Zenodo archive (https://doi.org/10.5281/zenodo.7655136) has separate ascii files of the QF0 and QF1 curves, and ascii files of the 3D cloud effect biases for QF0 and QF1 data for land nadir and land glint data. The files specify a tabulation of nearest cloud distance versus 3D cloud effect bias. Positive (negative) biases indicate that the operational retrieval and post-retrieval bias processing underestimates (overestimates) bcXCO2.

There are many data points in the averages displayed in Fig. 16. For the 20° S - 0° S band example, the number of data points N(D) in the nearest cloud distance bins of 1 and 20 km are 2.38 $10^6$ and 181,909. For a bcXCO2 error of 0.39 ppm (calculated from a 50° S to 50° N average of Lite file XCO2err standard errors for the Fig. 7 date), the 2 sigma 95% confidence limits of the determination of the bcXCO2 averages for these bins are 0.0006 and 0.0017 ppm assuming uncorrelated errors. The 95% confidence limits of the bcXCO2 averages for the 40° N - 60° N band are also small (0.0017 and 0.0036 ppm). The increase in the bcXCO2 averages in the 40° N - 60° N band is inherently present in the Lite file bcXCO2 and is not due to too few data points in the nearest cloud distance bins.

Figure 17 indicates that the 3D cloud effect biases are largest (and consistently positive) for the ocean glint QF1 data. QF1 3D cloud effect biases are generally larger than the QF0 biases, and generally positive, for ocean glint, land nadir, and land glint observations. For QF0 data, the biases are both positive and negative. As an example of negative biases, Fig. 17 biases are negative for the 40° N - 60° N latitude band, since bcXCO2 increases as nearest cloud distance decreases.

To correct a Lite file bcXCO2 value of 3D cloud effect bias, the 3D metric data (available from https://doi.org/10.5281/zenodo.4008765) can be used in conjunction with the latitude-dependent 3D cloud effect bias data B(D) (available from https://doi.org/10.5281/zenodo.7655136). Given a nearest cloud distance D (from the 3D metric file data, or calculated from MODIS or geosynchronous satellite radiance data), the corrected bcXCO2 value is then bcXCO2 + B(D).

Latitudinal averages of 3D cloud effect biases, calculated by numerically weighting the biases B(D) by the number of observations N(D) in each of the nearest cloud distance bins D for each of the latitude bands (the ratio $\sum$B(D) N(D) / $\sum$N(D)), are presented in Fig. 18 for ocean glint QF0 and QF1 data. Since there are more observations at small nearest cloud distances, compared to large nearest cloud distances, the weighted biases are dominated by the left side of the curves in

Fig. 16. Biases are larger over the southern hemisphere than over the northern hemisphere. This may be due to the fact that there are more TCCON observations in the northern hemisphere.

     Table 5 presents the weighted biases, and the biases for the 0 to 1 km nearest cloud distance bin. For Fig. 16, the biases for the 0 t o1 km bin are the largest possible biases. The calculations that generated Table 5 were carried out with and without temporal corrections. Since XCO2 is

705 increasing at 2.5 ppm/yr (and at larger values in the high northern latitudes), and since we analyze bcXCO2 data for years between 2015-2018, the average year temporal value for e.g. nearest cloud distances of 3 and 20 km, may differ. Knowing the temporal values for each latitude band and each nearest cloud distance bin, and the temporal bcCO2 trend (in ppm/yr, calculated from a linear fit to bcXCO2 for the four years), the bcXCO2 average for a specific latitude band and nearest cloud

distance bin can be temporally corrected to a common latitude-band specific temporal value (see Appendix B for details). The C columns of Table 5 include the temporal corrected biases, and the nC columns are biases with no temporal corrections. Figs. 16-18 display temporal corrected bcXCO2 calculations. A graph of bcXCO2 (not shown), comprised of bcXCO2 not temporally corrected, is very similar to Fig. 16.

Inspection of Table 5 indicates that averages of weighted 3D cloud effect temporally corrected biases in the northern and southern hemispheres, are 0.16 (1.31) and 0.26 (1.41) ppm, respectively, over the ocean, and - 0.13 (0.51) and -0.08 (0.47) ppm over land, for QF0 (QF1) data. All of the QF1 biases in Table 5 are positive, while the QF0 biases are positive over the ocean and mostly negative over land.

## 8 Conclusions

While 1D radiative transfer theory is extensively covered in papers and textbooks, 3D radiative transfer "rule of thumb" knowledge is not well established. The calculation of idealized bar clouds

(Fig. 1) provides insight as to how 1D / 3D ratios vary as a function of 3D cloud metrics. Of the various possible cloud metrics to consider, nearest cloud distance readily comes to mind since the 3D cloud effect obviously becomes small if clouds are far away from observation points. Figure 2 indicates that the 1D / 3D ratio is closest to unity far away from clouds, and decreases towards smaller values as cloud distance D decreases, and that the $O_2$ A-band will likely have the largest

3D cloud effect, followed by the WCO2 and SCO2 bands. Figure 2 also indicates that the 1D / 3D ratios are appreciable in size. With 80 % of the Amazon retrievals within 4 km of clouds (Table 2), $O_2$ A-band 1D / 3D ratios are less than 0.96.

     The Fig. 4 curves from idealized bar cloud SHDOM calculations are presented to convey insights into the factors which *modulate* 3D cloud effects. Fig. 4 indicates the sensitivity of 1D /

3D ratios as a function of vertical gas optical depth, solar zenith angle, cloud top height (cloud

vertical extent), surface reflectance, cloud LWC, and aerosol composition. The sensitivity is largest for the gas vertical optical depth. While there are important variations in the details shown in Fig. 4, all panels display a *highly non-linear* decrease in 1D / 3D ratios as nearest cloud distance decreases.

The OCO-2 cloud preprocessor does a very good job in screening observations near and over clouds. The preprocessor, however, does not necessarily identify clear sky observations, impacted by 3D cloud effects, that are located *between* low-altitude clouds. Figs. 6 and 7 illustrate typical scenes over land and ocean in which successful OCO-2 retrievals are located between clouds. Figures 8, 9, and 10 illustrate that 3D cloud effects usually occur within 4 km of clouds, since 1D

745     / 3D ratios and raw XCO2 concurrently decrease at nearest cloud distances less than 4 km.

        Cloud brightening events are prevalent compared to cloud shadowing events. Yearly analysis of Amazon and Pacific regions (Table 2) yields retrieval percentages associated with cloud shadowing to be less than 4% for cloud heights less than or equal to 8 km. The unequal percentage of cloud brightening (96%) versus the cloud shadowing 4% percentage in the retrieved

observations imposes an asymmetry in the imposed 1D / 3D intensity ratio radiance perturbations, with fewer cloud shadowing events, compared to cloud brightening events.

        3D cloud effects likely are more important for ocean glint observations than for land nadir observations since nearest cloud distances are smaller over the ocean than land (see Fig. 11). We assert that the predominant presence of QF1 data over the Amazon, which is subject to many low

altitude clouds, is in part due to the difficulties imposed by 3D radiative transfer upon the 1D interpretation of measured Amazon OCO-2 radiances.

        Figure 16 illustrates how one can calculate a 3D cloud effect bias look-up table for bcXCO2 data, with Fig. 17 presenting ocean glint table look-up data. Application of the look-up table data (which can be accessed from https://doi.org/10.5281/zenodo.7655136), is illustrated in Fig. 18,

which indicates that the operational processing underestimates QF0 bcXCO2 over the southern ocean on average by 0.3 ppm, relative to observations free of 3D cloud effects (as estimated from observations near 20 km nearest cloud distance). Though the QF0 hemispherical biases are between -0.31 and 0.26 ppm (Table 5), biases in bcXCO2 on regional scales as small as a few tenths of a part-per-million (ppm) in XCO2 can lead to spurious values of inferred $CO_2$ fluxes

(Chevallier et al., 2010). The operational retrieval and post-retrieval processing consistently underestimates QF1 bcXCO2 by 0.3 to 1.4 ppm relative to observations free of 3D cloud effects (Table 5).

        While Massie et al. (2021) focused on comparisons of bcXCO2 and TCCON, the analysis of 275 times more bcXCO2 data between 2015-2018 (without reference to TCCON data) *enabled*

*calculations of 3D cloud effect biases as a function of latitude. The biases are larger in the southern hemisphere.* This is possibly due to the fact that there are fewer TCCON observations in the southern hemisphere. The magnitude of the 3D cloud effect biases discussed in Massie et al. (2021) and this paper are similar in size, with QF1 biases generally larger than QF0 biases. Since the post-retrieval bias correction process exclusively uses QF0 data, and dP and dPfrac variables,

which are correlated with nearest cloud distance (Massie et al. 2021), it is expected that the QF0 biases will be small. The post-retrieval bias correction process *indirectly* accounts for 3D effects, but Fig. 16 and the Table 5 entries indicate that 3D cloud effect biases remain in the Lite file data.

        Future work includes the development of a quick parameterization of 1D / 3D ratios as a function of aerosol and cloud optical depth, given an arbitrary geospatial distribution of clouds.

This work will examine a wider range of parameters such as cloud height, aerosol height, aerosol composition, in addition to an examination of scenes not covered in this paper, such as brighter

surfaces. Aerosol characteristics are of interest since Bell et al. (2023) used OCO-3 data to demonstrate that "swath bias" is related to aerosol characteristics and viewing geometry. While our paper focuses on OCO-2, the results are applicable to OCO-3 on the International Space Station. Geosynchronous satellite radiance and cloud data can be used to derive nearest cloud distance for studies related to OCO-3 and 3D cloud effects.

**Appendix A**

The following paragraphs discuss the specifics of the algorithm used to calculate the percentage of cloud brightening and cloud shadowing events.

Using the Amazon observations on 22 June 2015 (150622) as an example, the L2ABP preprocessor data file specifies the cloud flag (0=clear, 1=cloudy, 2=undetermined) and $O_2$ A-band continuum radiances for each OCO-2 measurement are specified by the OCO-2 Level 1B data files. Average clear and cloudy continuum radiances ($Clear_{bin}$ and $Cloudy_{bin}$ are determined in 0.5 latitude bin steps for the 10° S to 3° N latitude range. Some of the latitude bins will be fully cloudy. These bins are not used in the calculation of a $Clear_{ave}$ radiance average. Some of the latitude bins have too few clouds, and are excluded since it is of interest to determine clear radiances in the vicinity of clouds. A $Clear_{ave}$ radiance average is calculated from the $Clear_{bin}$ averages when the percentage of clear flags for a latitude bin is greater than 50%. A similar calculation is done for the $Cloudy_{ave}$ average. Guided by SHDOM calculations, for bar clouds similar to Fig. 1, with cloud tops at 2, 4, 6, and 8 km, the 3D radiances are analyzed to determine an average observation to cloud radiance ratio $Ratio_{obs,cloud}$, with $Ratio_{obs,cloud}$ determined from 3D radiance (distance D from the cloud) / SHDOM 3D radiance (position located over the cloud) ratios. These calculations yield a SHDOM threshold $Ratio_{threshold}$ near 0.30. Since there is a range of $Cloudy_{bin}$ values for the various latitude bins on an individual day, $Ratio_{threshold}$ is conservatively increased arbitrarily to 0.60 (which will overestimate the cloud shadow "Percentage Geom" values in Table 2).

For a specific observation point, L2ABP locations surrounding the observation point are examined. A 10 km by 10 km box surrounds the observation point, and the algorithm loops over L2ABP data file longitude I, latitude J indices. $Ratio_{obs;\ I,J\ point}$ values, equal to the $O_2$ A-band L2ABP $radiance_{obs\ point}$ divided by the $O_2$ A-band L2ABP $radiance_{I,\ J\ point}$, are calculated. The distances $Distance_{obs;\ I,J\ point}$ from the observation point to the I, J positions are calculated. The angles $Angles_{obs;\ I,\ J\ point}$ between the fixed solar vector (observation point to Sun location) and geometry vectors (from the observation point to the L2ABP I, J positions are also calculated. For a given cloud height (e.g. 4 km), the solar zenith angle and cloud height determines the $X_{shadow}$ spatial length (in km) that the shadow corresponds to.

In general, if a) the positions of the sun, L2ABP file I, J position, and observation point are in a line with the I, J point in the middle of the line, if b) $Distance_{obs,\ I,\ J\ point}$ is less than $X_{shadow}$, if c) $Ratio_{obs;\ I,\ J\ point}$ is less than $Ratio_{threshold}$, if d) the I, J point has a L2ABP cloud flag equal to 1 (cloudy case), and if e) $Angles_{obs;\ I,J\ point}$ is less than a threshold angle difference (e.g. 30°), then the observation point is associated with a shadow. The code-wise loops check to see if a shadow is associated with each observation point, and then calculates the percentage of retrievals that are associated with shadows for each cloud height.

**Appendix B**

Temporal corrections of bcXCO2 for years 2015-2018 are incorporated into Table 5 and Figs. 16 – 18.

For each latitude band, the bcXCO2 trend $T_{lat}$ (ppm/yr) is calculated utilizing QF0 bcXCCO2 data from all nearest cloud distance D (km) values. The trends are on the order of 2.5 ppm /yr, though higher (3.0 ppm /yr) at higher northern latitudes. The trends $T_{lat}$ from the QF0 bcXCO2
calculation are applied to both QF0 and QF1 bcXCO2.

For a specific latitude band, the observation number weighted temporal averages $Y_{lat,D}$ are calculated for each nearest cloud distance D bin, with $Y_{lat,D}$ calculated using day of year information for each individual data point. The $Y_{lat,D}$ are used to calculate an average time $Y_{lat\ ave}$ for each latitude band.

To produce Figs. 16-18, each bcXCO2 average (before temporal correction) for each D bin was modified by adding the correction term $- T_{lat} (Y_{lat,D} - Y_{lat\ ave})$ for each bin D. The $-$ sign is applied since $T_{lat}$ is positive, with positive and negative signed temporal corrections for data in 2015 and 2018.

*Data availability*. The data used to generate the Fig. 4 curves, and the 3D cloud effect bias look-up tables can be downloaded from the Zenodo archive (https://doi.org/10.5281/zenodo.7655136), and the 3D metric files are available at the archive https://doi.org/10.5281/zenodo.4008765

*Author Contribution*. Steven Massie performed the calculations presented in this paper, and was the primary author of the text. Heather Cronk supplied the MODIS-based nearest cloud distance
data. Aronne Merrelli generated the 3D metric files. Steffen Mauceri and Sebastian Schmidt provided suggestions on the content of the paper.

*Competing interests*. At least one of the (co-)authors is a member of the editorial board of Atmospheric Measurement Techniques.

*Acknowledgements*. STM and KSS acknowledges support by NASA Grant 80NSSC21K1063
"Mitigation of 3D cloud radiative effects in OCO-2 and OCO-3 XCO2 retrievals". AM acknowledges support by JPL subcontract 1678415 and NASA Grant 80NSSC22K1287. Appreciation is expressed to Frank Evans for helpful advice in regard to the SHDOM program. Appreciation is expressed to the OCO-2 computer staff at the Jet Propulsion Laboratory, and to Garth D'Attillo and Timothy Fredrick of the Atmospheric Chemistry Observations and Modeling
(ACOM) division at the National Center for Atmospheric Research (NCAR), supported by the National Science Foundation, for maintaining the operational capabilities of computer systems during the ongoing global COVID-19 pandemic.

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

Table 1. Baseline and perturbations of the Fig. 5 calculations.

| Case | Baseline | Perturbation |
|---|---|---|
| Surface pressure (hPa) | 988.719 | 2 hPa |
| O2 surface reflectance | 0.2536 | 0.025 |
| WCO2 surface reflectance | 0.140 | 0.014 |
| SCO2 surface reflectance | 0.043 | 0.0043 |
| O2 aerosol optical depth | 0.067 | 0.0067 |
| WCO2 aerosol optical depth | 0.039 | 0.0039 |
| SCO2 aerosol optical depth | 0.029 | 0.0029 |
| XCO2 | 402.29 | 1 ppm |

Table 2. Statistics of various quantities (in %) for single day and 2016 yearly averages based upon an analysis of L2ABP preprocessor data files.

| | Amazon Nadir 150622 | Pacific Glint 160612 | Amazon Nadir 2016 | Amazon Glint 2016 | Pacific Glint 2016 |
|---|---|---|---|---|---|
| Latitude range | 10 S – 3 N | 8 N – 22 N | 10 S – 3N | 10 S – 3N | 10 N – 20N |
| Longitude range | 64 – 45 W | 156 – 176 E | 65 – 45 W | 65 – 45 W | 160 – 170E |
| Total Number Obs[a] | 5162 | 6004 | 602939 | 636240 | 547808 |
| Number of Retrievals[b] | 589 | 4320 | 58477 | 77366 | 276944 |
| Retrieved Percentage | 11 | 71 | 9.6 | 12 | 50 |
| Percentage QF0 | 40 | 55 | 43 | 34 | 60 |
| Percentage QF1 | 60 | 44 | 56 | 65 | 39 |
| Percentage Cloud< 4km[c] | 80 | 50 | 56 | 53 | 56 |
| Percentage Geom, 2km[d] | 0 | 0 | 0 | 0.1 | 1.2 |
| Percentage Geom, 4 km | 0 | 0 | 0.5 | 0.9 | 2.1 |
| Percentage Geom, 6 km | 1.1 | 1.1 | 1.0 | 2.6 | 2.9 |
| Perecentage Geom, 8 km | 1.3 | 1.4 | 1.2 | 3.7 | 3.4 |

a Total Number Obs is the total number of observations in the latitude – longitude range as specified by the L2ABP file.

b Number of retrievals is the number of successful retrievals as specified by the Lite file.

c Percentage Cloud < 4km is the percentage of retrievals for which the cloud distance is less than 4km.

d Percentage Geom, 2km is the percentage of retrievals associated with cloud shadows for the cloud height in 0 to 2km altitude range.

Table 3. Specifics of the multiple scenes.

| Scene | Latitude | Longitude | Solar zenith angle | Sensor zenith angle |
|---|---|---|---|---|
| OG[a] 160612[b] | 15.50 | 166.65 | 20 | 16 |
| OG 160612 | 10.56 | 167.78 | 24 | 19 |
| OG 160612 | 10.22 | -128.78 | 24 | 19 |
| OG 160612 | 5.26 | -127.69 | 28 | 22 |
| OG 160612 | -10.13 | -124.53 | 41 | 32 |
| OG 160612 | -14.28 | -123.64 | 44 | 35 |
| OG 160612 | -17.55 | -161.62 | 47 | 37 |
| OG 160612 | -26.15 | -121.23 | 55 | 43 |
| OG 160611 | 7.87 | -142.19 | 26 | 20 |
| OG 160611 | 5.77 | -141.80 | 27 | 22 |
| OG 160611 | 10.71 | -142.85 | 24 | 19 |
| OG 160611 | 11.48 | -143.00 | 23 | 18 |
| LN 150622 | 3.82 | -58.18 | 30 | 0 |
| LN 150622 | 0.45 | -57.44 | 32 | 0 |
| LN 150622 | -5.24 | -56.23 | 37 | 0 |
| LN 150622 | -5.87 | -56.07 | 38 | 0 |
| LN 150622 | -6.52 | -55.98 | 38 | 0 |
| LN 150622 | -8.58 | -55.15 | 40 | 0 |
| LN 160308 | -11.78 | -48.57 | 24 | 0 |
| LN 160308 | -18.11 | -47.15 | 27 | 0 |
| LN 160308 | -21.08 | -46.47 | 29 | 0 |
| LN 160107 | -23.29 | -53.65 | 24 | 0 |
| LN 160107 | -16.96 | -55.16 | 24 | 0 |
| LN 160107 | -25.44 | -53.18 | 24 | 0 |
| LG 150625 | -2.31 | -65.89 | 34 | 27 |
| LG 150625 | -4.03 | -65.51 | 35 | 28 |
| LG 150625 | -1.20 | -66.13 | 33 | 26 |
| LG 150625 | 3.53 | -67.08 | 29 | 23 |
| LG 200603 | -3.57 | -57.89 | 35 | 27 |
| LG 200603 | 1.99 | -59.09 | 30 | 24 |
| LG 200603 | 0.35 | -58.77 | 31 | 24 |
| LG 200603 | 2.89 | -59.24 | 29 | 23 |

| | | | | |
|---|---|---|---|---|
| LG 200610 | -10.59 | -58.04 | 41 | 32 |
| LG 200610 | -9.44 | -58.21 | 40 | 31 |
| LG 200610 | -3.69 | -59.39 | 35 | 28 |
| LG 200610 | -1.45 | -59.87 | 33 | 26 |

a OG, LN, and LG refer to ocean glint, land nadir, and land glint observing modes

    b 160612 refers to 12 June 2016

Table 4. Minimum, average, and maximum ranges (in ppm) of the six curves in Fig. 17.

| Curve | Minimum | Average | Maximum |
|---|---|---|---|
| For 50° S to 50° N | | | |
| Ocean glint QF0 | -0.11 | 0.42 | 0.62 |
| Land glint QF0 | -1.34 | -0.02 | 0.44 |
| Land nadir QF0 | -1.01 | -0.05 | 0.58 |
| Ocean glint QF1 | 1.04 | 1.59 | 2.33 |
| Land glint QF1 | 0.42 | 1.66 | 2.67 |
| Land  nadir QF1 | -0.05 | 1.28 | 2.66 |
| | | | |
| For 50° S to 0° N | | | |
| Ocean glint QF0 | 0.43 | 0.52 | 0.56 |
| Land glint QF0 | 0.02 | 0.25 | 0.44 |
| Land nadir QF0 | -0.21 | 0.23 | 0.57 |
| Ocean glint QF1 | 1.04 | 1.42 | 2.33 |
| Land glint QF1 | 1.80 | 2.18 | 2.67 |
| Land nadir QF1 | 1.34 | 1.98 | 2.66 |
| | | | |
| For 0° N to 50° N | | | |
| Ocean glint QF0 | -0.11 | 0.34 | 0.62 |
| Land glint QF0 | -1.34 | -0.25 | 0.44 |
| Land nadir QF0 | -1.01 | -0.29 | 0.49 |
| Ocean glint QF1 | 1.56 | 1.72 | 1.96 |
| Land glint QF1 | 0.42 | 1.22 | 2.64 |
| Land nadir QF1 | -0.05 | 0.70 | 2.23 |

Table 5. bcXCO2 3D cloud biases (ppm) for northern (0°- 60° N, NH) and southern (60° S - 0° S, SH) hemispheres, and for the 60° S and 60° N range (NH+SH). A positive (negative) bias indicates that the operation retrieval and post-retrieval bias processing underestimates (overestimates) bcXCO2 due to 3D cloud radiative effects. WC and WnC refer to observation number weighted biases with and without temporal corrections. DiffC and DiffnC refer to (bcXCO2(20 km) – bcXCO2(0 km) bias differences, for nearest cloud distances of 20 and 0 km, with and without temporal corrections. 0 km refers to the 0 to 1 km cloud distance bin. See Sect. 7 for discussion of this Table.

| Case | WC QF0 | WnC QF0 | WC QF1 | WnC QF1 | DiffC QF0 | DiffnC QF0 | DiffC QF1 | DiffnC QF1 |
|---|---|---|---|---|---|---|---|---|
| Ocean Glint | | | | | | | | |
| NH | 0.16 | 0067 | 1.31 | 1.05 | 0.39 | 0.24 | 2.07 | 1.77 |
| SH | 0.26 | 0.20 | 1.41 | 1.36 | 0.53 | 0.45 | 2.02 | 1.98 |
| NH+SH | 0.21 | 0.14 | 1.36 | 1.21 | 0.46 | 0.34 | 2.05 | 1.87 |
| Land Nadir | | | | | | | | |
| NH | -0.084 | -0.099 | 0.45 | 0.50 | -0.60 | -0.61 | 1.46 | 1.55 |
| SH | -0.31 | -0.21 | 0.67 | 0.41 | -0.36 | -0.21 | 2.04 | 1.77 |
| NH+SH | -0.20 | -0.15 | 0.56 | 0.46 | -0.48 | -0.41 | 1.74 | 1.66 |
| Land Glint | | | | | | | | |
| NH | -0.17 | -0.095 | 0.57 | 0.70 | -0.70 | -0.57 | 2.00 | 2.22 |
| SH | 0.18 | -0.21 | 0.29 | 0.85 | 0.27 | -0.34 | 1.54 | 2.23 |
| NH+SH | 0.007 | -0.15 | 0.43 | 0.78 | -0.21 | -0.45 | 1.77 | 2.23 |
| Land Nadir + Glint | | | | | | | | |
| NH | -0.13 | -0.098 | 0.51 | 0.61 | -0.65 | -0.59 | 1.73 | 1.88 |
| SH | -0.085 | -0.21 | 0.47 | 0.64 | -0.04 | -0.28 | 1.79 | 2.00 |
| NH+SH | -0.11 | -0.15 | 0.49 | 0.62 | -0.35 | -0.43 | 1.76 | 1.94 |

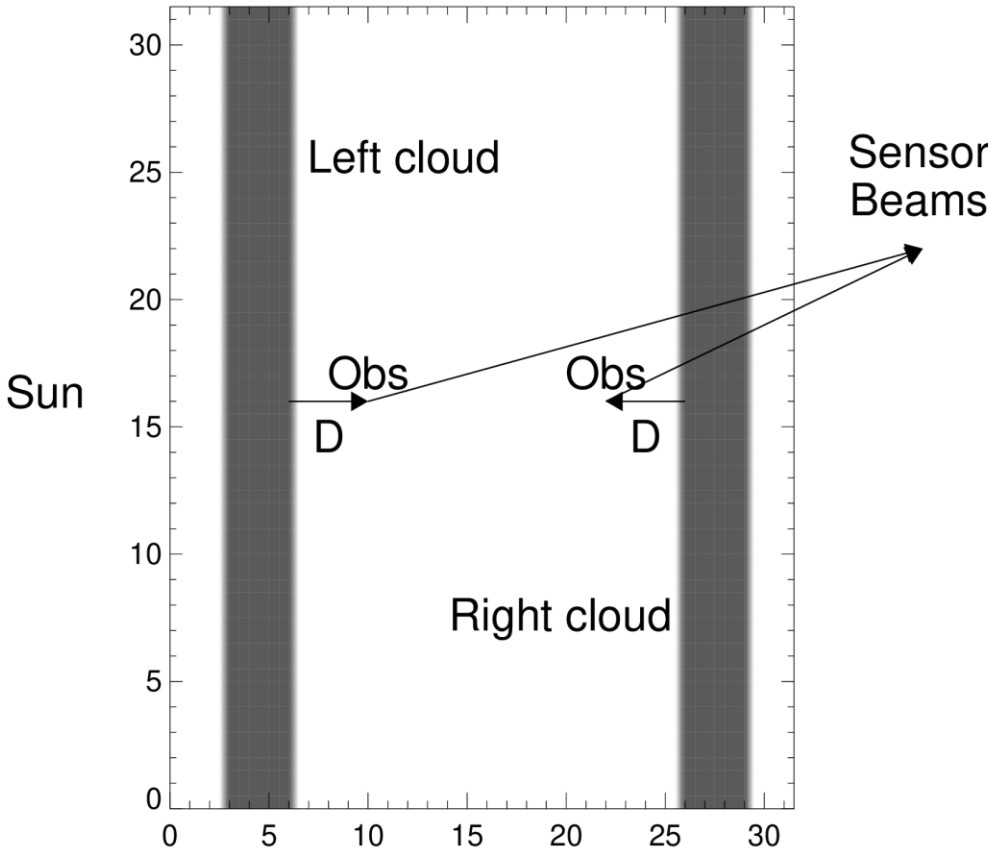

**Figure 1**. Idealized bar clouds for a cloud height of 1.4 km. The Sun and OCO2 are to the left and right of the 32 km x 32 km SHDOM scene, and two observation points are distance D from the Left and Right clouds. Two sensor beams are indicated by the upward sloping arrows.

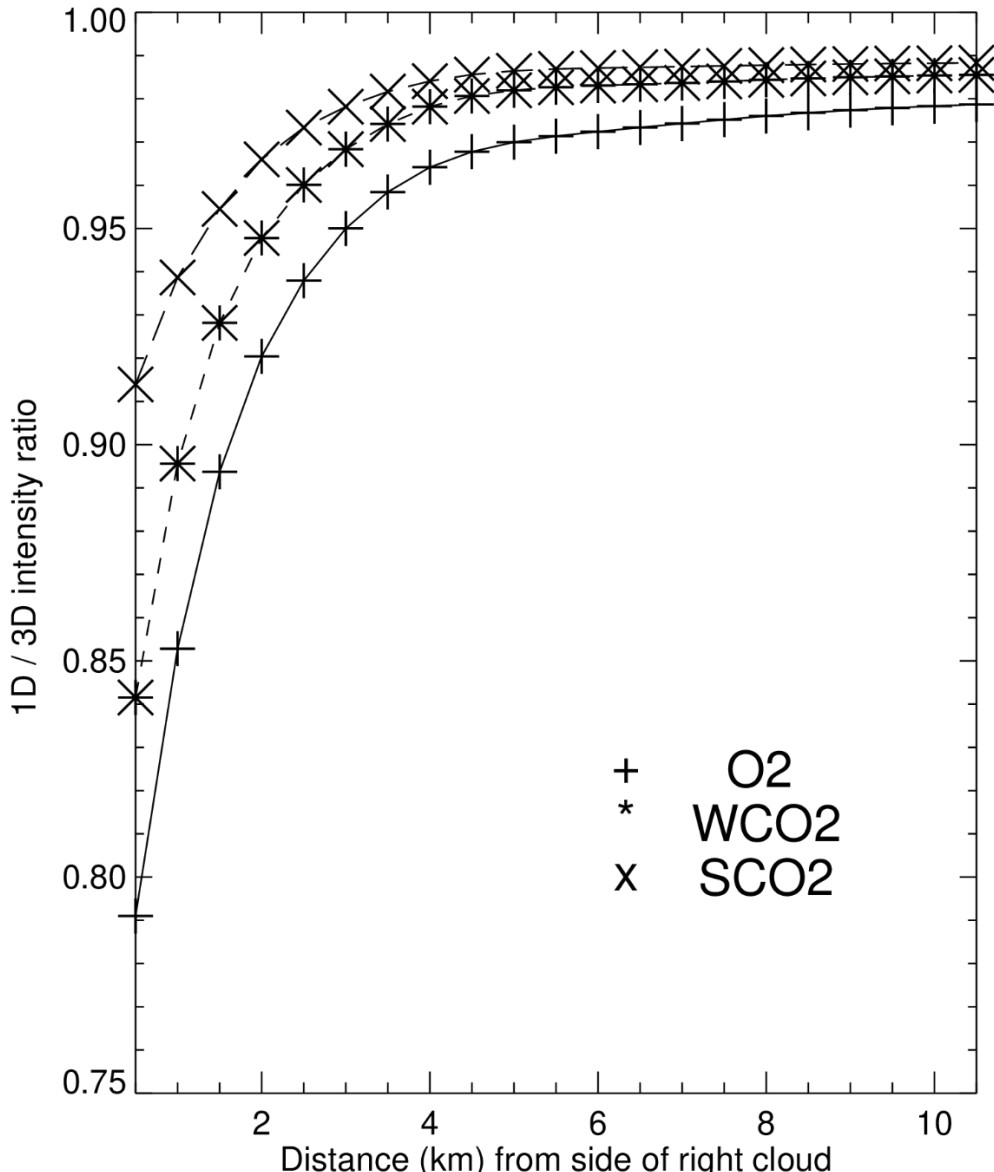

**Figure 2**. Illustration of 1D / 3D intensity ratios as a function of distance D (in km) from the Right cloud in Fig. 1. 3D cloud effects are present in all three OCO-2 bands, with largest 3D effects (smallest 1D / 3D ratios) in the $O_2$ A-band. The ABSCO vertical gas optical depth is near 0.01.

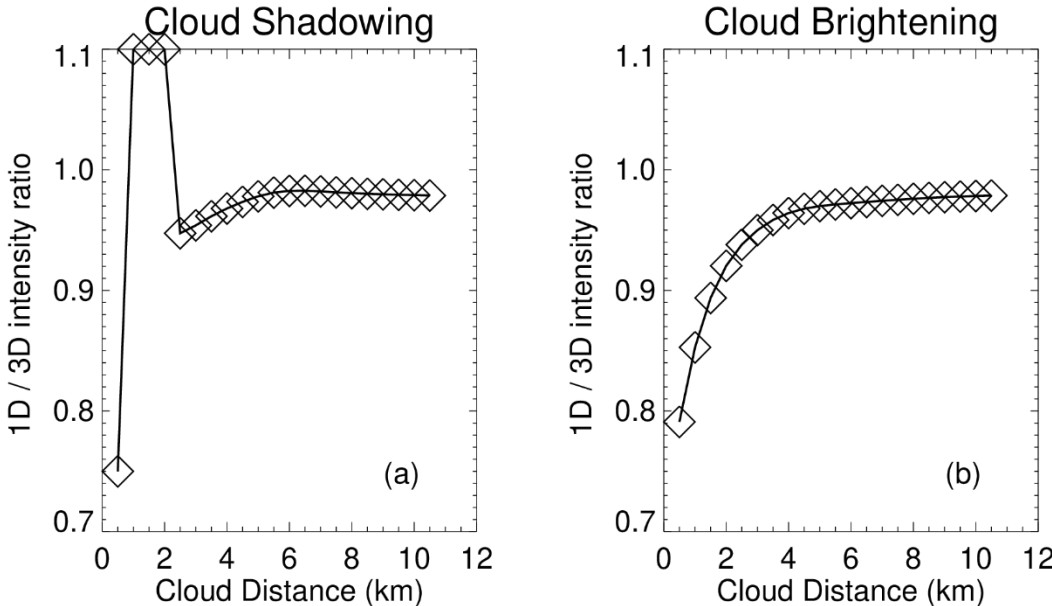

**Figure 3** 1D / 3D intensity ratios as a function of distance D for (a) Left and (b) Right idealized bar clouds, for the $O_2$ A-band continuum (vertical gas optical depth is 0.0008). Cloud shadows are present in (a) (with ratios greater than unity), and 3D cloud brightening is present in (b). The idealized bar clouds have the same altitude, pressure, temperature profile as that of the detailed SHDOM calculation of Fig. 6 for 22 June 2015 at 6.52° S and longitude -55.98° W. Ratios greater

than 1.1 in (a) were reset to 1.1, and the minimum of 0.45 was reset to 0.75, in order to have the same y axis range in both panels.

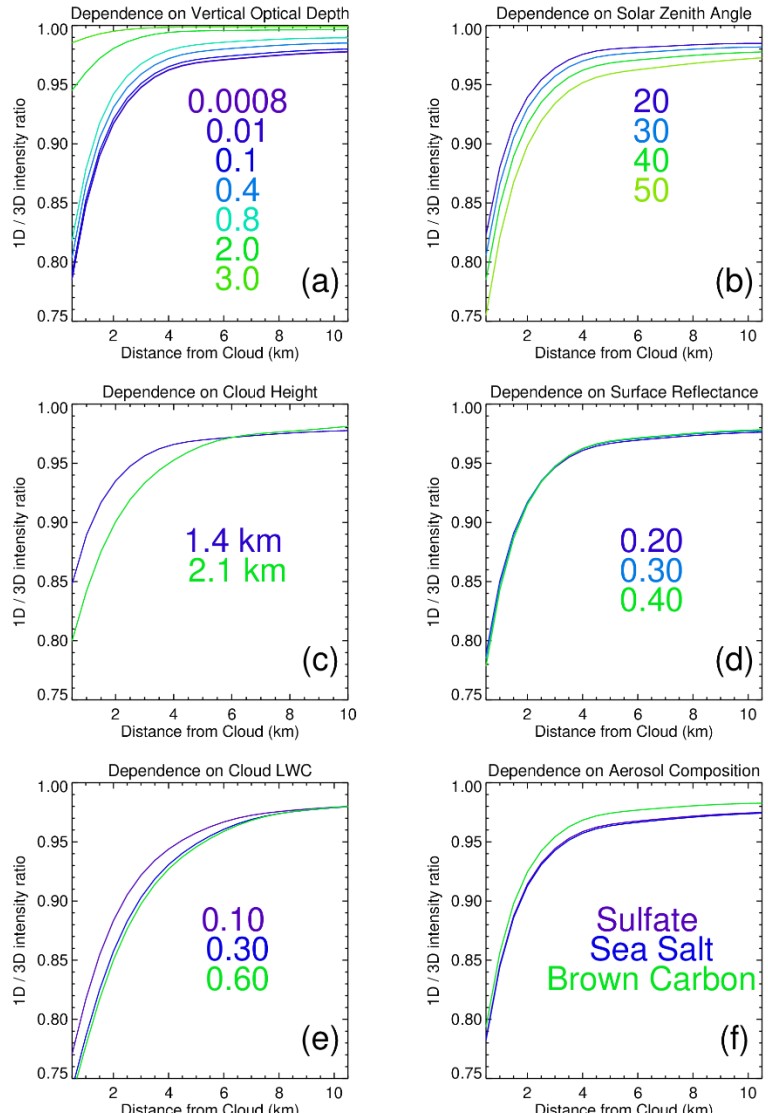

**Figure 4**. All parts of this figure pertain to a cloud brightening situation near the right-side cloud. (a) Dependence of $O_2$ A-band 1D / 3D ratios on the ABSCO gas vertical optical depth and as a function of nearest cloud distance. The cloud top height is 1.4 km, the Lambertian surface reflectance is 0.30, and the aerosol vertical optical depth is 0.16. (b) Dependence of 1D / 3D ratios on the solar zenith angle. The ABSCO gas vertical optical depth is 0.0008 and the Lambertian surface reflectance is 0.30. (c) Dependence of 1D / 3D ratios on cloud vertical thickness, labeled by the cloud top height, with a cloud base at 0.1 km. The ABSCO gas vertical optical depth is 0.0008, the Lambertian surface reflectance is 0.30, and the solar zenith angle is 40°. (d) Dependence of 1D /3D ratios on the Lambertian surface reflectance. The ABSCO gas vertical optical depth is 0.0008 and the solar zenith angle is 40°. (e) Dependence of 1D / 3D ratios on the cloud LWC (gm/ m3) value. (f) Dependence of 1D /3D ratios on the aerosol size distribution and indices of refraction.

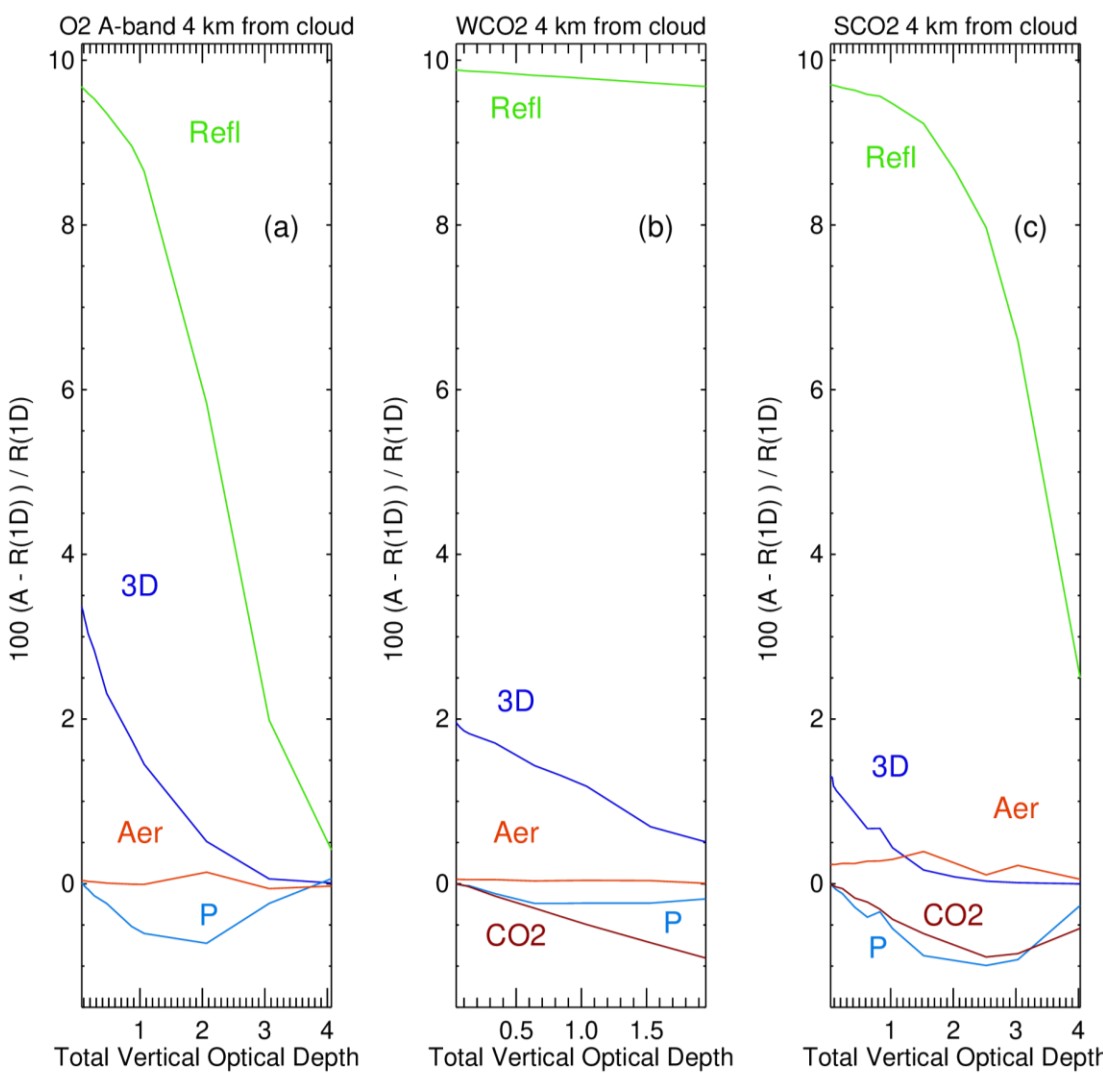

**Figure 5**. Radiance perturbations based upon an Amazon nadir scene atmospheric profile using idealized bar clouds. The Total Vertical Optical Depth on the x axis is the sum of the gas and aerosol vertical optical depths. Each vertical optical corresponds to a specific ABSCO wavelength. Lowest vertical optical depths are in the continuum portion of the spectra, while the largest vertical depths are in absorption lines. The observation point is 4 km from the cloud. "A" in the y axis label stands for the 1D perturbation radiance or the 3D radiance. Aerosol (Aer) optical depths and surface reflectivity (Refl) are perturbed by 10% from the Table 1 baseline values, while the surface pressure (P) is perturbed by 2 hPa and $CO_2$ in the WCO2 and SCO2 bands is perturbed by 1 ppm.

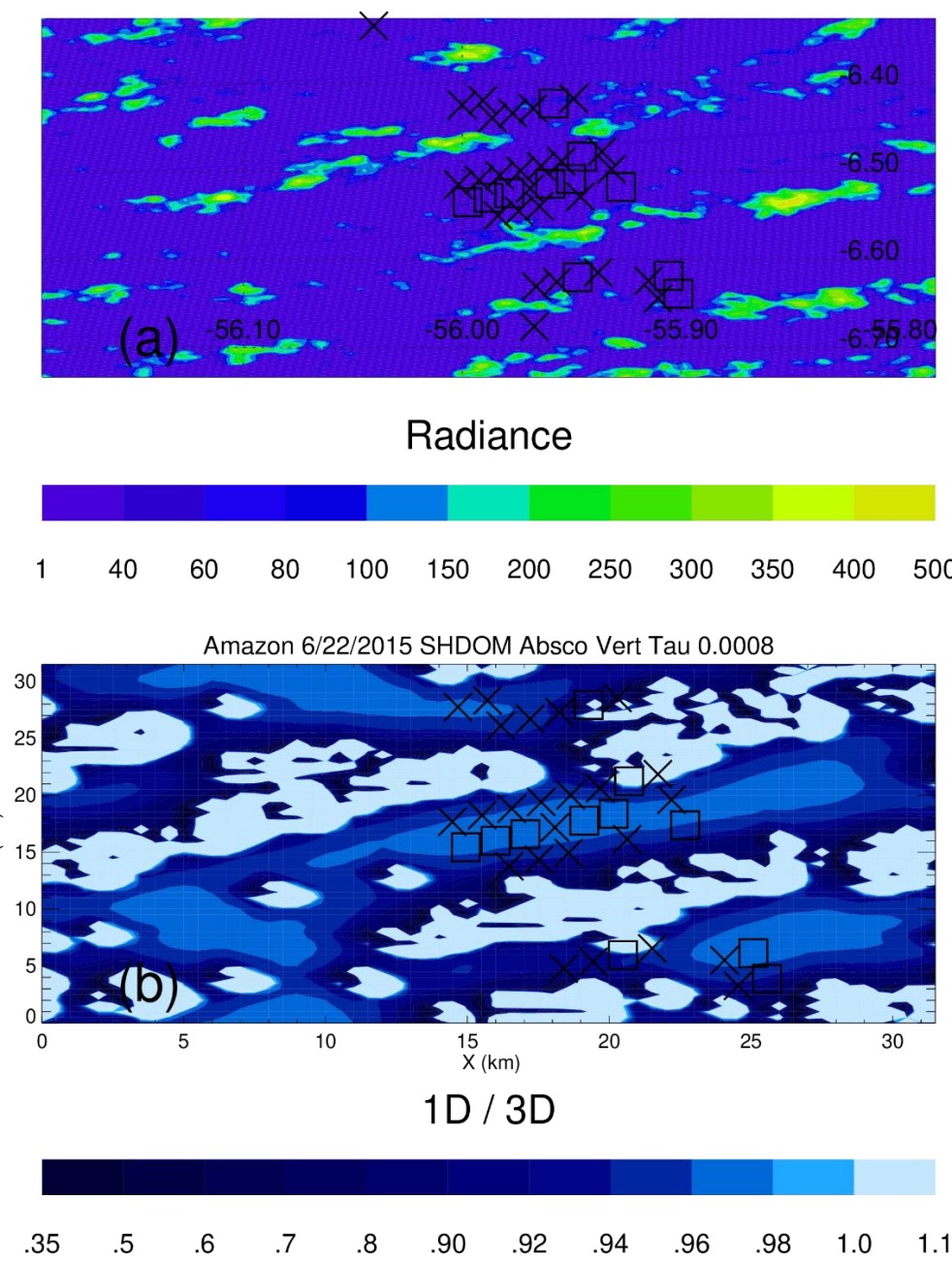

**Figure 6.** (a) MODIS radiance field (units are W/m²/sr/μm) on 22 June 2015 over the Amazon as a function of longitude and latitude, and the location of V10 Lite file observations, marked by the square □ (QF0), and X (QF1) symbols. The observations are between the clouds (the irregular green, yellow areas). The direction of the incident sunbeam is from the northwest, and north is at the top. (b) SHDOM calculation of 1D / 3D ratios for the 22 June 2015 Amazon scene. The smallest 1D / 3D ratios are located on the sunward side of clouds (white). Note that panel (b) only covers a portion of the spatial extent of panel (a).

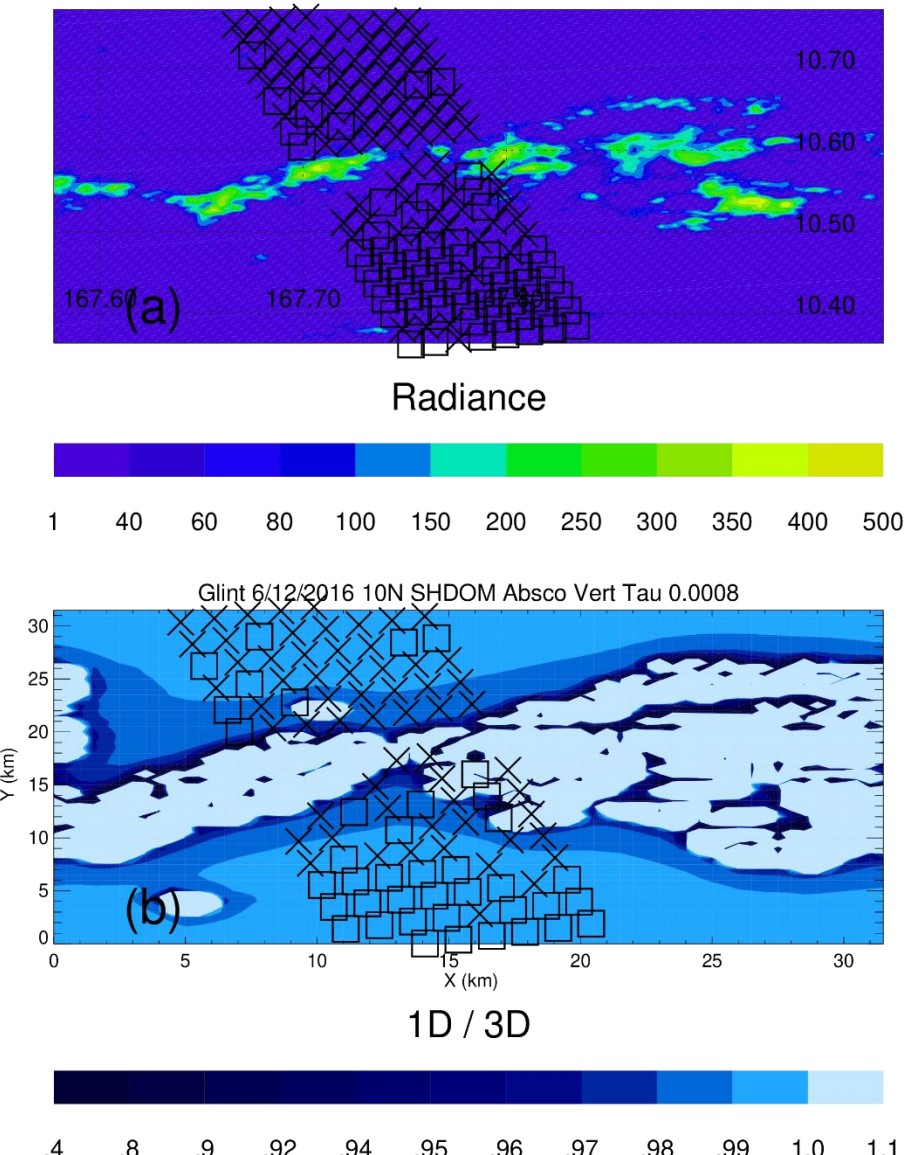

**Figure 7** (a) MODIS radiance field (units are W/m$^2$/sr/μm) on 12 June 2016 over the ocean as a function of longitude and latitude, and the location of V10 Lite file observations, marked by the square □ (QF0) and X (QF1) symbols. The observations are between the clouds (the irregular green, yellow areas). The direction of the incident sunbeam is from the northwest, and north is at the toop. (b) SHDOM 1D / 3D ratio field for the 12 June 2016 ocean glint scene. Notice the increase in the 1D / 3D ratios as distance from clouds increases, with clouds corresponding to areas with 1D / 3D ratios greater than unity. Panel (b) only covers a portion of the spatial extent of panel (a).

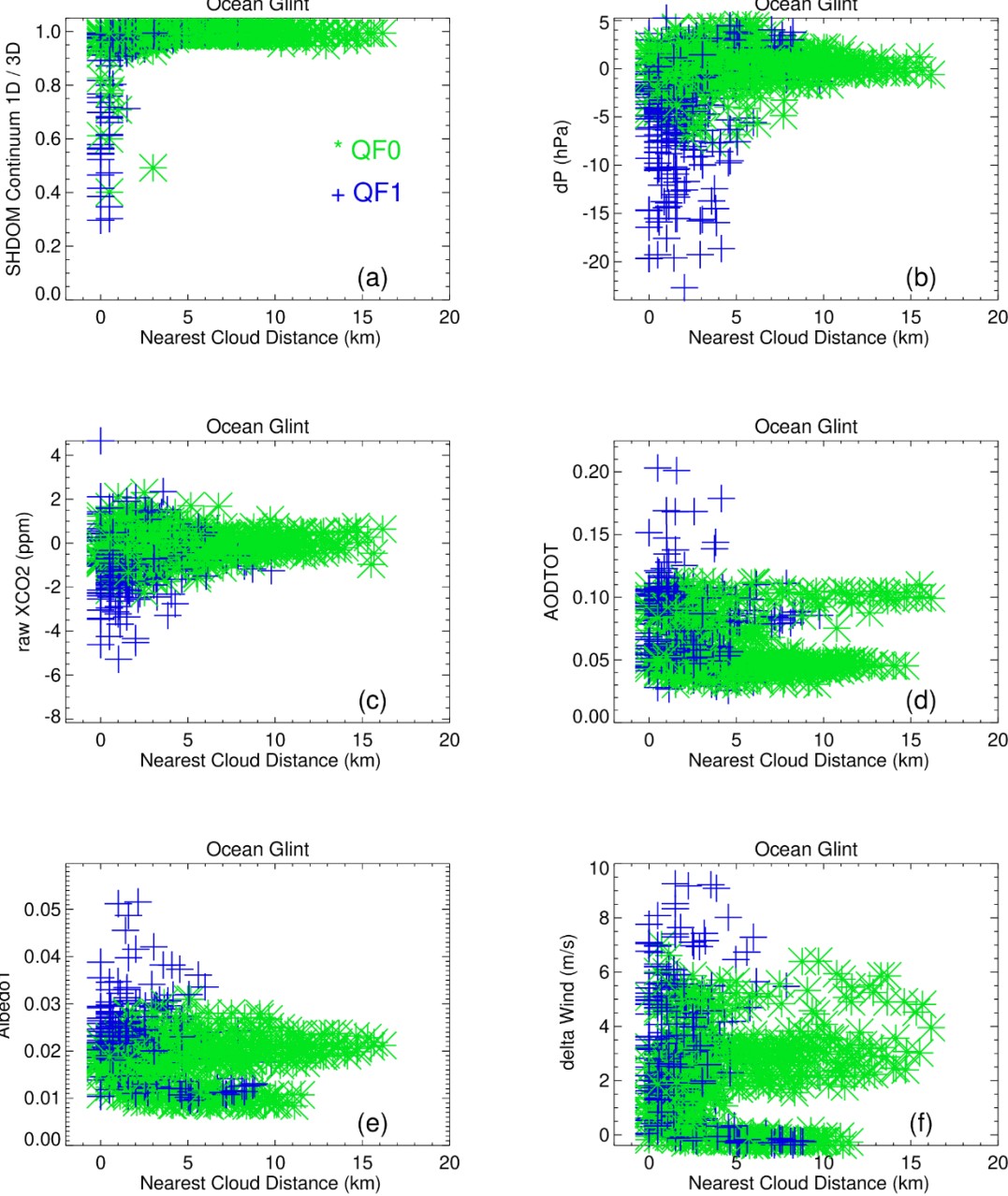

**Figure 8**. Individual Lite file QF=0 (*, green) and QF=1 (+, blue) SHDOM continuum 1D / 3D, OCO-2 dP, raw XCO2, total aerosol optical depth, Albedo1 (for the $O_2$ A-band), and "delta Wind" (see text), as a function of nearest cloud distance for the 12 ocean glint scenes.

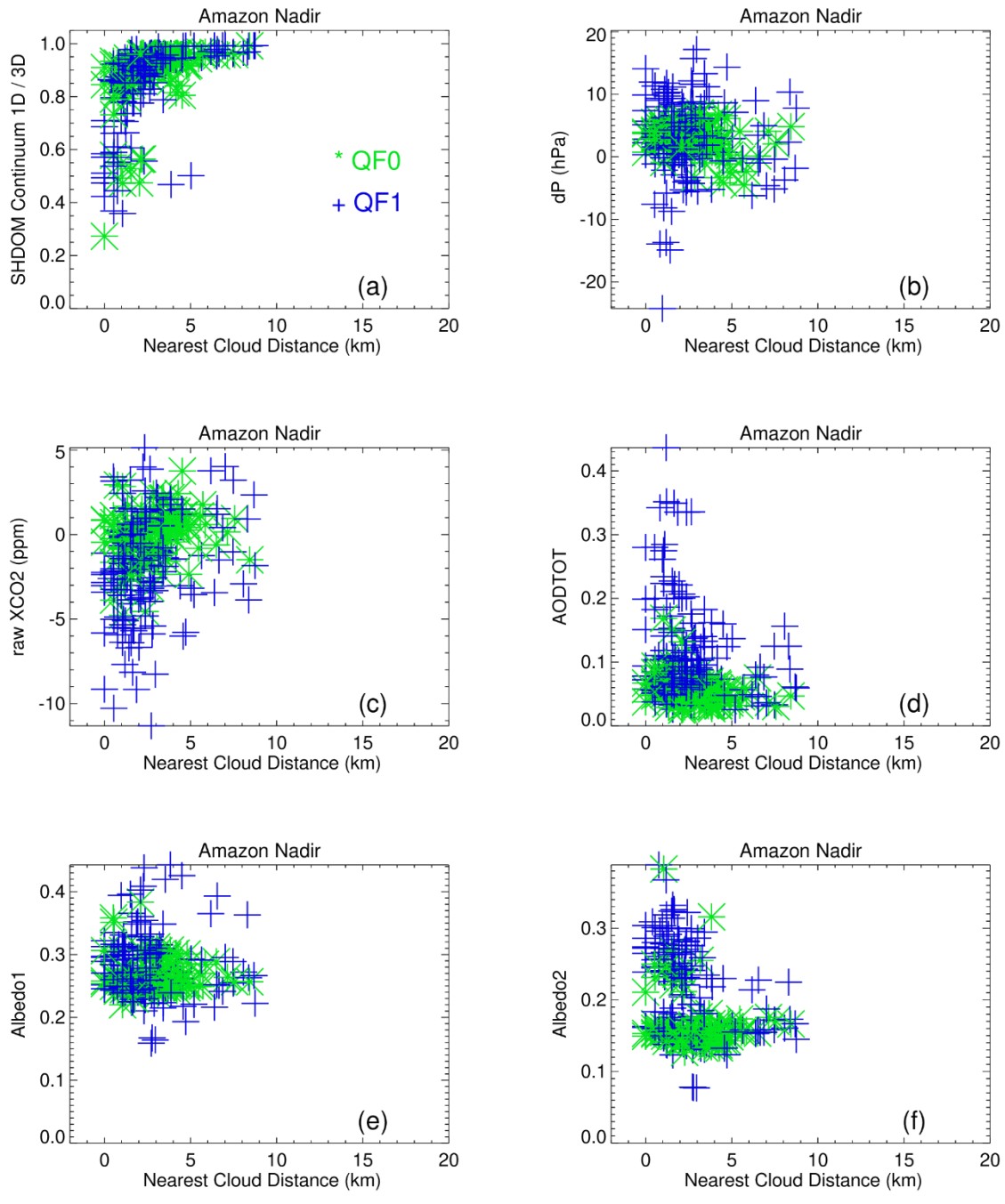

**Figure 9**. Individual Lite file QF=0 (*, green) and QF=1 (+, blue) SHDOM continuum 1D / 3D, OCO-2 dP, raw XCO2, total aerosol optical depth, Albedo1 (for the $O_2$ A-band) and Albedo2 (for the WCO2 band) data as a function of nearest cloud distance for the 12 Amazon nadir scenes.

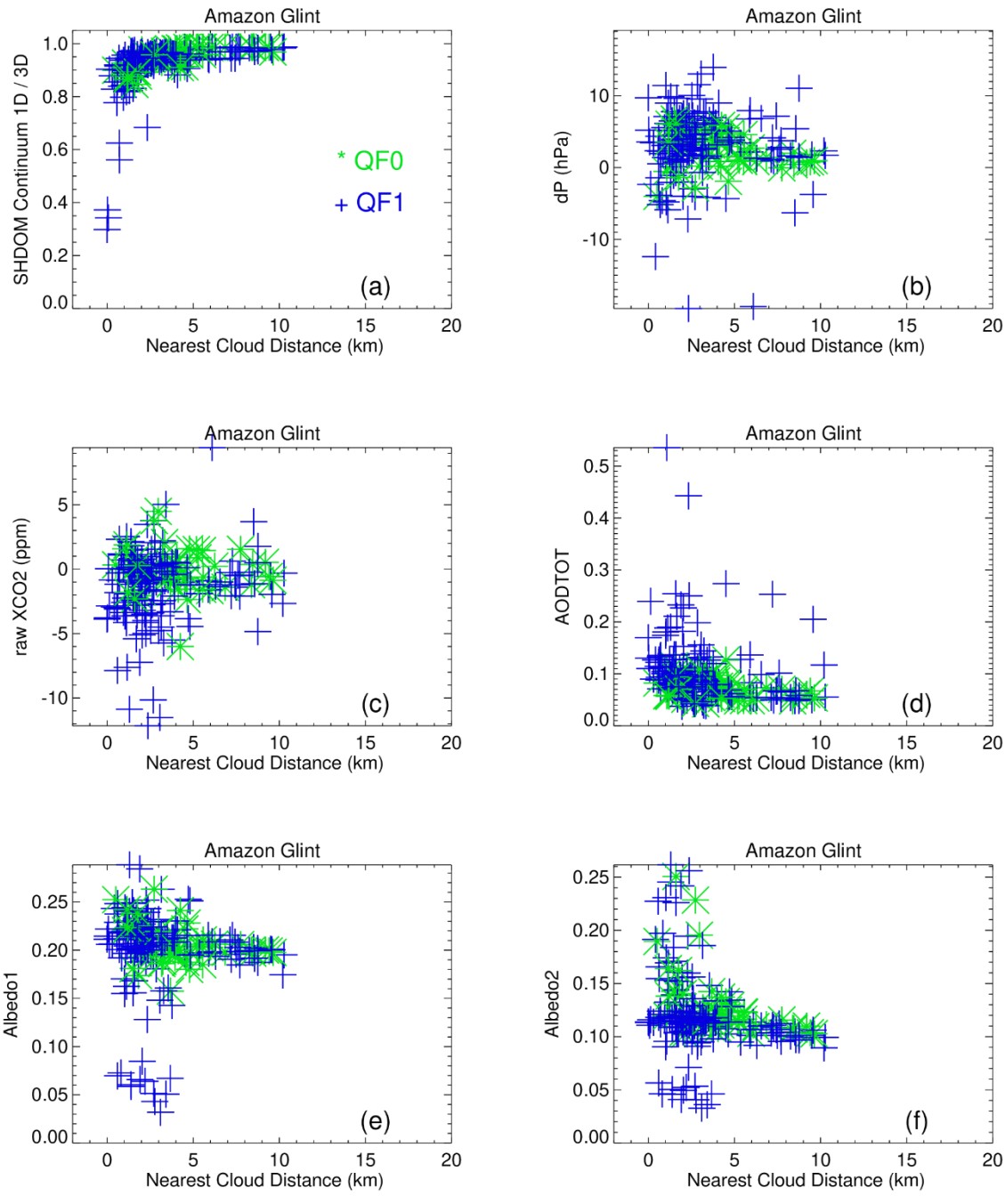

**Figure 10**. Individual Lite file QF=0 (*, green) and QF=1 (+, blue) SHDOM continuum 1D / 3D, OCO-2 dP, raw XCO2, total aerosol optical depth, Albedo1 (for the $O_2$ A-band) and Albedo2 (for the WCO2 band) data as a function of nearest cloud distance for the 12 Amazon glint scenes.

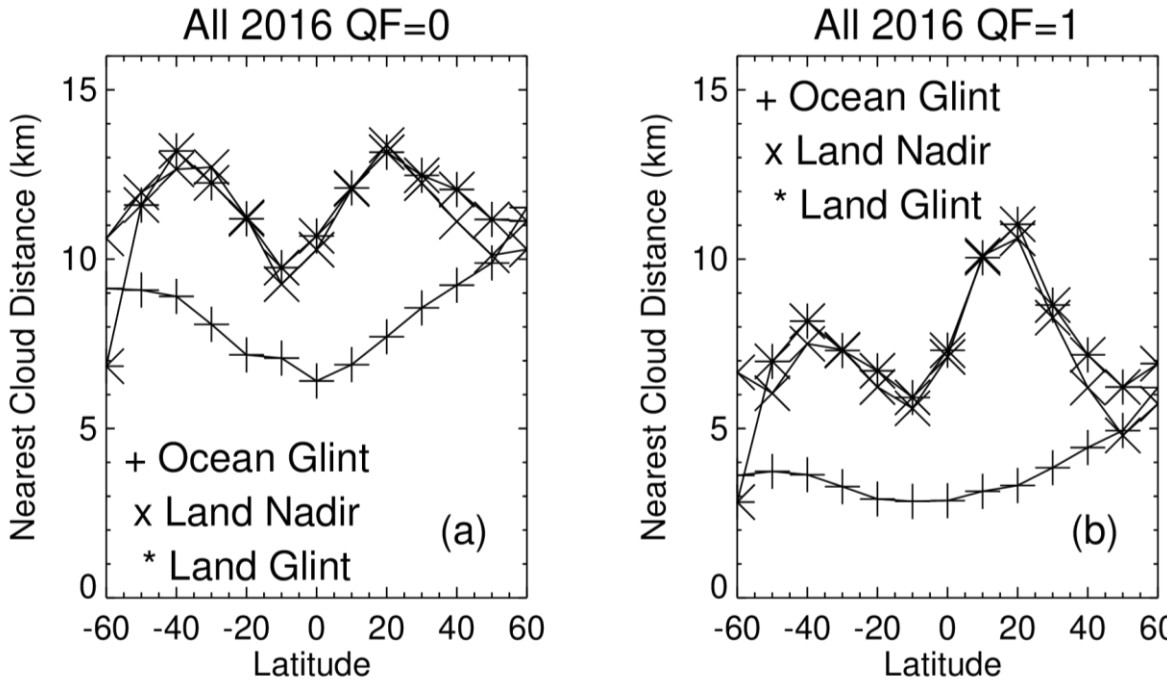

Figure 11. Zonal averages of nearest cloud distance over ocean and land for 2016 QF0 and QF1 data.

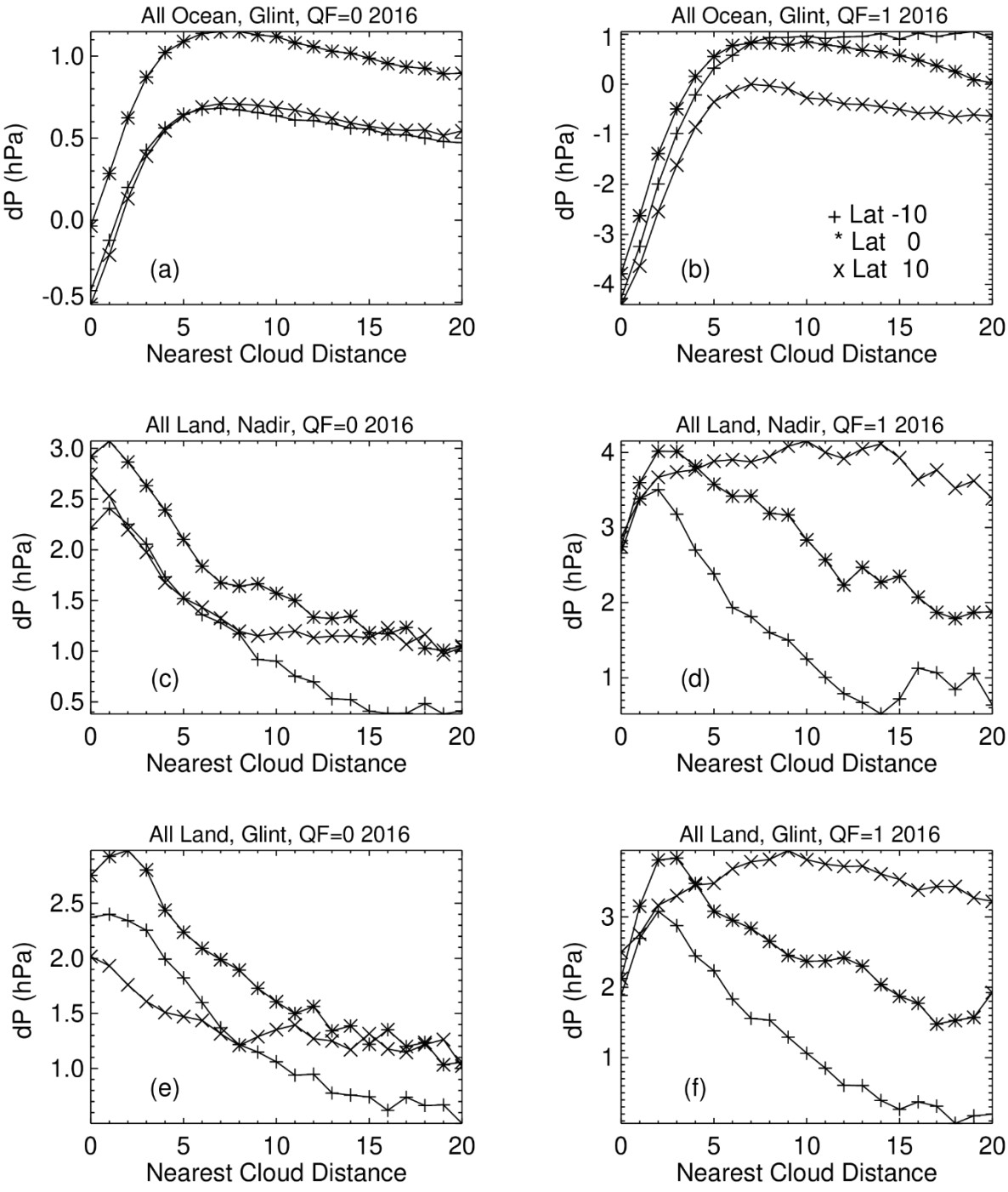

**Figure 12**. dP averages as a function of nearest cloud distance. -10 refers to the 10° S - 0° S latitude band.

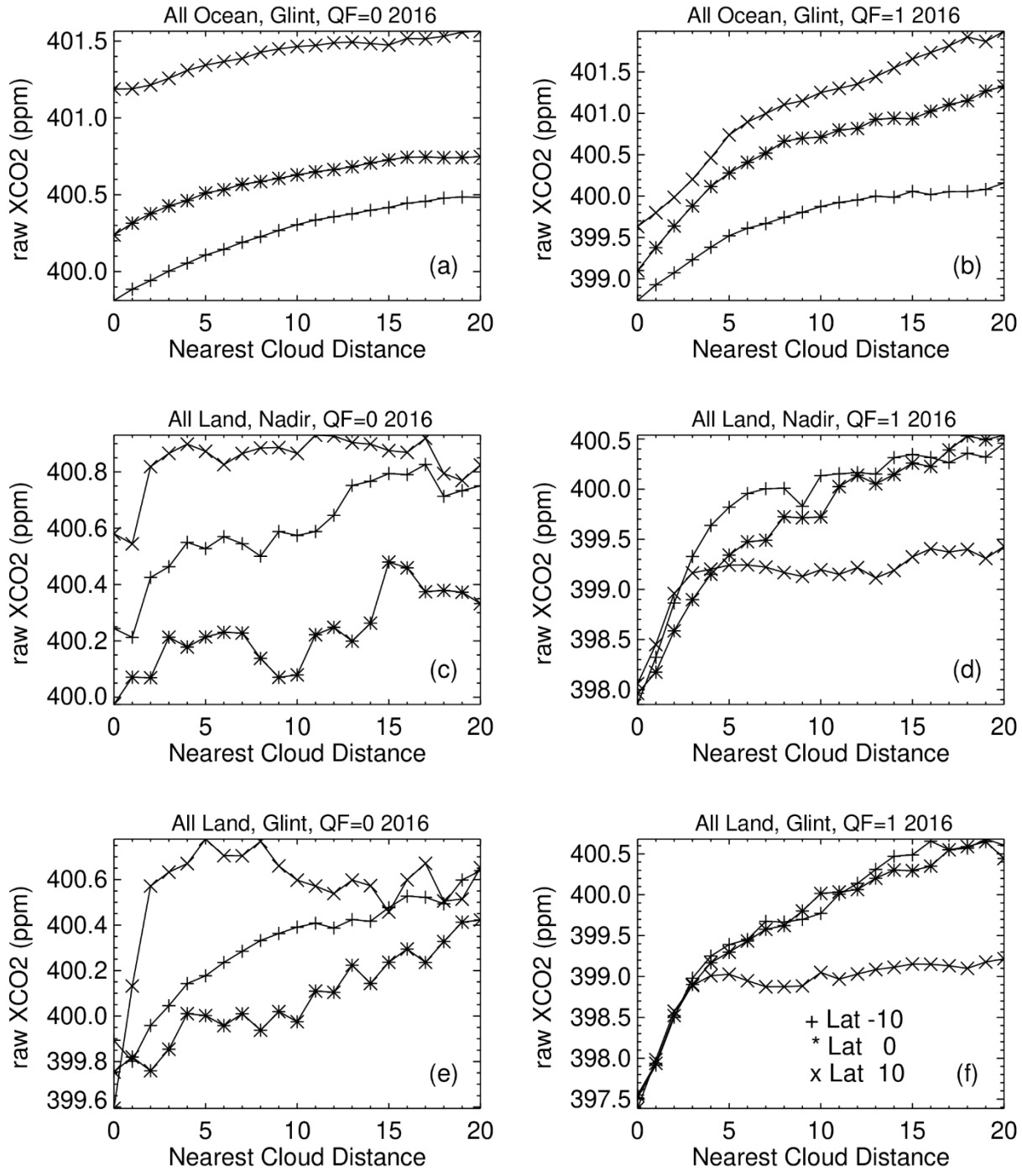

**Figure 13**. Raw XCO2 averages as a function of nearest cloud distance. -10 refers to the 10° S -
0° S latitude band.

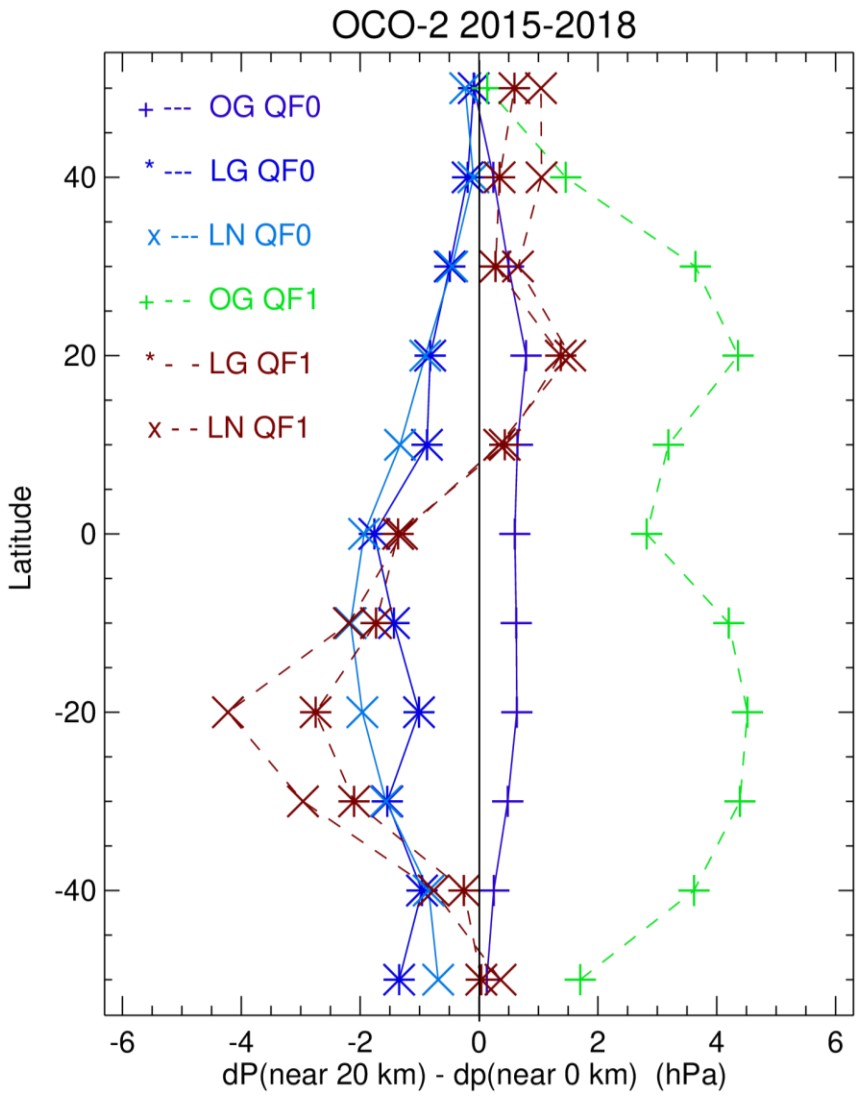

**Figure 14.** Latitudinal averages of dP (near 20 km) – dP (near 0 km) for 2015 – 2018 for QF0 (solid line) and QF1 (broken line) data. OG, LN, and LG refer to ocean glint, land nadir, and land glint observing modes. -40 refers to the 40° S - 30° S latitude bin.

1150

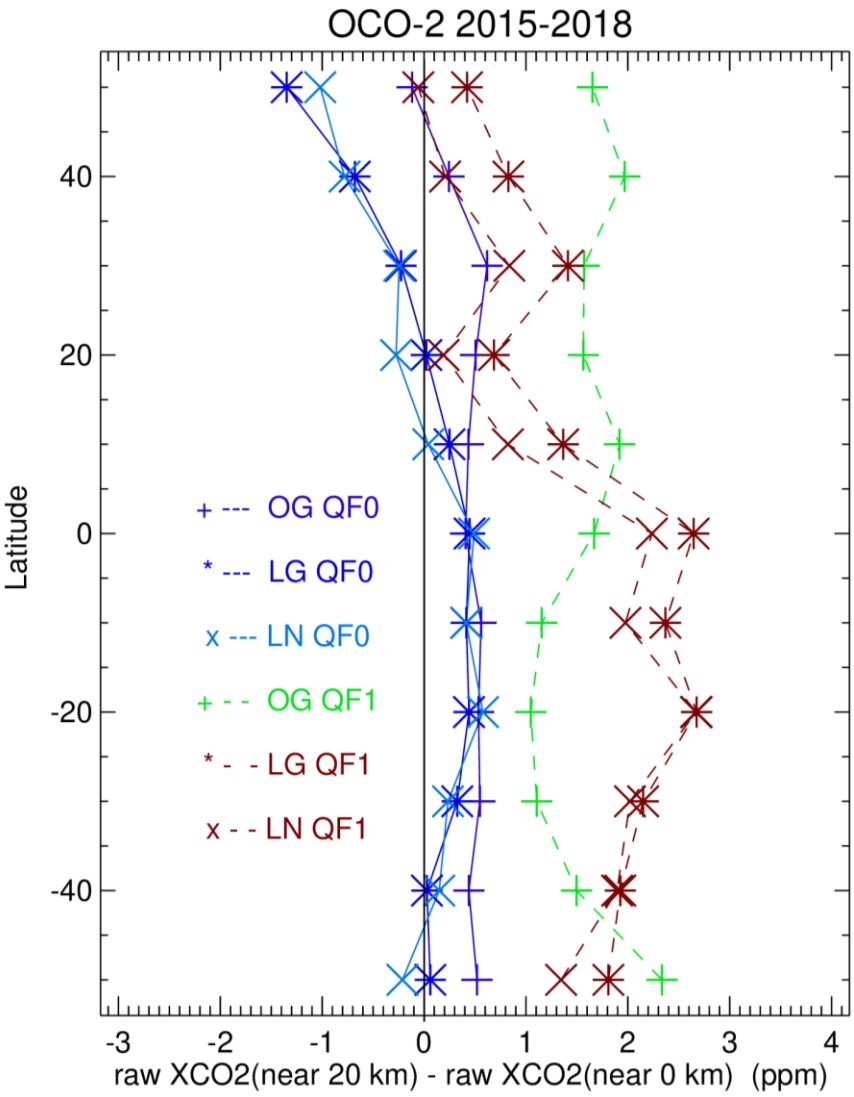

**Figure 15**. Latitudinal averages of raw XCO2 (near 20 km) – raw XCO2 (near 0 km) for 2015 – 2018 for QF0 (solid line) and QF1 (broken line) data. OG, LN, and LG refer to ocean glint, land nadir, and land glint observing modes. -40 refers to the 40° S - 30° S latitude bin.

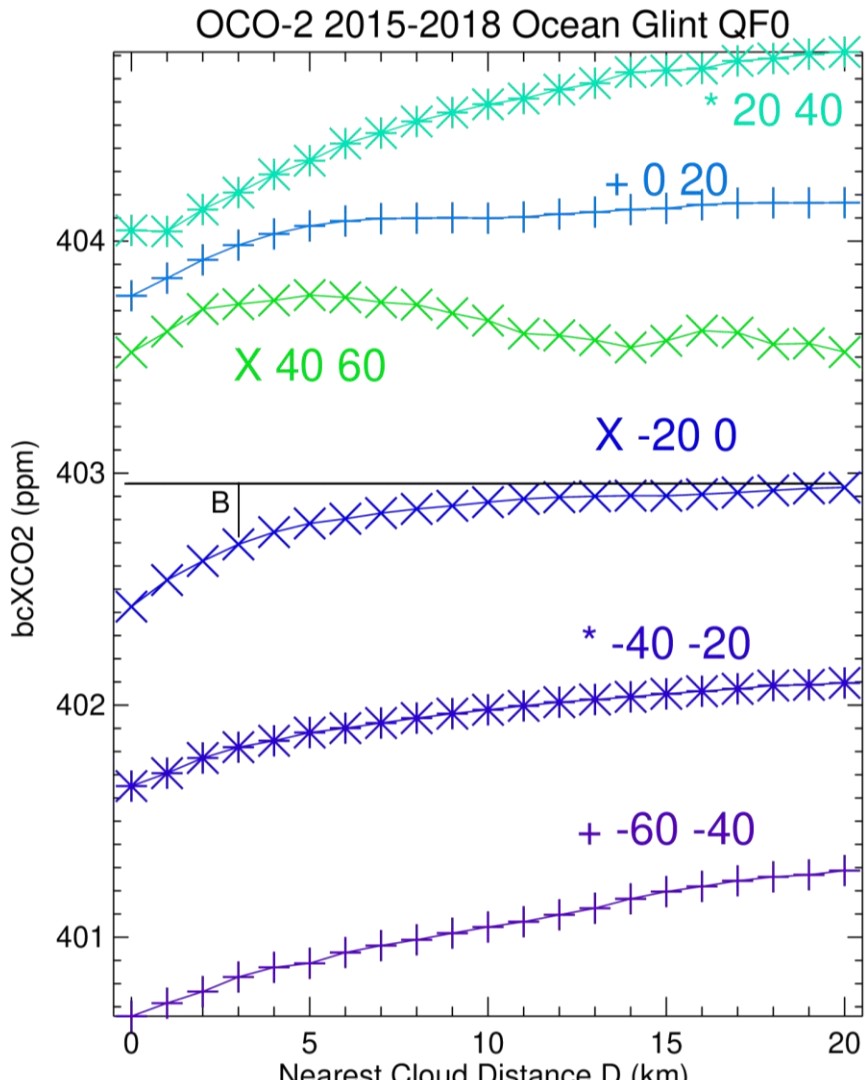

**Figure 16**. Latitudinal variation of bcXCO2 as a function of nearest cloud distance. The black lines illustrate how one calculates the bias B value for each cloud distance for each latitude band. Using the -20° to 0° latitude band as an example, the value of B is 0.25 ppm when the nearest cloud distance is 3 km.

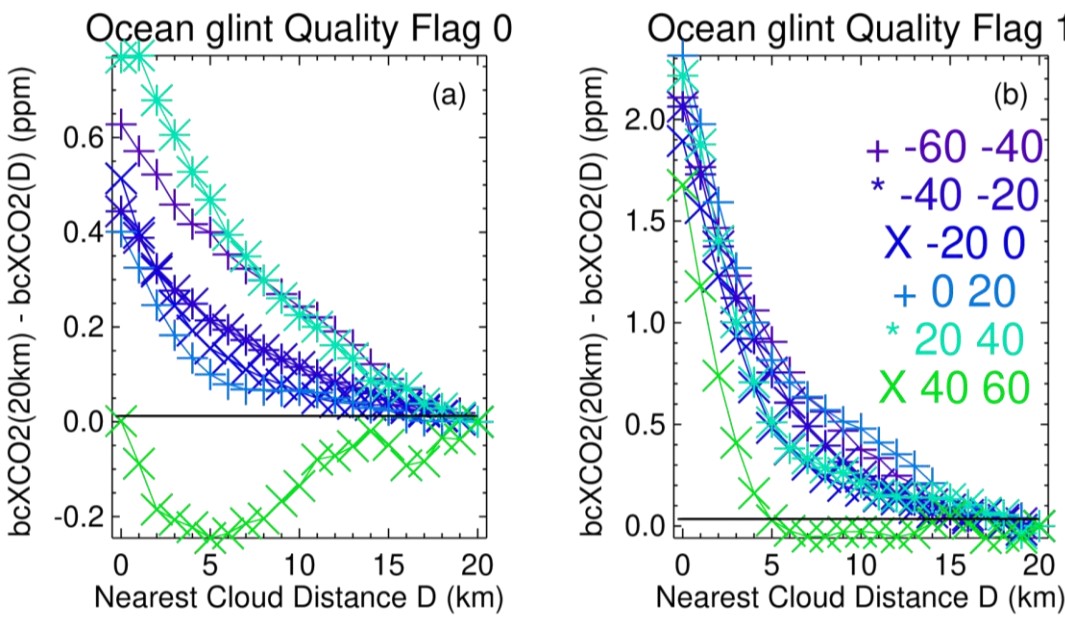

**Figure 17**. The 3d cloud effect biases (ppm), calculated from Fig. 16 B values, as a function of latitude for (a) QF0 and (b) QF1 data.

1170

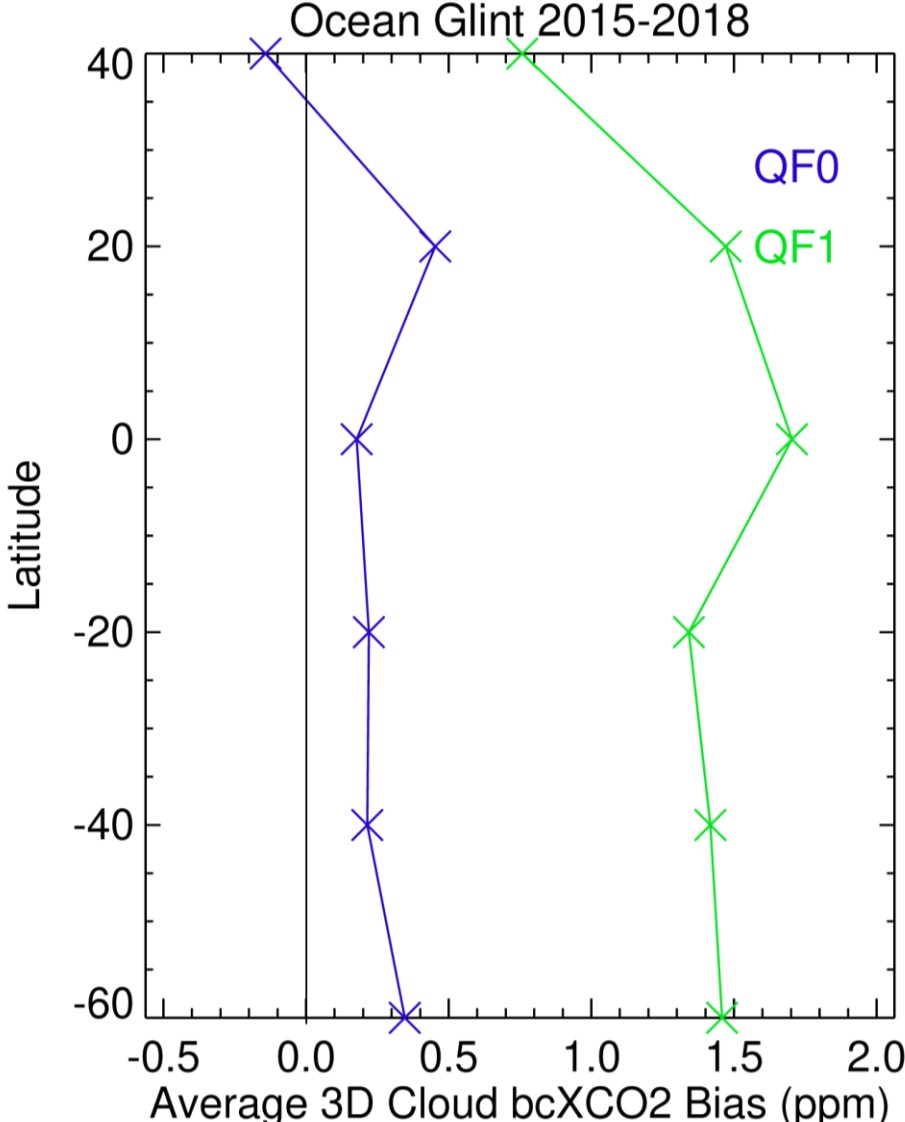

**Figure 18**. 3D cloud effect average biases as a function of latitude for QF0 and QF1 data. The average biases are those based on Fig. 17 data, weighted by the number of observations in each of the nearest cloud distance bins for observations in 2015 – 2018. -40 refers to the 40° S to 20° S latitude bin, etc.

1175