# Peer review of "Insights into 3D cloud radiative transfer effects for OCO-2"

_Atmospheric Measurement Techniques, 2022_

## Referee Comment (RC1)

**Review: Insights into 3D cloud radiative transfer for OCO-2**

S. Massie, H. Cronk, A. Merrelli, S. Schmidt, S. Mauceri
January 13, 2023

**1  Principal criteria**

- **Scientific significance:** Does the manuscript represent a substantial contribution to scientific progress within the scope of Atmospheric Measurement Techniques (substantial new concepts, ideas, methods, or data)?
  **FAIR**

- **Scientific quality:** Are the scientific approach and applied methods valid? Are the results discussed in an appropriate and balanced way (consideration of related work, including appropriate references)? Note that papers do not necessarily need to be long to be scientifically sound.
  **GOOD**

- **Presentation quality:** Are the scientific results and conclusions presented in a clear, concise, and well-structured way (number and quality of figures/tables, appropriate use of English language)?
  **FAIR**

- **Reviewer recommendation:** **ACCEPT WITH MAJOR REVISIONS**

**2  Review summary**

This paper provides an evaluation of the effects of three dimensional (3D) radiative transfer (RT) on the spectra measured by NASA's Orbiting Carbon Observatory -2 (OCO-2) due to clouds within close proximity (several to tens of kilometers) of the ground footprints. Clouds impose a 3D effect that increases as the distance to the nearest cloud decreases. The 3D effect is currently unaccounted for in the full physics retrieval algorithm that is used to derive estimates of total column carbon dioxide (XCO2) from the measured radiances. Using simulations of 1D and 3D radiances (from the SHDOM RT model) for realistic OCO-2 scenes, the authors demonstrate that 3D perturbations to the radiances are of the same order (or greater) than reasonable perturbations to other key retrieval state vector elements, such as surface pressure and aerosol loading. The effect is shown to be significant relative to the precision requirements of the mission on the order of 1 part per million (ppm). It is therefore important to characterize and quantify the 3D effect in order to continue improving the XCO2 estimates in future versions of the data products. **Overall, the paper is well organized and written in fluent English, with appropriate figures to describe the results of the analysis. However, due to a couple of significant concerns, I have to recommend that major revisions be performed.** My primary constructive comments are itemized below, approximately in order of importance.

- **It seems to me that the current work is very similar in nature to the work presented in [Massie, AMT, 2021].** The only clear difference I can determine is that the current work utilizes "bar cloud" calculations, which were not present in the 2021 manuscript. The authors need to make it explicitly clear how the implementation of bar clouds in the current research leads to either substantially different results compared to [Massie, AMT, 2021], or corroborates the previous results. Otherwise, I do not really understand the purpose of the current publication. Is it fair to say that the current research refines the 2021 research, and demonstrates the 3D effect using more first principle physics (or something like that)? In general, the previous publication did clear up some of my questions related to the current research. For example, the description of Fig. 2 in [Massie, AMT, 2021] is more clear than the description of Fig. 5 in the current research. In effect, the figures are the same.

- **The overall conclusions drawn from the analysis are not clear or strong enough.** I'd prefer the conclusions section to be more succinct and void of the analysis details (which should appear in Sections 3 through 6). The authors effectively demonstrate the high frequency of occurrence of the cloud 3D effect on the OCO-2 measurements (40% of all OCO-2 measurements are within 4 km of a cloud!). They effectively demonstrate that the average 3D perturbations will cause errors in the XCO2 estimates on the order of 0.5 ppm, which is highly significant to the scientific goals of the OCO-2 project. However, they also effectively demonstrate, by

analysis of both good (QF=0) and bad (QF=1) data, that the current operational OCO-2 quality filtering routinely identifies the scenes with larger 3D contamination, which would thereby have larger XCO2 errors. **The main criticism is that no real conclusion or recommendation is drawn about the potential for correcting or improving future versions of the OCO-2 XCO2 product based on the results presented in this analysis.** This work seems to be an important stepping stone towards a robust solution, but that thought is never expressed. Ideally, in the future, some methodology could be put into place to "save" the "bad" quality flagged (QF=1) data, or to minimize the size of the d$p$ bias correction term necessary to correct the raw XCO2, but the authors never really say that directly. As it currently stands, one could make the argument that the operational QF/BC is already doing a pretty good job of accounting for the 3D cloud effect, so why should additional effort/resources be put into mitigating it using (closer to) first principles physics? I recommend that the authors address this issue, at least in theory, and discuss proposed future work. Furthermore, I recommend reducing the verbiage in the Conclusions section itself. Many of the sentences there are very detailed and seem misplaced, as if they should belong in the technical discussion in the main body of the text. I have a number of specific recommendations in the technical corrections below.

- **Any new conclusive thoughts should also be amplified in the Abstract to bring the message home quickly to the casual reader.** I feel like in general, the abstract is a bit too detailed and could use more focus on the "big picture" of why the work is important, how/why it is a difficult problem to address, and what the possible solutions might be.

- **Interpretation of Fig 5. and the discussion of the results in Sec. 3.3 (radiance perturbation sensitivity) are of paramount importance to this work, but unfortunately, as currently written they are quite confusing.** I wonder if both the figure and the discussion could be reorganized, or at least better explained, to aid the reader. After a while I think I determined that the x-axis is the 17 values of the Total Vertical Gas Optical Depth (TVGOD) that is described on Line 217. That point needs to be made explicit, as I got stuck thinking that these were iterations of aerosol optical depth in the retrieval for sensitivity testing. But when I read the analysis of the results in Sec 3.3 I cant really follow along in the figure. For example: Lines 312-316 suggest that there will be an increase in radiance if surface reflectance or AOD are increased. What am I supposed to see in the figure that suggest that, because I cannot see it. Are the results in Fig. 5 due to a single perturbation of each variable? If so, then are the results robust to further perturbations? If the figure already somehow contains results for multiple perturbations of state vector values, then I dont understand how that is displayed. That to me would suggest that the TVGOD on the x-axis is not in fact a proxy for channel/wavelength. My impression is that the graph is meant to demonstrate that the 3D effect is much larger than the effect due to perturbing the state variables, seen by the fact that the 3D curve is far away from zero for most TVGOD, while most of the other variables are close to zero at all TVGOD. The exception being Refl.

- Some discussion should be given (might be very brief) as to why the 3D effect cannot be accounted for in the L2FP retrieval simply by embedding the SHDOM RT model to calculate it directly due to the massive computation cost, plus difficulty in parameterizing 3D cloud effects. I'm not sure that this is ever mentioned explicitly (?).

- In Sec. 4 (Amazon and Ocean Glint scenes), a large number of lines (about 40, 6% of total paper) are used to describe the details of the calculation for determining the fraction of cloud shadowing scenes compared to the fraction of cloud brightening. It felt like a bit of a distraction since the conclusion is that cloud shadowing only occurs for on the order of 1-4% of the converged retrievals. I wonder if the details of the cloud shadowing determination would be better moved to a technical appendix. Or, perhaps the authors could work to shorten the description if they prefer to leave it in the main body of the document.

- In general, throughout the text, I found the mathematical notation unwieldy. I would recommend making improvements to aid the readers ability to interpret. Specific examples are given below in the Technical errors/corrections section.

- For all of the results presented in the various sections, it seems like a number of important assumptions were made. I have a feeling that these were all well thought out, but there is no indication as to the robustness of the results for different sets of assumptions. Specific examples are given below in the Technical errors/corrections section.

- The scope of the research is limited to the Amazon for land observations, as well as ocean observations. I understand the interest and the fact that the cloud morphology is such that the 3D effect is highly relevant. But have other regions been examined? Are the results robust? I'm thinking that regions with brighter surfaces will likely have additional complications of multiple scattering effects, perhaps yielding different 3D contributions (potentially even in sign)? Please state clearly what you have and haven't studied in this work, and that only Amazon forest  oceans have been examined, and potentially also mention this as future work, as characterizing 3D effects over different surfaces is an important follow-on.

- Most of the figures would benefit from the use of sub panel lettering, e.g., (a), (b), (c).

- The title and all of the analysis/discussion is focused on OCO-2. But the general principles are applicable to OCO-3. Might we worth mentioning in some form or fashion to broaden the utility. (I'm aware that the "distance to nearest cloud" data does not exist for OCO-3, making actual analysis difficult/impossible.)

- I wonder if there is a tie-in to the results presented in Emily Bell's swath-bias paper (https://doi.org/10.5194/amt-16-109-2023)? Although that work was specific to biases in XCO2 for OCO-3 Snapshot Area Maps, statements, such as;

  "These simulation studies reveal that SB is primarily and intimately connected to the presence of aerosols and the interplay of their optical properties with the solar and instrument viewing geometries. We now have a better understanding of the types of scenes that are likely to suffer from SB – those with high aerosol depths, or aerosols that are lofted higher in the atmospheric column, and in geometries with broader ranges of observation and solar zenith angles. Future work may involve a more detailed study of how the physics of aerosol optical properties with viewing and solar geometries combine to produce an SB response."

  seem highly relevant to your cloud 3D work. Just something to consider.

**3   Technical errors/corrections**

- Title: might be useful to insert the word "effects" between "transfer" and "for".

- Abstract: Line 33 provides summary result for ocean glint, but does not mention land results.

- Introduction: Line 39 mentions XCO2 measurement accuracy of $0.25\%$ ($\approx 1$ ppm). Note that although this is the formal project requirement, I think the science is suggesting that biases on the order of 1 ppm would actually be much to large for source/sink inversion work, e.g., [Chevallier, GRL, 2014, Toward robust and consistent regional CO2 flux estimates from in situ and spaceborne measurements of atmospheric CO2]

- Line 44: "The OCO-2 experiment" could/should be "The OCO-2 satellite".

- Introduction: would be good to add a sentence or phrase in the satellite description (near line 50) that mentions sun-sychronous polar orbit moving from SE to NW. That seems relavant to some of the figures and discussions of the sun/sensor geometries.

- Line 51: mention ACOS version (v10).

- Line 52: This sentence uses "XCO2" where I think "CO2" is the appropriate term. XCO2 is a calculated quantity and is not directly measured in the WCO2 and SCO2 as suggested. And I'm not sure that XCO2 was defined?

- Line 53: someone recently informed me that the Rodgers 2000 citation should be replaced with Rodgers 2004, which was a reprint that corrected a lot of typos (small thing). @bookRodgers:2004:InverseMethods, address = 5 Toh Tuck Link, Singapore 596224, author = Clive D. Rodgers, edition = Reprint, isbn = 981022740X, isbn-13 = 9789810227401, keywords = Atmospheric Science, pages = 238, publisher = World Scientific Publishing Co. Pte. Ltd., title = Inverse Methods for Atmospheric Sounding: Theory and Practice, year = 2004

- Line 56: mention which version of the ABSCO files (v5.1) are used in the ACOS v10 retrieval.

- Lines 56-58: Seems like the sentence describing the specific advances in ABSCO v5.1 is unnecessary. Is this information used anywhere else in the paper directly?

- Line 62: check spelling for TCCON Wunch citation!

- Line 69: Equation (1) seems unnecessary. Is this information used anywhere else in the paper directly?

- Lines 64-72 describes in some detail the various individual bias correction parameters. I'm not sure if this is really needed? For the most part these terms are not used again, with the definite exception of dp. I would recommend either remove the descriptions if they are not needed, or alternatively, discuss somewhere in the paper how they may relate to the 3D cloud effect.

- Line 72: The description of the bias correction increasing XCO2 by 2 ppm is misleading. It sounds like just a single global offset. There are a number of parameters (you described just above), a footprint correction term and global land/ocean values. You would probably be best off to just point to the details in the OCO-2 v10 Data User's Guide, which I dont think you currently cite. Alternatively, as mentioned above, you could flesh out more in the results discussion how the 3D effect may (or may not) influence the necessary bias correction and whether mitigation of the 3D effect could in theory mitigate the bias correction. That could be interesting since great effort is spent to achieve small improvements in the successive versions of the XCO2 product.

- Line 74: Could be useful to cite some of the UQ publications at the end of this discussion, e.g., Connor, AMT, 2016, 10.5194/amt-9-5227-2016 and/or Hobbs, JUQ, 2017, 10.1137/16M1060765.

- Line 81: "Of the [approximately one] million...collected by OCO-2, [only] about 25%...passed into the operational retrieval [due to prescreening for scenes contaminated by clouds and heavy aerosol loadings]."

- Line 83: replace "radiances" with "soundings"

- Line 83: The sentence "A quick determination..." should be preceded with "One preprocessor, uses only the O2 A-band to provide a computationally quick...".

- Line 85: Replace "A second quick algorithm..." with "The second preprocessor performs single band retrievals of XCO2 using both the WCO2 and SCO2 bands independently".

- Line 87: Add sentence: "This often identifies scenes with aerosol contamination due to the spectral dependence of aerosol absorption."

- Line 88: Here you mention that the 3D effect is not included in the operational retrieval. Here, and/or elsewhere, I feel like it is relevant to drive this point home. And also to comment on why SHDOM cannot effectively be used as a direct solution in the retrieval due to computation cost, and difficulty in parameterizing 3D cloud effects in the state vector.

- Line 91: Here or somewhere you should mention approximate footprint size of OCO-2 and note that the effective footprint size is not fixed but increases/decreases with viewing geometry. I think Crisp's 2017 AMT paper 10.5194/amt-10-59-2017 has the best description.

- Line 101: It's a bit awkward for Fig. 6 to be the first figure referenced.

- Line 110: "OCO-2 type spectra"?...do you mean "OCO-2 like spectra"? That sounds awkward too. Same comment about "OCO-2 type retrievals".

- Lines 114-116. This is the first example of what I find to be awkward/confusing in-line mathematical notation that I mentioned in my summary comments. I'm not sure if you want to break these out as numerically labeled equations (like you have for the current Eq. (1)). At any rate, if you are using Latex, the notation could be made more elegant by using subscripts and other formatting tricks. If you are using MS Word or some other "low level" editor, then well, I guess you're stuck!

- Line 116. Recommend replacing "zero" with "0.0".

- Line 186: Mention of "3D Stokes radiation field" needs to be fleshed out with a bit more description.

- Line 195: Either mention ABSCO v5.1 here, or since it was already discussed earlier, maybe this whole sentence is unnecassary.

- Line 202-203: Regarding "windspeed specified from the Lite file data". These estimates were erroneous in ACOS L2FP v10, were they not? Aronne would know. May or may not be relevant to your work.

- Line 206: Are the assumptions for cloud and aerosol particle radii fairly robust? Is it a safe assumption that your overall results are pretty robust to reasonable perturbations of those parameters? Is there any way to

easily defend? In reality these values vary, especially by aerosol type, right?

- Line 207: insert "up" between "km" and "to".

- Line 209. Missing period at end of sentence. Also, what is this assumption based on besides convenience? Is it defensible and/or do you know if results are robust to perturbations?

- Line 215: "sulfate aerosol". Why? What is this assumption based on? For Amazon scenes I would think organic carbon would be a better aerosol type assumption, while for ocean glint it would seem sea salt would be best assumption. Furthermore, ACOS L2FP uses 2 aerosol types per soundings. Any comments? Not sure how relevant any of that is to the overall results.

- Line 216: "Stokes field". Needs ellaboration. See previous comment. May be able to get away with just citing the relevant section in the L2FP ATBD (?). Later I see this is done at Line 226! Maybe move citation earlier?

- Line 216-219: "for 17 total vertical gas optical depths". It took me some time to realize that total vertical gas optical depth (TVGOD, haha) is a proxy for the wavelength/channel dimension, where, as you say "the lowest gas optical depth is each band corresponds to the band continuum". And by extension, the highest TVGOD corresponds to deep absorption lines. Is that right? I think this is also part of my confusion with trying to interpret the results presented in Fig. 5. I suggest that this point be made a bit more clear. I just wonder how many readers familiar with OCO, but maybe not so familiar with RT theory, may get a bit confused like me, or worse. Since discussion of radiances is an integral part of this work, I also wonder if it might be instructive to generate a new 3 panel figure plotting example OCO-2 spectra for each band, with the 17 selected TVGOD points highlighted. To me that one simple figure would alleviate much of th confusion about this dimension. Again, this pertains directly to my confusion in initial interpretation of Fig. 5!

- Line 226: Would it be instructive to make detailed comparisons of observed and simulated SHDOM spectra as spectral residuals? Not sure? Or what about plots of the simulated 1D vs 3D spectra? Seems like that could be a valuable lead-in to sensitivity discussion in Sec. 3.3. Again, not sure how useful it really would be. Looking ahead to Line 250, I can visualize a plot of 1D and 3D spectra plotted against the left ordinate for an example scene, with the ratio plotted against the right ordinate (and wavelength or TVGOD plotted on the abscissa).

- Line 234: Again, a bit odd to reference Fig. 6 when we have not yet gotten to Fig. 1. But maybe okay.

- Line 265: "...the ratio decreases [slightly] as the cloud distance...then increases [drastically] as distance D..."

- Line 267: Might be helpful to indicate the value that the ratio actually increases to.

- Line 302 and 305-306: Another example of somewhat unwieldy in-line mathematical notation.

- Line 325: where does the "2 to 4%" come from? I dont see anything in Fig. 5 that matches this description. I see the 3D curve ranging from 0 to 4% in the A-band as a function of TVGOD, 1-3% in the WCO2 band, and 0 to 2% in the SCO2 band. I must be misinterpreting the figure. See previous comments.

- Line 337: The sentence "Fig. 5 illustrates the zero-order physics..." seems like it would be better placed near the opening of the section. Again, I feel like Fig. 5 either needs to be reformed somehow to be more intuitive, or at least the orientation of it in the text needs to be more upfront. Walk the reader through the interpretation of a single variable for instance, and then the reader can extrapolate that knowledge to the other variables and spectral bands.

- Line 343-346: The last sentence in this section is very awkward. I think I know what you are saying, but recommend that it be reworded to be more concise and clear.

- Line 350: "detailed SHDOM calculation". I'm pretty sure that this is a typo and should be MODIS radiance fields, as is correctly given in the figure caption.

- Line 353: "atmosphere [used in simulations] is derived from the..."

- Line 367: typo: change "point that those" to "points than those"

- Lines 379-420: As mentioned in my summary, I feel that this discussion is overly long in context of it's importance to the overall results and relative to the overall length of the paper. I recommend either moving most of the details to a technical appendix (if you feel like it is important to preserve the details for someone to reproduce) or just shorten the discussion to get to the main point that cloud brightening events heavily dominant

cloud shadowing events. It's an important point, but probably does not need to take up so much space and kind of derail the reader.

- Line 419-420: what's the split between QF0/QF1 for the cloud darkening scenes? Similar to or different than the cloud brightening scenes?

- Line 423: change "data points" to "soundings".

- Line 435: typo? Should "Fig. 6" be "Fig 7"?

- Line 439: recommend replacing "can reach" with " are significant".

- Line 445: Table 3 - is it necessary? Doesn't add much useful information to me.

- Line 451: "presents [results for] individual...".

- Line 453: "...for the smallest gas optical depth) [versus the nearest cloud distance taken from the 3D metric file...".

- Line 458: "In the [operational OCO-2] bias correction...".

- Line 459: Instead of citing O'Dell, AMT, 2018 for the v8 data, it would be better to cite here the OCO-2 v10 DUG which contains the tables of variables and thresholds used for filtering and bias correction.

- Line 470 area: need to refer to the lettering labels for individual subplots, e.g., "panel (a) of Fig. x...". See comment in Summary section.

- Line 503: typo? Remove "a" after "Table 1".

- Line 506: The phrase "are amply sufficient to add sufficient 3D radiances to bring forward model and observed radiances into agreement" is super awkward. Recommend rewording.

- Line 508-509: It's likely that the ACOS v10 land/ocean dp terms have different thresholds. You could check the DUG tables.

- Line 510: "Figure 12 displays [the Amazon] land glint data."

- Line 528: " approximately 4km [on average] from clouds..."

- Line 536: Awkward to first discuss panels (e) and (f) of the Fig. 14. Either rearrange the figure to put ocean glint panels at the top row, or rearrange the text.

- Line 541: "displayed in Fig. 15 [for year 2016]".

- Line 553: "by approximately [2.2] ppm per year"

- Line 556: "as a function of nearest cloud distance". I don't see this metric in Fig 16?

- Line 557: "raw XCO2" Typo? Should that be "dp"?

- Line 555-562: What is the conclusion from Fig. 16? Is it just as it relates to the analysis of the results presented in Fig. 17? (line 574).

- Line 564: "as a function of nearest cloud distance". Again, I don't understand how that metric appears in this figure. It seems like you mean that the rawXCO2 is a function of nearest cloud distance, so maybe the phrase is just misplaced.

- "toss out" should be "identify".

- Line 579: Recommend "between clouds" should perhaps be "in the vacinity of clouds".

- Line 586: Recommend "But cloud..." should be "However, cloud..."

- Line 592: Recommend putting "ratios greater than unity" and "ratios less than unity" in (parenthesis).

- Line 594-603: All of this detailed discussion would be better in Sec 5 rather than the Conclusions.

- Line 610-617: All of this detail might be better in the analysis sections.

- Line 623-631: Most of this paragraph belongs in analysis section. Or reword each of these multi-sentence paragraphs down into a single conclusive sentence.

- Line 632-640: This paragraph should be earlier in the Conclusions. This is good "high level" discussion that sets the tone for why the work is important. But I would avoid citation and analysis of individual Figures in the conclusions, but that maybe personal taste.

- Line 641-647: This paragraph also should come at the outset of the Conclusions, but in a general way without referring directly to the figures.

- Table 1: I cannot tell from the discussion if the simulations were a single iteration of each state parameter by the perturbation amount listed in the table. Or were there multiple iterations of the perturbations to state vector elements? This point needs to be clarified here and in the main text.

- Table 2: I dont love using footnotes in the table. Might be better to describe in the caption? Personal taste?

- Table 3: Is this table actually helpful in anyway? I feel like it is fairly irrelevant to the analysis. Not sure.

- Table 4: For sure the footnote should just be in the table caption.

- Figure 2: abscissa label should include "[km]". My interpretation of this figure is that as the observation is made closer to the cloud, the 1D radiances (as modeled by SHDOM) become an increasingly smaller fraction of the SHDOM 3D radiances. This implies that the L2FP modeled 1D radiances would underestimate the true measured radiances if all of the retrieval physics were included and setup correctly. But since that is not the case, the L2FP must modify ancillary parameters such as surface albedo and aerosol in order for the modeled radiances to match the measurements. Presumably, a brightening of the surface, coupled with a lofting and/or loading of the aerosol would be required. If that interpretation is correct, it might be helpful to state that more explicitly in Sec. 3.2 when Fig. 2 is discussed (or elsewhere where relevant). This might help readers with the physical interpretation of how the L2FP retrieval is sensitive to the 3D cloud effect.

- Figure 5: I already have made a number of comments about interpretation of this figure above. A couple of more technical comments here: the contrast between the white background and the yellow Refl curve is very poor. Recommend changing to another color. Is there any concern about dependence among the state variables as each seems to have been perturbed individually? I still dont understand if this figure reflects a single perturbation of each state vector element, or if multiple perturbations are somehow wrapped in this figure. If the former, then it seems like a logical question to ask if the results are robust to further perturbations. If the latter, then I'm afraid that I'm lost.

- Figure 6: caption says MODIS radiance, while text discussion says SHDOM. I think the latter is in error.

- Figures 6+7 and 8+9: The utility of these figures would be increased if they were combined into two two panel plots (or even a single 4 panel plot?). Figs 7+9 would be more useful if they displayed the exact same X-Y range, presumably in lat/lon space. This would make them much more comparable. Plus add the OCO individual sounding points (the Xs and Os) to Figs 7+9.

- Figures 10-15: All of these figures would benefit from subpanel lettering, e.g., (a), (b), (c), which could be directly referenced in the text (rather than using awkward phrases like "the middle right panel in figure X" for example. In fact, I'm pretty sure that subpanel lettering is a requirement for AMT.

- Figures 10-12: These graphs take up a lot of space, and what are they showing besides that the operational QF procedure works?

- Figures 13-15: Might be good to utilize color to aid in interpretation. Not strictly necessary.

- Figure 16-17: The yellow curve does not show up well against the white background.

---

## Author Comment (AC1)

**Author's Response: Insights into 3D cloud radiative transfer for OCO-2**

S. Massie, H. Cronk, A. Merrelli, S. Schmidt, S.
Mauceri January 13, 2023

We appreciate the extensive suggestions of the reviewer (RC1). The revised paper incorporates the vast majority of the suggestions of the reviewer. We have put the reviewer's original comments in italics. New line numbers in the revised paper (e.g. N142) indicates the line number where revisions are located.

**1   Review summary**

*This paper provides an evaluation of the effects of three dimensional (3D) radiative transfer (RT) on the spectra measured by NASA's Orbiting Carbon Observatory -2 (OCO-2) due to clouds within close proximity (several to tens of kilometers) of the ground footprints. Clouds impose a 3D effect that increases as the distance to the nearest cloud decreases. The 3D effect is currently unaccounted for in the full physics retrieval algorithm that is used to derive estimates of total column carbon dioxide (XCO2) from the measured radiances. Using simulations of 1D and 3D radiances (from the SHDOM RT model) for realistic OCO-2 scenes, the authors demonstrate that 3D perturbations to the radiances are of the same order (or greater) than reasonable perturbations to other key retrieval state vector elements, such as surface pressure and aerosol loading. The effect is shown to be significant relative to the precision requirements of the mission on the order of 1 part per million (ppm). It is therefore important to characterize and quantify the 3D effect in order to continue improving the XCO2 estimates in future versions of the data products. **Overall, the paper is well organized and written in fluent English, with appropriate figures to describe the results of the analysis. However, due to a couple of significant concerns, I have to recommend that major revisions be performed.** My primary constructive comments are itemized below, approximately in order of importance.*

- ***It seems to me that the current work is very similar in nature to the work presented in [Massie, AMT, 2021].** The only clear difference I can determine is that the current work utilizes "bar cloud" calculations, which were not present in the 2021 manuscript. The authors need to make it explicitly clear how the implementation of bar clouds in the current research leads to either substantially different results compared to [Massie, AMT, 2021], or corroborates the previous results. Otherwise, I do not really understand the purpose of the current publication. Is it fair to say that the current research refines the 2021 research, and demonstrates the 3D effect using more first principle physics (or something like that)? In general, the previous publication did clear up some of my questions related to the current research. For example, the description of Fig. 2 in [Massie, AMT, 2021] is more clear than the description of Fig. 5 in the current research. In effect, the figures are the same.*

The revised paper includes the following paragraph (N142) in the Introduction to clearly state why the current paper differs substantially from the previous paper:

"This paper is follow-on work of Massie et al. (2021) and expands the previous research in significant ways. Massie et al. (2021) focused on quantifying 3D cloud effects based on comparisons of bcXCO2 and TCCON data. The geographical distribution of TCCON sites, however, is concentrated over north America and Europe, with sparse coverage over the tropics and over the ocean. The current paper calculates 3D cloud biases as a function of latitude using the Zenodo "3D metric" files for 275 times more data points. *The latitudinal dependence of 3D cloud biases is not addressed in Massie et al. (2021), but addressed in this paper.* Massie et al. (2021) demonstrated that 3D cloud biases frequently indicate an underestimation of XCO2 as nearest

cloud distance decreases. The current paper provides a physical reason why this is the case. Massie et al. (2021) examined one 32 km x 32 km scene, while this paper examines 36 scenes over the ocean and land for a variety of viewing geometries over the Amazon and the oceans, areas which are problematic for OCO-2. The current paper presents side by side graphs of 1D / 3D intensity ratios (and other variables) as a function of nearest cloud distance to illustrate the non-linearities that are present in the OCO-2 data files. The non-linearities that are present in our graphs for several variables are non-linearities which Machine Learning (ML) bias correction methods (see Mauceri, Massie, and Schmidt 2022) will need to mitigate."

Fig. 2 of Massie AMT 2021 had a curve with a scaling error, so the current paper's Fig. 5 corrects this issue.

> ***The overall conclusions drawn from the analysis are not clear or strong enough.*** *I'd prefer the conclusions section to be more succinct and void of the analysis details (which should appear in Sections 3 through 6). The authors effectively demonstrate the high frequency of occurrence of the cloud 3D effect on the OCO-2 measurements (40% of all OCO-2 measurements are within 4 km of a cloud!). They effectively demonstrate that the average 3D perturbations will cause errors in the XCO2 estimates on the order of 0.5 ppm, which is highly significant to the scientific goals of the OCO-2 project. However, they also effectively demonstrate, by analysis of both good (QF=0) and bad (QF=1) data, that the current operational OCO-2 quality filtering routinely identifies the scenes with larger 3D contamination, which would thereby have larger XCO2 errors.* ***The main criticism is that no real conclusion or recommendation is drawn about the potential for cor- recting or improving future versions of the OCO-2 XCO2 product based on the results presented in this analysis.*** *This work seems to be an important stepping stone towards a robust solution, but that thought is never expressed.*

The revised paper now has an added Sect. 7. This section (N659) discusses how one can calculate an empirical look-up table of 3D cloud effect biases, and use it to correct bcXCO2 for 3D cloud effects that are present in the V10 Lite files. New graphs in Figs. 16, 17, 18 illustrate how the empirical look-up table is calculated, curves of the look-up Table, and an application of the look-up table to calculate observation number weighted 3D cloud biases for bcXCO2 as a function of latitude. The Abstract has been modified and a new Zenodo archive of the 3D cloud effect biases was created.

> *Ideally, in the future, some methodology could be put into place to "save" the "bad" quality flagged (QF=1) data, or to minimize the size of the dp bias correction term necessary to correct the raw XCO2, but the authors never really say that directly.*

We are of the opinion that the development of machine-learning techniques will eventually do so and be adopted by the OCO-2 Science team. The added paragraph in the Introduction refers (N155) to the machine learning paper of Mauceri, Massie, and Schmidt (2023) which has been accepted for AMT publication, and is s part of our funded NASA project.

> *As it currently stands, one could make the argument that the operational QF/BC is already doing a pretty good job of accounting for the 3D cloud effect, so why should additional effort/resources be put into mitigating it using (closer to) first principles physics? I recommend that the authors address this issue, at least in theory, and discuss proposed future work.*

The revised text in the last paragraph of the Conclusions section (N778) now states "Future work includes the development of a quick parameterization of 1D / 3D ratios as a function of aerosol and cloud optical depth, given an arbitrary geospatial distribution of clouds."

Massie, AMT, 2021 demonstrated that the cloud bias of retrieved XCO2 was on the order of

0.5 ppm for QF0 data. The reviewer acknowledges that this is an important issue, so "pretty good" is not good enough. As recommended by the reviewer, the Chevallier GRL 2010 is added to the revised paper. Chevallier asserts that biases in XCO2 on regional scales as small as a few tenths of a part-per-million (ppm) in XCO2 can lead to spurious values of inferred $CO_2$ fluxes.

*Furthermore, I recommend reducing the verbiage in the Conclusions section itself. Many of the sentences there are very detailed and seem misplaced, as if they should belong in the technical discussion in the main body of the text. I have a number of specific recommendations in the technical corrections below.*

Three paragraphs have been moved from the Conclusions to the analysis sections in the revised paper. Quite a few sentences have been removed in the other paragraphs of the Conclusions.

- ***Any new conclusive thoughts should also be amplified in the Abstract to bring the message home quickly to the casual reader.*** *I feel like in general, the abstract is a bit too detailed and could use more focus on the "big picture" of why the work is important, how/why it is a difficult problem to address, and what the possible solutions might be.*

The revised Abstract now specifies the results calculated and discussed in the new Sect. 7 of the paper, in regard to northern and southern bcXCO2 3D cloud effect biases.

- ***Interpretation of Fig 5. and the discussion of the results in Sec. 3.3 (radiance perturbation sensitivity) are of paramount importance to this work, but unfortunately, as currently written they are quite confusing.*** *I wonder if both the figure and the discussion could be reorganized, or at least better explained, to aid the reader. After a while I think I determined that the x-axis is the 17 values of the Total Vertical Gas Optical Depth (TVGOD) that is described on Line 217.*

The revised paper now explains (N365) the x axis variable in both the text and figure caption.

- *That point needs to be made explicit, as I got stuck thinking that these were iterations of aerosol optical depth in the retrieval for sensitivity testing. But when I read the analysis of the results in Sec 3.3 I cant really follow along in the figure. For example: Lines 312-316 suggest that there will be an increase in radiance if surface reflectance or AOD are increased. What am I supposed to see in the figure that suggest that, because I cannot see it. Are the results in Fig. 5 due to a single perturbation of each variable? If so, then are the results robust to further perturbations? If the figure already somehow contains results for multiple perturbations of state vector values, then I dont understand how that is displayed. That to me would suggest that the TVGOD on the x-axis is not in fact a proxy for channel/wavelength. My impression is that the graph is meant to demonstrate that the 3D effect is much larger than the effect due to perturbing the state variables, seen by the fact that the 3D curve is far away from zero for most TVGOD, while most of the other variables are close to zero at all TVGOD. The exception being Refl.*

The revised paper explicitly states (N375) that there are no iterations (the partial derivatives are for a fixed model atmosphere). The model atmosphere is specified using OCO-2 and MODIS data, and once specified, is fixed in the SHDOM calculation.

- *Some discussion should be given (might be very brief) as to why the 3D effect cannot be accounted for in the L2FP retrieval simply by embedding the SHDOM RT model to calculate it directly due to the massive computation cost, plus difficulty in parameterizing 3D cloud effects. I'm not sure that this is ever mentioned explicitly (?).*

Sebastian Schmidt is developing the EaR3T 3D model which can be run on multiple processors (he has done this at the University of Colorado). This is work in progress. This development is for Sebastian to report on, and lies outside of the scope of the paper.

The revised paper now states (N101) "Until the wall-clock advantages of parallel computing can be implemented in an operational setting, 3D cloud effects will remain computationally expensive in an operational setting."

The revised paper (N778) states in the Conclusions section that "Future work includes the development of a quick parameterization of 1D / 3D ratios as a function of aerosol and cloud optical depth, given an arbitrary geospatial distribution of clouds."

- *In Sec. 4 (Amazon and Ocean Glint scenes), a large number of lines (about 40, 6% of total paper) are used to describe the details of the calculation for determining the fraction of cloud shadowing scenes compared to the fraction of cloud brightening. It felt like a bit of a distraction since the conclusion is that cloud shadowing only occurs for on the order of 1-4% of the converged retrievals. I wonder if the details of the cloud shadowing determination would be better moved to a technical appendix. Or, perhaps the authors could work to shorten the description if they prefer to leave it in the main body of the document.*

The algorithm discussion is now moved to Appendix A (N789).

- *In general, throughout the text, I found the mathematical notation unwieldy. I would recommend making improvements to aid the readers ability to interpret. Specific examples are given below in the Technical er- rors/corrections section.*

The revised paper now uses subscripts for the mathematical notation. Looks much better.

- *For all of the results presented in the various sections, it seems like a number of important assumptions were made. I have a feeling that these were all well thought out, but there is no indication as to the robustness of the results for different sets of assumptions. Specific examples are given below in the Technical errors/corrections section.*

The paper is constructed to provide *insights*, and presents *representative* calculations. The revised paper now includes curves in new Fig. 4 with sensitivities in regard to cloud LWC (Fig. 4e) and aerosol composition and size distribution (Fig. 4f), as requested by the reviewer.

*The scope of the research is limited to the Amazon for land observations, as well as ocean observations. I understand the interest and the fact that the cloud morphology is such that the 3D effect is highly relevant. But have other regions been examined? Are the results robust? I'm thinking that regions with brighter surfaces will likely have additional complications of multiple scattering effects, perhaps yielding different 3D contributions (potentially even in sign)? Please state clearly what you have and haven't studied in this work, and that only Amazon forest oceans have been examined, and potentially also mention this as future work, as characterizing 3D effects over different surfaces is an important follow-on.*

The revised text (last paragraph of the Conclusions section) now states (N778) "Future work includes the development of a quick parameterization of 1D / 3D ratios as a function of aerosol and cloud optical depth, given an arbitrary geospatial distribution of clouds. This work will examine a wider range of parameters such as cloud height, aerosol height, aerosol composition, in addition to an examination of scenes not covered in this paper, such as brighter surfaces."

- *Most of the figures would benefit from the use of sub panel lettering, e.g., (a), (b), (c).*

Figures in the revised paper have (a), (b), (c) labeling.

- *The title and all of the analysis/discussion is focused on OCO-2. But the general principles are applicable to OCO-3. Might we worth mentioning in some form or fashion to broaden the utility. (I'm aware that the "distance to nearest cloud" data does not exist for OCO-3, making actual analysis difficult/impossible.)*

Yes, the results in this paper applies to OCO-3. The last paragraph of the Conclusions now states (N784) that "While our paper focuses on OCO-2, the results likely are applicable to OCO-3 on the International Space Station. Geosynchronous satellite data can be used to derive nearest cloud distance for studies related to OCO-3 and 3D cloud effects."

- *I wonder if there is a tie-in to the results presented in Emily Bell's swath-bias paper (https://doi.org/10.5194/amt- 16-109-2023)? Although that work was specific to biases in XCO2*

*for OCO-3 Snapshot Area Maps, statements, such as;*

*"These simulation studies reveal that SB is primarily and intimately connected to the presence of aerosols and the interplay of their optical properties with the solar and instrument viewing geometries. We now have a better understanding of the types of scenes that are likely to suffer from SB – those with high aerosol depths, or aerosols that are lofted higher in the atmospheric column, and in geometries with broader ranges of obser- vation and solar zenith angles. Future work may involve a more detailed study of how the physics of aerosol optical properties with viewing and solar geometries combine to produce an SB response."*

*seem highly relevant to your cloud 3D work. Just something to consider.*

The last paragraph of the Conclusions section now states (N782) "Aerosol characteristics are of interest since Bell et al. (2023) used OCO-3 data to demonstrate that "swath bias" is related to aerosol characteristics and viewing geometry."

**2   Technical errors/corrections**

- *Title: might be useful to insert the word "effects" between "transfer" and "for".*

We have done so in the revised paper.

- *Abstract: Line 33 provides summary result for ocean glint, but does not mention land results.*

Land and ocean results are now both mentioned (N34) in the revised abstract.

- *Introduction: Line 39 mentions XCO2 measurement accuracy of 0.25% (1 ppm). Note that although this is the formal project requirement, I think the science is suggesting that biases on the order of 1 ppm would actually be much to large for source/sink inversion work, e.g., [Chevallier, GRL, 2014, Toward robust and consistent regional CO2 flux estimates from in situ and spaceborne measurements of atmospheric CO2]*

We have revised the text (N43)  to include the Chevallier GRL 2010 assertion that biases in XCO2 on regional scales as small as a few tenths of a part-per-million (ppm) in XCO2 can lead to spurious values of inferred $CO_2$ fluxes.

- Line 44: "The OCO-2 experiment" could/should be "The OCO-2 satellite".

We have done so in the revised paper.

- *Introduction: would be good to add a sentence or phrase in the satellite description (near line 50) that mentions sun-sychronous polar orbit moving from SE to NW. That seems relevant to some of the figures and discussions of the sun/sensor geometries.*

We have done so in the revised paper.

- *Line 51: mention ACOS version (v10).*

We have done so in the revised paper (N54).

- *Line 52: This sentence uses "XCO2" where I think "CO2" is the appropriate term. XCO2 is a calculated quantity and is not directly measured in the WCO2 and SCO2 as suggested. And I'm not sure that XCO2 was defined?*

Line 52 has been revised. XCO2 is defined in the first line of the Introduction (N39).

- *Line 53: someone recently informed me that the Rodgers 2000 citation should be replaced with Rodgers 2004, which was a reprint that corrected a lot of typos (small thing). @bookRodgers:2004:InverseMethods, address*
= *5 Toh Tuck Link, Singapore 596224, author = Clive D. Rodgers, edition = Reprint, isbn = 981022740X, isbn-13*
= *9789810227401, keywords = Atmospheric Science, pages = 238, publisher = World Scientific Publishing Co. Pte. Ltd., title = Inverse Methods for Atmospheric Sounding: Theory and Practice, year = 2004*

We have done so in the revised paper.

- *Line 56: mention which version of the ABSCO files (v5.1) are used in the ACOS v10 retrieval.*

We have done so in the revised paper.

- *Lines 56-58: Seems like the sentence describing the specific advances in ABSCO v5.1 is unnecessary. Is this information used anywhere else in the paper directly?*

We deleted this sentence in the revised paper.

- *Line 62: check spelling for TCCON Wunch citation!*

The revised paper corrects this typo.

- *Line 69: Equation (1) seems unnecessary. Is this information used anywhere else in the paper directly?*

The equation is useful to those outside of the OCO-2 Science Team since it succinctly defines the dPfrac variable. The equation is the quickest way to define dPfrac.

- *Lines 64-72 describes in some detail the various individual bias correction parameters. I'm not sure if this is really needed? For the most part these terms are not used again, with the definite exception of dp. I would recommend either remove the descriptions if they are not needed, or alternatively, discuss somewhere in the paper how they may relate to the 3D cloud effect.*

Members of the OCO-2 Science Team know this information, but those outside of the OCO-2 Science Team do not. We consider these sentences are important to convey minimal information to the general reader.

- *Line 72: The description of the bias correction increasing XCO2 by 2 ppm is misleading. It sounds like just a single global offset. There are a number of parameters (you described just above), a footprint correction term and global land/ocean values. You would probably be best off to just point to the details in the OCO-2 v10 Data User's Guide, which I dont think you currently cite. Alternatively, as mentioned above, you could flesh out more in the results discussion how the 3D effect may (or may not) influence the necessary bias correction and whether mitigation of the 3D effect could in theory mitigate the bias correction. That could be interesting since great effort is spent to achieve small improvements in the successive versions of the XCO2 product.*

The revised text adds a sentence (N74) which informs the reader to consult the details discussed in the Data User's Guide.

- *Line 74: Could be useful to cite some of the UQ publications at the end of this discussion, e.g., Connor, AMT, 2016, 10.5194/amt-9-5227-2016 and/or Hobbs, JUQ, 2017, 10.1137/16M1060765.*

The revised paper now (N78) includes reference to these uncertainty quantification papers.

- *Line 81: "Of the [approximately one] million...collected by OCO-2, [only] about 25%...passed into the operational retrieval [due to prescreening for scenes contaminated by clouds and heavy aerosol loadings]."*

The sentence was revised as per the reviewer's suggestion.

- *Line 83: replace "radiances" with "soundings"*

The sentence is revised (N88) as per the revewier's suggestion.

- *Line 83: The sentence "A quick determination..." should be preceded with "One preprocessor, uses only the O2 A-band to provide a computationally quick...".*

The text has been revised as per the reviewer's suggestion.

- *Line 85: Replace "A second quick algorithm..." with "The second preprocessor performs single band retrievals of XCO2 using both the WCO2 and SCO2 bands independently".*

The text has been revised as per the reviewer's suggestion.

- *Line 87: Add sentence: "This often identifies scenes with aerosol contamination due to the spectral dependence of aerosol absorption."*

We have done so in the revised paper.

- *Line 88: Here you mention that the 3D effect is not included in the operational retrieval. Here, and/or else- where, I feel like it is relevant to drive this point home. And also to comment on why SHDOM cannot effectively be used as a direct solution in the retrieval due to computation cost, and difficulty in parameterizing 3D cloud effects in the state vector.*

A sentence has been added to the text (N101) which states that "Until the wall-clock advantages of parallel computing can be implemented in an operational setting, 3D cloud effects will remain computationally expensive in an operational setting."

- *Line 91: Here or somewhere you should mention approximate footprint size of OCO-2 and note that the effective footprint size is not fixed but increases/decreases with viewing geometry. I think Crisp's 2017 AMT paper 10.5194/amt-10-59-2017 has the best description.*

The revised text now states this fact.

- *Line 101: It's a bit awkward for Fig. 6 to be the first figure referenced.*

Yes, but having Fig. 6 as the first figure leads to much more awkwardness. The mentioning of Fig. 6 (older line 101) is helpful to the reader.

- *Line 110: "OCO-2 type spectra"? ...do you mean "OCO-2 like spectra"? That sounds awkward too. Same comment about "OCO-2 type retrievals".*

"Type" has been replaced by "like" in the revised text.

- *Lines 114-116. This is the first example of what I find to be awkward/confusing in-line mathematical notation that I mentioned in my summary comments. I'm not sure if you want to break these out as numerically labeled equations (like you have for the current Eq. (1)). At any rate, if you are using Latex, the notation could be made more elegant by using subscripts and other formatting tricks. If you are using MS Word or some other "low level" editor, then well, I guess you're stuck!*

I use MS word, and revised the text using subscripts.

- *Line 116. Recommend replacing "zero" with "0.0".*

Zero is replaced by 0.0 in the revised sentence.

- *Line 186: Mention of "3D Stokes radiation field" needs to be fleshed out with a bit more description.*

A sentence has been added to the revised text (N217). Since we only focus on the total intensity of the Stokes field, a detailed discussion of the Stokes field is not necessary.

- *Line 195: Either mention ABSCO v5.1 here, or since it was already discussed earlier, maybe this whole sentence is unecessary.*

The ABSCO v5.1 version number is now mentioned here.

- *Line 202-203: Regarding "windspeed specified from the Lite file data". These estimates were erroneous in ACOS L2FP v10, were they not? Aronne would know. May or may not be relevant to your work.*

Since the 1D /3D ratios for the continuum do not have a large dependence on surface reflectance (Fig. 4d), errors in Lite file windspeed should have a minor influence in our paper.

*Line 206: Are the assumptions for cloud and aerosol particle radii fairly robust? Is it a safe assumption that your overall results are pretty robust to reasonable perturbations of those parameters? Is there any way to easily defend? In reality these values vary, especially by aerosol type, right?*

The revised paper now includes sensitivity calculations for cloud LWC (Fig. 4e) and aerosol composition and size distributions (Fig. 4f).

- *Line 207: insert "up" between "km" and "to".*

We have done so in the revised text.

- *Line 209. Missing period at end of sentence. Also, what is this assumption based on besides convenience? Is it defensible and/or do you know if results are robust to perturbations?*

Added the period. The 1.8 km aerosol height is representative. The scope of the paper is to gain *insights from representative calculations*, and not to report on an exhaustive set of calculations with numerous perturbations. The last paragraph of the Conclusions refers to future calculations.

- *Line 215: "sulfate aerosol". Why? What is this assumption based on? For Amazon scenes I would think organic carbon would be a better aerosol type assumption, while for ocean glint it would seem sea salt would be best assumption. Furthermore, ACOS L2FP uses 2 aerosol types per soundings. Any comments? Not sure how relevant any of that is to the overall results.*

The revised Fig. 4 now includes a panel with sensitivity calculations (Fig. 4f) on aerosol composition and size distributions. The last paragraph of the Conclusions now states that future work will consider a wider range of aerosol characteristics.

- *Line 216: "Stokes field". Needs ellaboration. See previous comment. May be able to get away with just citing the relevant section in the L2FP ATBD (?). Later I see this is done at Line 226! Maybe move citation earlier?*

Our paper focuses on a discussion of the total intensity component of the Stokes field, and not an examination of the Stokes parameters which can be used to specify the degree of polarization of linear and circular polarization. The text emphasizes this point in the last paragraph of section 3.1

- *Line 216-219: "for 17 total vertical gas optical depths". It took me some time to realize that total vertical gas optical depth (TVGOD, haha) is a proxy for the wavelength/channel dimension, where, as you say "the lowest gas optical depth is each band corresponds to the band continuum". And by extension, the highest TVGOD corresponds to deep absorption lines. Is that right? I think this is also part of my confusion with trying to interpret the results presented in Fig. 5. I suggest that this point be made a bit more clear. I just wonder how many readers familiar with OCO, but maybe not so familiar with RT theory, may get a bit confused like me, or worse. Since discussion of radiances is an integral part of this work, I also wonder if it might be instructive to generate a new 3 panel figure plotting example OCO-2 spectra for each band, with the 17 selected TVGOD points highlighted. To me that one simple figure would alleviate much of th confusion about this dimension. Again, this pertains directly to my confusion in initial interpretation of Fig. 5!*

Yes, the lowest optical depth (the x axis of Fig. 5) corresponds to the continuum, while the highest optical depths correspond to deep absorption lines. The revised text and revised Fig. 5 caption now states this as soon as possible in the revised paper (N365).

- *Line 226: Would it be instructive to make detailed comparisons of observed and simulated SHDOM spectra as spectral residuals? Not sure? Or what about plots of the simulated 1D vs 3D spectra? Seems like that could be a valuable lead-in to sensitivity discussion in Sec. 3.3. Again, not sure how useful it really would be. Looking ahead to Line 250, I can visualize a plot of 1D and 3D spectra plotted against the left ordinate for an example scene, with the ratio plotted against the right ordinate (and wavelength or TVGOD plotted on the abscissa).*

The paper focuses on 1D / 3D ratios in the continuum portions of the OCO-2 spectra. The last paragraph of section 3.1 emphasizes that the scope of the paper does not address a comparison of observed and simulated SHDOM spectra residuals.

- *Line 234: Again, a bit odd to reference Fig. 6 when we have not yet gotten to Fig. 1. But maybe okay.*

Having Fig. 6 as the first figure leads to much more awkwardness.

- *Line 265: "...the ratio decreases [slightly] as the cloud distance...then increases [drastically] as distance D..."*

We revised the text as per the suggestion.

- *Line 267: Might be helpful to indicate the value that the ratio actually increases to.*

The revised text (N312) now mentions that the ratio increases towards a value of 5.4.

- *Line 302 and 305-306: Another example of somewhat unwieldy in-line mathematical notation.*

The revised paper now uses suggested sub-scripting.

- *Line 325: where does the "2 to 4%" come from? I dont see anything in Fig. 5 that matches this description. I see the 3D curve ranging from 0 to 4% in the A-band as a function of TVGOD, 1-3% in the WCO2 band, and 0 to 2% in the SCO2 band. I must be misinterpreting the figure. See previous comments.*

The revised text now clarifies these statements. The Fig. 5 curves are different from the original paper curves since a more appropriate SHDOM azimuth angle was selected in the figure revision.

- *Line 337: The sentence "Fig. 5 illustrates the zero-order physics…" seems like it would be better placed near the opening of the section. Again, I feel like Fig. 5 either needs to be reformed somehow to be more intuitive, or at least the orientation of it in the text needs to be more upfront. Walk the reader through the interpretation of a single variable for instance, and then the reader can extrapolate that knowledge to the other variables and spectral bands.*

In the revised text (N364) the interpretation (clarification) of the curves is now present in the first paragraph of section 3.3, with two sentences added to the paragraph.

- *Line 343-346: The last sentence in this section is very awkward. I think I know what you are saying, but recommend that it be reworded to be more concise and clear.*

The text has been revised (N343) to be clearer.

- *Line 350: "detailed SHDOM calculation". I'm pretty sure that this is a typo and should be MODIS radiance fields, as is correctily given in the figure caption.*

The typo is corrected in the revised text.

- *Line 353: "atmosphere [used in simulations] is derived from the…"*

The sentence is revised as per the suggestion.

- *Line 367: typo: change "point that those" to "points than those"*

The sentence is revised as per the suggestion.

*Lines 379-420: As mentioned in my summary, I feel that this discussion is overly long in context of it's impor- tance to the overall results and relative to the overall length of the paper. I recommend either moving most of the details to a technical appendix (if you feel like it is important to preserve the details for someone to repro- duce) or just shorten the discussion to get to the main point that cloud brightening events heavily dominant cloud shadowing events. It's an important point, but probably does not need to take up so much space and kind of derail the reader.*

The paragraphs that discuss the algorithm are now placed in Appendix A (N789).

- *Line 419-420: what's the split between QF0/QF1 for the cloud darkening scenes? Similar to or different than the cloud brightening scenes?*

The revised text now states (N463) "The retrievals associated with cloud shadows are QF1 data points, while Table 2 indicates that retrievals associated with cloud brightening have QF0 percentages between 34% and 60% for the various cases."

- *Line 423: change "data points" to "soundings".*

Revised as suggested.

- *Line 435: typo? Should "Fig. 6" be "Fig 7"?*

Typo is corrected.

- *Line 439: recommend replacing "can reach" with " are significant".*

Fig. 9 has been revised to Fig. 7 (to present the 1D / 3D field for the most appropriate SHDOM azimuth angle), and the paragraph differs from the original paper, so the recommended rewording is irrelevant.

- *Line 445: Table 3 - is it necessary? Doesn't add much useful information to me.*

Yes, Table 3 is necessary. If interested, a reader can use NASA Worldview and the information in Table 3 to visually examine the cloud field of a particular land or ocean scene.

- *Line 451: "presents [results for] individual…".*

Revised as suggested.

- *Line 453: "…for the smallest gas optical depth) [versus the nearest cloud distance taken from the 3D metric file…".*

Revised as suggested.

- *Line 458: "In the [operational OCO-2] bias correction…".*

Revised as suggested.

- *Line 459: Instead of citing O'Dell, AMT, 2018 for the v8 data, it would be better to cite here the OCO-2 v10 DUG which contains the tables of variables and thresholds used for filtering and bias correction.*

Revised as suggested.

- *Line 470 area: need to refer to the lettering labels for individual subplots, e.g., "panel (a) of Fig. x…". See comment in Summary section.*

Figures now have (a), (b), (c) labels.

- *Line 503: typo? Remove "a" after "Table 1".*

The typo has been corrected in the revised text.

- *Line 506: The phrase "are amply sufficient to add sufficient 3D radiances to bring forward model and observed radiances into agreement" is super awkward. Recommend rewording.*

The sentence has been corrected in the revised text.

- *Line 508-509: It's likely that the ACOS v10 land/ocean dp terms have different thresholds. You could check the DUG tables.*

Table 3-4 of the suggested DUG does not specify the filter dP or dPfrac range for QF0 ocean glint data. The sentence is valid since it comments on the behavior of the data points, irrespective of the filter thresholds.

- *Line 510: "Figure 12 displays [the Amazon] land glint data."*

The phrase is revised as suggested.

- *Line 528: " approximately 4km [on average] from clouds…"*

The phrase is revised as suggested.

- *Line 536: Awkward to first discuss panels (e) and (f) of the Fig. 14. Either rearrange the figure to put ocean glint panels at the top row, or rearrange the text.*

The Figure is rearranged in the revised paper.

- *Line 541: "displayed in Fig. 15 [for year 2016]".*

The sentence is revised as suggested.

- *Line 553: "by approximately [2.2] ppm per year"*

Revised as suggested.

- *Line 556: "as a function of nearest cloud distance". I don't see this metric in Fig 16?*

The typo is corrected in the revised text.

- *Line 557: "raw XCO2" Typo? Should that be "dp"?*

The typo is corrected.

- *Line 555-562: What is the conclusion from Fig. 16? Is it just as it relates to the analysis of the results presented in Fig. 17? (line 574).*

It relates to the results presented in Fig. 17 (in the revised paper, Fig. 15).

- *Line 564: "as a function of nearest cloud distance". Again, I don't understand how that metric appears in this figure. It seems like you mean that the rawXCO2 is a function of nearest cloud distance, so maybe the phrase is just misplaced.*

The confusing phrase ( a typo) is deleted in the revised text (N642).

- *"toss out" should be "identify".*

"toss out" is replaced by "identify"

- *Line 579: Recommend "between clouds" should perhaps be "in the vacinity of clouds".*

The "between clouds" phrase is important, since as shown in Figs. 6 and 8 (in revised paper, Fig. 6 and 7), successful retrievals are between clouds, not close to clouds.

Revised the sentence to (N740): "The OCO-2 cloud preprocessor does a very good job in screening observations near and over clouds. The preprocessor, however, does not necessarily identify clear sky observations, impacted by 3D cloud effects, that are located *between* low-altitude clouds. Figs. 6 and 7 illustrate typical scenes over land and ocean in which successful OCO-2 retrievals are located between clouds. "

The sentence "Figures 8, 9, and 10 illustrate that 3D cloud effects usually occur within 4 km of clouds, since 1D /3D ratios and raw XCO2 concurrently decrease at nearest cloud distances less than 4 km." is added to the paragraph to be quantitative, since the phrase "in the vicinity of clouds" is ambiguous.

- *Line 586: Recommend "But cloud…" should be "However, cloud…"*

Revised the sentence

- *Line 592: Recommend putting "ratios greater than unity" and "ratios less than unity" in (parenthesis).*

Revised as suggested.

- *Line 594-603: All of this detailed discussion would be better in Sec 5 rather than the Conclusions.*

The paragraph is moved to Sect. 5.

- *Line 610-617: All of this detail might be better in the analysis sections.*

The paragraph is moved to Sect. 5.

- *Line 623-631: Most of this paragraph belongs in analysis section. Or reword each of these multi-sentence paragraphs down into a single conclusive sentence.*

Content from the paragraph is moved from Sect. 8 and moved into Sect. 6. An additional paragraph from the last section was deleted to decrease the length of the paper.

- *Line 632-640: This paragraph should be earlier in the Conclusions. This is good "high level" discussion that sets the tone for why the work is important. But I would avoid citation and analysis of individual Figures in the conclusions, but that maybe personal taste.*

The paragraph is now placed earlier in Conclusions. The Conclusions discussion proceeds mentioning the Figs in numerical order.

- *Line 641-647: This paragraph also should come at the outset of the Conclusions, but in a general way without referring directly to the figures.*

The paragraph is now placed earlier in the Conclusions. The Conclusions discussion proceeds in Figure numerical order.

*Table 1: I cannot tell from the discussion if the simulations were a single iteration of each state parameter by the perturbation amount listed in the table. Or were there multiple iterations of the perturbations to state vector elements? This point needs to be clarified here and in the main text.*

A simulation is a single iteration (other than for the fact that SHDOM iterates for the solution, with the iterations stopping when a specified solution accuracy is specified). Once specified, the model atmosphere is fixed (does not change). The two sentences added to the beginning of section 3.3, the added sentence (N375) "Once specified, the model atmosphere is fixed in a SHDOM simulation" and the phrase "partial derivative" is present in the revised text to avoid confusion. Adding the phrase "single iteration" in Table 1 will confuse everyone.

- *Table 2: I dont love using footnotes in the table. Might be better to describe in the caption? Personal taste?*

Yes, personal taste. Many papers use footnotes. There is a need to clarify the meaning of the terms in the Table. It is hard for a reader to go back and forth from text to table when there are several undefined terms used in a Table. Footnotes were removed in several of the tables (with information placed in the caption).

- *Table 3: Is this table actually helpful in anyway? I feel like it is fairly irrelevant to the analysis. Not sure.*

As remarked above, the information stated in Table 3 allows an interested reader to use NASA Worldview to examine the cloud field of the scenes used to generate Figures 8, 9, and 10.

- *Table 4: For sure the footnote should just be in the table caption.*

Footnote is removed, and ppm is used in the table caption.

- *Figure 2: abscissa label should include "[km]". My interpretation of this figure is that as the observation is made closer to the cloud, the 1D radiances (as modeled by SHDOM) become an increasingly smaller fraction of the SHDOM 3D radiances. This implies that the L2FP modeled 1D radiances would underestimate the true measured radiances if all of the retrieval physics were included and setup correctly. But since that is not the case, the L2FP must modify ancillary parameters such as surface albedo and aerosol in order for the modeled radiances to match the measurements. Presumably, a brightening of the surface, coupled with a lofting and/or loading of the aerosol would be required. If that interpretation is correct, it might be helpful to state that more explicitly in Sec. 3.2 when Fig. 2 is discussed (or elsewhere where relevant). This might help readers with the physical interpretation of how the L2FP retrieval is sensitive to the 3D cloud effect.*

The figure caption now indicates that D is in km.

Yes, as nearest cloud distance D decreases, the ratio of 1D column radiance to 3D cloud effect radiance also decreases.

The true radiances are measured by OCO-2. The true radiances are 3D cloud effect radiances, since the actual atmosphere exchanges photons between adjacent columns.

The OCO-2 operational retrieval just calculates 1D column radiances.

Since the 1D / 3D ratio is less than unity for observations impacted by cloud brightening, and the operational retrieval does not incorporate any 3D cloud effect physics into the retrieval, the retrieval adjusts a variety of variables to bring forward model and observed radiances into agreement.

Yes, "the L2FP must modify ancillary parameters such as surface albedo and aerosol in order for the modeled radiances to match the measurements".

This is an extremely important point that you make, and is illustrated visually in Figures 8, 9, and 10 (revised paper). That is why Figures 8, 9, and 10 are so important to the paper.

The revised paper now includes in section 3.2 the following paragraph (N301) right after the first reference to Figure 2:

"Considering the case where the 1D /3D ratio is less than unity due to 3D cloud effects, the OCO-2 experiment measures the true radiance, which is a 3D radiance since the real atmosphere exchanges photons between adjacent columns. The operational OCO-2 retrieval calculates 1D

column radiances, and inserts no physics due to adjacent column 3D cloud effects. With 1D / 3D less than unity, the retrieval needs to "enhance" the 1D radiance by modifying variables (such as surface reflectance, aerosol, surface pressure, and XCO2) in order to bring forward model and observed radiances into agreement. This is illustrated below in section 5. in relation to Figs. 8, 9, and 10."

- *Figure 5: I already have made a number of comments about interpretation of this figure above. A couple of more technical comments here: the contrast between the white background and the yellow Refl curve is very poor. Recommend changing to another color. Is there any concern about dependence among the state variables as each seems to have been perturbed individually? I still dont understand if this figure reflects a single perturbation of each state vector element, or if multiple perturbations are somehow wrapped in this figure. If the former, then it seems like a logical question to ask if the results are robust to further perturbations. If the latter, then I'm afraid that I'm lost.*

The revised text (N375) at the start of section 3.3 states that "Once specified, the model atmosphere is fixed in a SHDOM simulation." The yellow curve is changed to a green curve.

- *Figure 6: caption says MODIS radiance, while text discussion says SHDOM. I think the latter is in error.*

The text Typo is corrected.

- *Figures 6+7 and 8+9: The utility of these figures would be increased if they were combined into two two panel plots (or even a single 4 panel plot?). Figs 7+9 would be more useful if they displayed the exact same X-Y range, presumably in lat/lon space. This would make them much more comparable. Plus add the OCO individual sounding points (the Xs and Os) to Figs 7+9.*

Figs. 6 and 7, and Figs. 8 and 9 are combined into Figs. 6 and and 7 in the revised paper.

- *Figures 10-15: All of these figures would benefit from subpanel lettering, e.g., (a), (b), (c), which could be directly referenced in the text (rather than using awkward phrases like "the middle right panel in figure X" for example. In fact, I'm pretty sure that subpanel lettering is a requirement for AMT.*

The graphs now have (a), (b), (c), .. labels

- *Figures 10-12: These graphs take up a lot of space, and what are they showing besides that the operational QF procedure works?*

They demonstrate that 1D /3D ratios and raw XCO2 concurrently decrease as nearest cloud distance decreases, especially for the QF=1 data points, at cloud distances less than 4 km. Other variables (dP, AODTOT, Albedo1, Albedo2) also show this behavior. *These figures illustrate how 3D cloud effects manifest themselves in the operational retrieval, for ocean glint, land glint, and land nadir observations.* The Figures, in making a direct correspondence between the 1D /3D ratios, and raw XCO2, indicate that 3D cloud biases are present at the rawXCO2 processing stage and are not just an artifact of the bias correction processing.

- *Figures 13-15: Might be good to utilize color to aid in interpretation. Not strictly necessary.*

These figures are revised figures (placed in a black and white setting) since the AMT editor suggested a reworking of the original color figures.

- *Figure 16-17: The yellow curve does not show up well against the white background.*

The yellow curve is changed to a green curve in the revised paper.

---

## Author Comment (AC2)

Authors response to the RC2 referee's comments on "Insights into 3D cloud radiative transfer for OCO-2" by Steven Massie et al., Atmos. Meas. Tech. Discuss., https://doi.org/10.5194/amt-2022-323-RC2, 2023.

We appreciate very much the helpful RC2 review comments. The indented lines contain our responses to the referee's comments, which we have put in italics. Line numbers of revisions are indicated by (e.g. N806) in the revised paper.

*This paper provides information on the errors three-dimensional radiative processes cause in carbon-dioxide retrievals by the OCO-2 mission. The methodology is appropriate and, although I found a few segments a bit difficult to read, the presentation is generally good. Even so, I do recommend some important changes, mainly to make the study more complete and clearer. My detailed comments are as follows.*

Main comments

*Line 392: Increasing the threshold from 0.3 to 0.6 appears quite arbitrary. Given a such arbitrary increase, it seems unnecessary to even bother determining the initial candidate threshold value of 0.3; it would be easier to just say that a threshold of 0.6 was chosen because that threshold (and the results it yielded) seemed reasonable in some respect (explaining in what respect the 0.6 value seemed to work well).*

> Yes, the increase from 0.3 to 0.6 is arbitrary, and the revised text (N806) in Appendix A) now states this is the case. Since the ratio of cloud shadow to cloud brightening events was surprisingly low, the increase from 0.3 to 0.6 was done to produce a conservative estimate of the cloud shadow frequency. We feel it is useful to mention the factor of 2 adjustment. The conservative adjustment still yielded a low ratio of the number of cloud shadow events to cloud brightening events.

*Lines 238-249, 373, and 644, plus Figures 8 & 9 and maybe other parts of the manuscript: While the manuscript examines a variety of parameters that influence the impact of 3D radiative processes in XCO2 retrievals, it does not discuss a key parameter: cloud optical depth. This is a critically important parameter, as previous studies (not dealing with XCO2 but with other aspects of 3D processes) showed that thicker clouds cast darker shadows and scatter more sunlight into surrounding regions. In other words, optically thicker clouds cause stronger 3D effects. The paper should discuss how cloud optical depth affects the magnitude of 1D/3D ratios and, in a wider sense, the impact of 3D effects on XCO2 retrievals. This includes specifying the cloud optical depth each time some other scene parameters are specified (e.g., for the bar clouds). In complex scenes, statistical parameters such as the scene average optical depth or the optical depth of nearby clouds should be specified.*

> The revised paper now includes (Fig. 4e which is added to the original figure) 1D /3D curves for cloud liquid water content of 0.1, 0.3, and 0.6 gm / m$^3$. While perturbations in a variable (such as LWC or surface reflectance) will noticeably change the 1D and 3D radiances, the perturbations in the 1D / 3D ratio are frequently small. Since OCO2 Lite files have few retrievals influenced by cloud shadowing events, darker shadows play a minor role (N460) in

the archived OCO-2 files.

*Section 7: While this section provides a thorough overview of the findings, it would be very helpful to add a brief overall take-home message. This take-home message should include the implications and/or prospects—for example, that based on the new findings, what (if anything) the authors believe we still need to learn or do about 3D effects in XCO2 retrievals.*

The last paragraph in the revised paper (N778) now states "Future work includes the development of a quick parameterization of 1D / 3D ratios as a function of aerosol and cloud optical depth, given an arbitrary geospatial distribution of clouds. This work will examine a wider range of parameters such as cloud height, aerosol height, aerosol composition, in addition to an examination of scenes not covered in this paper, such as brighter surfaces."

In response to the Reviewer's 1 suggestion, a new Sect. 7 (N659) discusses calculations of zonal averages of 3D cloud radiative effect biases as a function of latitude for bcXCO2 over ocean and land, and illustrates how the biases can be used to mitigate 3D cloud effect biases.

*Section 7: It would help to discuss how the new results relate to the results of earlier studies. For example, it would help to note whether there is any significant discrepancy or reassuring agreement between new and earlier results, and to highlight the instances in which in which the new study adds the most important new information to prior knowledge.*

The revised paper now includes this paragraph (N768) in the Conclusions:
"While Massie et al. (2021) focused on comparisons of bcXCO2 and TCCON, the analysis of 275 times more bcXCO2 data between 2015-2018 (without reference to TCCON data) *enabled calculations of 3D cloud effect biases as a function of latitude. The biases are larger in the southern hemisphere.* This is possibly due to the fact that there are fewer TCCON observations in the southern hemisphere. The magnitude of the 3D cloud effect biases discussed in Massie et al. (2021) and this paper are similar in size, with QF1 biases generally larger than QF0 biases. Since the post-retrieval bias correction process exclusively uses QF0 data, and dP and dPfrac variables, which are correlated with nearest cloud distance (Massie et al. 2021), it is expected that the QF0 biases will be small. The post-retrieval bias correction process *indirectly* accounts for 3D effects, but Fig. 16 and the Table 5 entries indicate that 3D cloud effect biases remain in the Lite file data."

A new paragraph (2$^{nd}$ to last paragraph of the Introduction) is added to the original text (N142) . This discusses how the current goes beyond the content of Massie et al. (2021).

**Other comments about substance:**

*Lines 272-273: The main text or the caption of Fig. 4 should clarify that the figure only shows results for the "cloud brightening" case shown in Fig. 3 (that is, near the right-side cloud). It might also help to include results for the "cloud shadowing" case or to mention briefly how they behave.*

The revised caption of Fig. 4 states that the curves pertain to the cloud brightening situation.

The cloud shadowing case situation was graphed in the same style as Fig. 4 and discussed in the revised paper (N350).

*Lines 318-321: It could be good to briefly mention that—and, if possible, why—the pressure perturbations reach a maximum around a vertical optical depth of 2, after which they decrease again and reach zero at 4.*

Additional code improvements were implemented to produce a new Fig. 5, in order to better answer the Reviewer's question. The improvements deal with a routine that does an improved Lagrange interpolation of ABSCO data in order to specify the ABSCO cross sections associated with each of the SHDOM model altitude levels. In the new Fig. 5, the pressure perturbations do not reach zero at optical depth of 4 in the SCO2 band in the revised paper.

Though not mentioned in the revised text, the derivatives are expected to vary as a function of optical depth since as optical depth increases, the weighting function altitude of the maximum contribution to the top of atmosphere radiance increases in altitude, and doppler contributions increase relative to pressure broadening contributions.

The intended focus of the paper is stated by the following paragraph added to the revised paper (N411):

"The wavelengths selected in Fig. 5 are representative. A different set of wavelengths would produce derivatives, especially for the pressure and $CO_2$ derivatives in the SCO2 band at optical depths greater than two, that differ from those shown in Fig. 5. The key point of Fig. 5 is that the pressure and $CO_2$ derivatives are negative, ranging from 0% to -1%, and are of similar absolute size to the 3D radiance perturbations, which vary from 0% to 3%, for an observation 4 km from the nearest cloud. Figs. 4 and 5 are the only figures in this paper that have information that relates to non-continuum wavelengths."

*Lines 371-372: It is not clear to me how "The prevalence of cloud brightening versus cloud shadowing effects for the Amazon scene is revealed in Table 2.". I can see the fraction of shadowed pixels, but the manuscript should clarify (a) whether retrievals are available for all shadowed pixels, and (b) whether brightening occurs for all pixels that are not shadowed.*

The revised text now states (N460) "Of the 589 successful retrievals for the Table 2 Amazon 150622 case, only eight retrievals (1.3%) are associated with shadows, and the other retrievals (100% – 1.3% = 98.7%) are associated with cloud brightening, assuming that all cloud heights are 8 km in vertical extent." in order to clarify the text.

*Line 373: Does MODIS or OCO-2 indicate which of these cloud heights is closest to what was observed?*

MODIS data can be used on a scene-by-scene basis to specify cloud top heights. This was used for the 36 scenes to generate the input data to SHDOM. For the global analysis we needed to use NASA Giovanni statistics. OCO-2 does not determine the locations of clouds nor cloud top heights. Line 415 of the original paper now states (N468):

"Cloud heights, however, are less than 8 km. Application of NASA Giovanni (https://giovanni.gsfc.nasa.gov/giovanni/) analysis of MODIS MYD08 data files yields histograms (not shown) of cloud top temperatures and pressure means which correspond to cloud top heights between 1 and 2 km for the 150622 and 160622 cases, and heights between 2 and 3 km for the 2016 Amazon and Pacific yearly averages."

*Lines 383-384: The causality (that is, the reason for the word "since") should be clarified in "It is necessary to consider two dozen latitude bins since some bins are fully cloudy, and some*

*bins have relatively few clouds.". Alternatively, the wording should be changed by removing "since".*

The sentence has been rewritten in the revised text (N797):

"Some of the latitude bins have too few clouds, and are excluded since it is of interest to determine clear radiances in the vicinity of clouds. A Clear$_{ave}$ radiance average is calculated from the Clear$_{bin}$ averages when the percentage of clear flags for a latitude bin is greater than 50%."

*Lines 430-434: It appears that in Figure 8 there are a lot of points that have QF1 data even very far to the north from clouds. Is there perhaps a reason not related to clouds that causes more QF1 (and less QF0) retrievals in the areas that happen to be on the north side of the clouds? (Perhaps different wind conditions or stronger glint?*

A visual examination of 12 graphs (similar to Fig. 7) only showed one other ocean glint scene with more QF1 data points north than south of the main clouds (so the new Fig. 7 behavior is not general). The revised text now states (N485):

"This situation is not, however, generally the case, since a visual examination of Figs. similar to Fig. 7 for the other ocean glint scenes listed in Table 3 did not show this behavior. An examination of the NASA Worldview imagery for the Fig. 7 scene did indicate that there are more very small "cloud remnants" north of the main cloud region with a very clear region south of the main cloud. The visual examination of the 12 scenes does indicate that QF1 data points are consistently closer to clouds than the QF0 data points".

*Lines 630-631: It would be interesting to comment on why the land results for the two hemispheres differ from each other.*

The hemispherical asymmetry is another surprising result in our calculations. We do not have a good speculative comment to offer. (V11 processing, in production, uses a different digital elevation map than V10 processing, but these effects kick at high northern latitudes. We feel most comfortable not offering a comment.)

Comments about minor issues

*Line 28: The spaces should be deleted from "1 D" and "3 D".*

Spaces are now deleted in the revised text.

*Line 134: The word "depth" or "thickness" seems to be missing after "vertical optical".*

Typo is corrected.

*Line 200: The wording should probably be changed in "agreement with the observed reflectance", as there is not clear exactly what observed reflectance (i.e., actual OCO2 observation) is referred to. Also, the wording suggests that SHDOM simulations are adjusted and repeated until a simulation using a certain surface albedo yields the expected result.*

The revised text now states (N231):

"The SHDOM calculations do not iterate for the surface reflectance. A constant Lambertian surface reflectance in each band for land observations is specified (hardwired as an input to SHDOM) based on the Lite file retrieved values. These values produce SHDOM 3D top of atmosphere reflectance in good agreement with the observed (archived Lite file) reflectance. For ocean glint observations the Mishchenko and Travis (1997) implementation of the Cox-Munk windspeed dependent surface reflectance formulation is used in the SHDOM calculations, with windspeed specified (hardwired as an input to SHDOM) based on the Lite file retrieval of the windspeed."

As mentioned above, a change in a variable such as surface reflectance will perturb the 1D and 3D radiances, but not so much perturb the 1D / 3D intensity ratios.

*Line 208: For added clarity, I suggest inserting the word "vertically" in front of "constant".*

       Revised as suggested.

*Lines 243-249: I suggest mentioning that cloud effects don't extend past 10 km (e.g., Fig. 2), which means that the periodic boundary conditions used by SHDOM don't cause clear- sky pixels near the left cloud to be impacted by photons that, after being scattered by the right-side cloud, move across the right edge of the scene and reappear at the left edge.*

       The revised text (N296) now states:
"From Fig. 2 it is apparent that 1D /3D ratios asymptote for a length scale of approximately 10 km. The periodic boundary conditions used by SHDOM therefore do not cause clear- sky pixels near the left cloud to be impacted by photons that, after being scattered by the right-side cloud, move across the right edge of the scene and reappear at the left edge. These considerations motivated our selection of the Fig1 geometry and selection of a 32 km by 32 km SHDOM grid."

*Figures 2 and 3: As in Figure 4, the vertical axis label should include the words "intensity ratio".*

       Revised as suggested.

*Figure 5: The vertical axis label should match the notation in Lines 302-303, with R(1D) instead of just 1D.*

       R(1D) is used in the revised figure.

*Line 353: The word "zenith" should be inserted in front of "angle" once or even twice, and perhaps "nadir" could be deleted.*

       The revised text (N432) is now "The direction of the incident sunbeam is from the northwest at the solar zenith angle of 38°, while the OCO-2 sensor angle is 0°."

*Figure 6: The caption could mention that north is at top.*

       Revised as suggested.

*Lines 368-369: I recommend considering a switch from fractions to percentages, for example changing 0.60 to 60%, etc.*

       Revised as suggested throughout the text and Table 2.

*Line 487 and Fig. 10: I guess the units for delta wind should be m/s instead of m. Also, similarly to the top right panel of Fig. 10, the bottom right and middle left panels could also include the units along the Y-axis.*

       Revised as suggested.

*Figures 14 and 15: I recommend moving the ocean glint panels up so they become the top row. This would fit because ocean panels are discussed first (e.g., in Lines 534-537). The ocean results were also discussed first in Section 5 and were displayed first in Fig. 10 (ahead of Figs. 11 & 12). I also suggest clarifying the extent of the three latitude bins (I guess they go from -15 to -5, -5 to 5, and 5 to 15 degree).*

Revised as suggested.

*Lines 556 and 564: The text "as a function of nearest cloud distance" seems to be in error, as Figs. 16 and 17 do not seem to show anything as a function of distance to clouds.*

These typos are corrected in the revised paper.

*Table 4: In the top row, the two "degree signs" (°) are both placed for the South value; one of them should be moved to the North value.*

Revised as suggested.

*Line 605: For consistency with other parts of the sentence, "decreases" should be changed to "decreased"—or perhaps both "decreases" and "decreased" should be changed to "reduced".*

The sentence is revised as suggested.

*Lines 656-657: It should be mentioned whether this sentence is also true for land glint observations and not only for land nadir observations (if we assume that the distance to cloud s very similar for land observations taken at nadir or at a glint direction.*

The new Fig. 11 has both land nadir and land glint curves. They are similar.

---

## Author Response (AR2)

Author's Response: Insights into 3D cloud radiative transfer for OCO-2 (AMT-2022-233) by S. Massie, H. Cronk, A. Merrelli, S. Schmidt, S. Mauceri

Report #2

We greatly appreciate the reviewer catching this error in the Figure 4 caption, and for the reviewer's comments on the first version of the paper. The new revised Fig. 4 caption now states, as suggested, "**Figure 4**. All parts of this figure pertain to a cloud brightening situation near the right-side cloud."

Report #1

We greatly appreciate the reviewer's comments on the first version of the paper. The reviewer's comments significantly improved the paper.